# A capsular myofibroblastic niche maintains hematopoietic stem cells in the spleen

Shubham Haribhau Mehatre [ID][1], Sreelakshmi Sanam [ID][1,3,5], Harsh Agrawal [ID][1,5], Amulya V Hejjaji [ID][1,4], Akhila S Kumar[1], Mohammad Tauqeer Alam [ID][2] & Satish Khurana [ID][1✉]

## Abstract

The spleen is a key site for extramedullary hematopoiesis that hosts a rare population of functional hematopoietic stem cells (HSCs). While the microenvironment that supports extramedullary hematopoiesis response has gained interest, a niche for splenic HSCs at steady-state remains undescribed. Here, we have uncovered a red-pulp-specific, myofibroblastic niche that supports murine splenic HSCs within a ≈ 200-μm-wide capsular zone. Detailed spatial-distribution and perturbation analysis showed the importance of myofibroblasts in maintaining HSCs in a quiescent state. Unlike reported for the adult bone marrow, the HSCs in splenic niche were not spatially associated with vascular components. G-CSF-mediated chemokine alteration and 5-FU-induced proliferation resulted in HSCs shifts away from the splenic capsule. Interestingly, upon regaining quiescence, the HSCs re-occupied niches close to capsular myofibroblasts. Proteomic interactome profiles confirmed the relevance of capsular myofibroblasts for splenic HSCs and identified potential niche regulators of HSC maintenance. Together, this study demonstrates a dynamic HSC localization in the spleen and its niche context at homeostasis and under stress. It offers a model to uncover novel regulators crucial for HSC function.

**Keywords** Hematopoietic Stem Cells; Spleen; Hematopoietic Niche; Myofibroblasts; Proteomics
**Subject Categories** Haematology; Immunology; Stem Cells & Regenerative Medicine

## Introduction

In adult mammals, the process of hematopoiesis starts with quiescent and self-renewing hematopoietic stem cell (HSC) population (Pietras et al, 2015). Tightly regulated quiescence, along with controlled proliferative and differentiation events, is crucial for maintaining steady state hematopoiesis (Bryder et al, 2006). The HSC-extrinsic regulatory factors, cumulatively known as the niche, in conjunction with the intrinsic hallmarks regulate these cellular processes (Crane et al, 2017). It is becoming increasingly clear that HSC niche is anatomically fluid and dynamic space, wherein multiple physical and cellular inputs are integrated by HSCs (Wang and Wagers, 2011). All through the hematopoietic development happening within different anatomical sites, unique cellular and molecular regulators are involved (Morrison and Scadden, 2014). In mammalian embryos, HSCs emerging from the aorta-gonad-mesonephros (AGM) region mature and expand in the fetal liver before transiting to the developing spleen and bone marrow (BM) (Wang and Wagers, 2011). Most studies aimed at understanding the details of the interplay between HSC-intrinsic and -extrinsic factors in the maintenance of HSC function have focussed on BM resident HSCs. The BM HSCs, initially believed to be supported by the osteoblasts (Calvi et al, 2003; Lo Celso et al, 2009; Tamma and Ribatti, 2017) have now been shown to stay in close contact with the vasculature (Acar et al, 2015; Kunisaki et al, 2013) and associated cell types (Méndez-Ferrer et al, 2010). In addition, megakaryocytes (Olson et al, 2013), macrophages (Winkler et al, 2010), T_reg cells (Fujisaki et al, 2011), CXCL12/SDF-1α expressing stromal cells (Wei and Frenette, 2018), neuronal cells (Katayama et al, 2006; Kwan et al, 2016; Pierce et al, 2017) and non-myelinating Schwann cells (Yamazaki et al, 2011) have also been shown to provide molecular regulation to hematopoietic processes. A significant volume of information has accumulated on the architectural details and molecular regulation provided by the HSC niche in the BM under homeostatic conditions.

Apart from the BM, notable size of HSC pool is hosted by the spleen (Kiel and Morrison, 2008) among other tissues that are less significant in hematopoietic activity (Mende and Laurenti, 2021). While the role of splenic HSCs in daily blood cell production is not clearly evaluated, their equivalence with BM HSCs has been demonstrated (Morita et al, 2011). In fact, studies have shown that during development, splenic hematopoiesis is established by the incoming fetal liver and placenta-derived HSCs, even prior to the BM (Christensen et al, 2004; Robin et al, 2009). Unlike in the BM, where most HSCs are maintained in the $G_0$ stage of the cell cycle (Kiel et al, 2005) the adult splenic HSCs have been shown to remain

[1]School of Biology, Indian Institute of Science Education and Research Thiruvananthapuram, Thiruvananthapuram, Kerala 695551, India. [2]Department of Biology, College of Science, United Arab Emirates University, Al-Ain, UAE. [3]Present address: College of Medicine, University of Cincinnati, Cincinnati, OH 45267, USA. [4]Present address: Centre for Molecular Biology of Inflammation (ZMBE), Institute of Cell Biology, University of Muenster, Röntgenstraße 20, D-48149 Münster, Germany. [5]These authors contributed equally: Sreelakshmi Sanam, Harsh Agrawal. ✉E-mail: satishkhurana@iisertvm.ac.in

in pre-activated stage (G$_1$ phase of cell cycle) (Coppin et al, 2018). Impairment in BM hematopoiesis, the 'poised' splenic HSCs engage in extramedullary hematopoiesis (EMH). Physiological states such as pregnancy (Nakada et al, 2014) and aging (Nilsson and Bertoncello, 1994); pathological states such as anemia (Bennett et al, 1968), myeloablation (Morrison et al, 1997), blood loss (Cheshier et al, 2007), and several types of infections (Baldridge et al, 2010) display an active engagement of splenic hematopoiesis. Mobilized BM HSCs are considered to be fundamental to the process of EMH induction (Yang et al, 2020). A spurt in the circulating HSPCs remains consistent in most of the physiological and pathological conditions that lead to the activation of EMH (Chiu et al, 2015). What role do the spleen resident HSCs play during EMH is not well understood. Interestingly, the functional robustness of splenic HSCs with a probable role in maintaining hematopoietic homeostasis was demonstrated (Wolber et al, 2002). However, little is known about the molecular regulation of splenic HSC function.

It has been demonstrated that extrinsic factors play an essential role in the activation of EMH (Short et al, 2019). Overexpression of *Tlx1*, a transcription factor expressed in spleen stromal cells, was sufficient to induce EMH (Oda et al, 2018). Deletion of key hematopoietic factors *Scf* or *Cxcl12* in stromal cells also inhibited EMH, without any effect on BM hematopoiesis (Inra et al, 2015). Studies performed to understand HSC niche in the spleen and extrinsic regulation of splenic hematopoiesis have been restricted to EMH models. Our experiments show that the primitive HSCs (pHSCs) exclusively occupy the sub-capsular niche lined by myofibroblasts. Within the 'hematopoietic zone' the quiescent pHSCs were spaced closer to the splenic capsule that expressed a number of vital hematopoietic factors. Alteration in molecular support or induction of proliferation led to rearrangement of pHSCs with reference to the capsular myofibroblasts specifically. Finally, quantitative proteomic profiling by nano-LC coupled with tandem mass spectrometry revealed a strong molecular interaction between splenic HSCs and myofibroblasts that also expressed ECM components and related proteins significantly more than the stromal cells. Overall, we present the first model of myofibroblastic niche that supports pHSCs at steady state in spleen tissue.

# Results

## Splenic HSCs locate themselves in the red pulp exclusive splenic capsular zone

We aimed at localizing primitive HSC population in the spleen based on well-known cell surface markers. Both in the case of BM and spleen (Kiel et al, 2005), primitive HSCs have been identified as Lin$^-$CD41$^-$c-kit$^+$Sca-1$^+$CD150$^+$CD48$^-$ cell population (hereafter, pHSCs). We first performed flow cytometry analysis to arrive at the best combination of markers that can allow identification of the most pHSC population in a fluorescence microscopy-based immunolocalization protocol (Fig. EV1Ai–Aiv). Splenic mono-nuclear cells (MNCs) were gated for Lin$^-$CD41$^-$CD48$^-$ population (hereafter, 3$^-$; Fig. EV1Ai) followed by selection on the basis of two additional markers among c-kit, Sca-1, and CD150. We observed that 77.5 ± 6.4% (Mean ± SEM) of c-kit$^+$CD150$^+$ cells among 3$^-$ population were Sca-1$^+$ and hence, were identified as pHSCs

(Fig. EV1Aii). In contrast, only 45.6 ± 2.9% of 3$^-$CD150$^+$Sca-1$^+$ (Fig. EV1Aiii) and 48.8 ± 1.5% of 3$^-$Sca-1$^+$c-kit$^+$ (Fig. EV1Aiv) cells were identified as pHSCs (Fig. 1A). Therefore, we concluded that 3$^-$c-kit$^+$CD150$^+$ cells would represent the pHSC most closely using confocal-based fluorescence microscopy methods allowing the addition of niche cell markers. For this purpose, we adapted our earlier published protocol (Biswas et al, 2022) designed to immunolocate pHSCs in fetal liver tissue. Immunostaining is performed on 10 μm thick cross-sections (CS) of spleen tissues using specific antibodies against chosen markers, followed by confocal microscopy-based fluorescence imaging (Fig. 1B). As expected, splenic tissues from healthy animals under homeostatic conditions harbours a rare population of pHSC with a density of 0.26 ± 0.02 cells/mm². Among all the nucleated cells, we estimated their number density to be 28.32 ± 2.01 cells/million. Under EMH conditions, most splenic HSCs were shown to reside within red pulp (Inra et al, 2015). To examine if it was consistent in the spleen at steady state, we performed immunostaining to detect pHSC along with CD169, a marker for marginal macrophages (Martinez-Pomares and Gordon, 2012) to demarcate the red pulp (RP) region (Fig. 1C,D). Our analysis confirmed these results, as no pHSC was detected within the white pulp (WP) of spleen tissues (Fig. 1C–E). Further, we observed that most pHSCs were located near the splenic capsule composed of myofibroblasts expressing α-smooth muscle actin (α-SMA) (Mehatre et al, 2021). To understand the association of splenic pHSCs to the capsular myofibroblasts, we performed immunolocalization studies on pHSCs vis-à-vis the capsular myofibroblasts identified by α-SMA$^+$ staining. Pseudo-surfaces were generated using Imaris, and Euclidean distances were measured between pHSCs and the nearest α-SMA$^+$ myofibroblasts (Fig. 1F). Quantification revealed that 66.6 ± 3.9% of all pHSCs identified were within the 100 μm of distance from the capsule (Fig. 1G). While 32.9 ± 3.8% of pHSCs were located from the 100–200 μm zone from the capsule, only rarely they found beyond this distance (Fig. 1G). To confirm the significance of these findings, we generated random dots (RDs) with 100 iterations of the number of pHSCs counted and distributed all across the CS area of the spleen tissue (Fig. 1H). We then repeated the cell proximity mapping for pHSCs ($n = 3$, $N = 667$) from myofibroblasts and quantified Euclidean distances in comparison to the RDs (Fig. 1I). Results from these experiments confirmed our observations as the mean Euclidean distance of pHSCs 68.8 ± 1.9 μm from the nearest myofibroblasts was significantly less than the distance calculated for the RDs 91.8 ± 0.5 μm (Fig. 1I). Next, we sub-divided the capsular zone further into the sequential intervals of 30 μm and quantified the proportion of pHSCs in comparison to the frequency of RDs in each interval. We noted a clear preference of pHSCs to occupy zones closer to the splenic capsule, where 30.9 ± 5.0% and 31.1 ± 3.4% of pHSCs occupied the first two intervals, significantly higher than the RDs (Fig. 1J). Hence, 62.2 ± 4.8% of the pHSCs were located within a 0–60 μm zone from the capsular myofibroblast lining of the spleen compared to 34.0 ± 1.2% of RDs. Importantly, as we go away from the capsule, the pHSCs showed a negative association as compared to the RDs, showing indeed capsular zone was the preferred niche (Fig. 1J). We also tested if a more heterogeneous population identified as 3$^-$c-kit$^+$ cells (hereafter, HSPCs) showed a difference in distribution (Fig. EV1B). We observed a similar distribution pattern of HSPCs in the splenic tissue, with 99.6 ± 0.33%

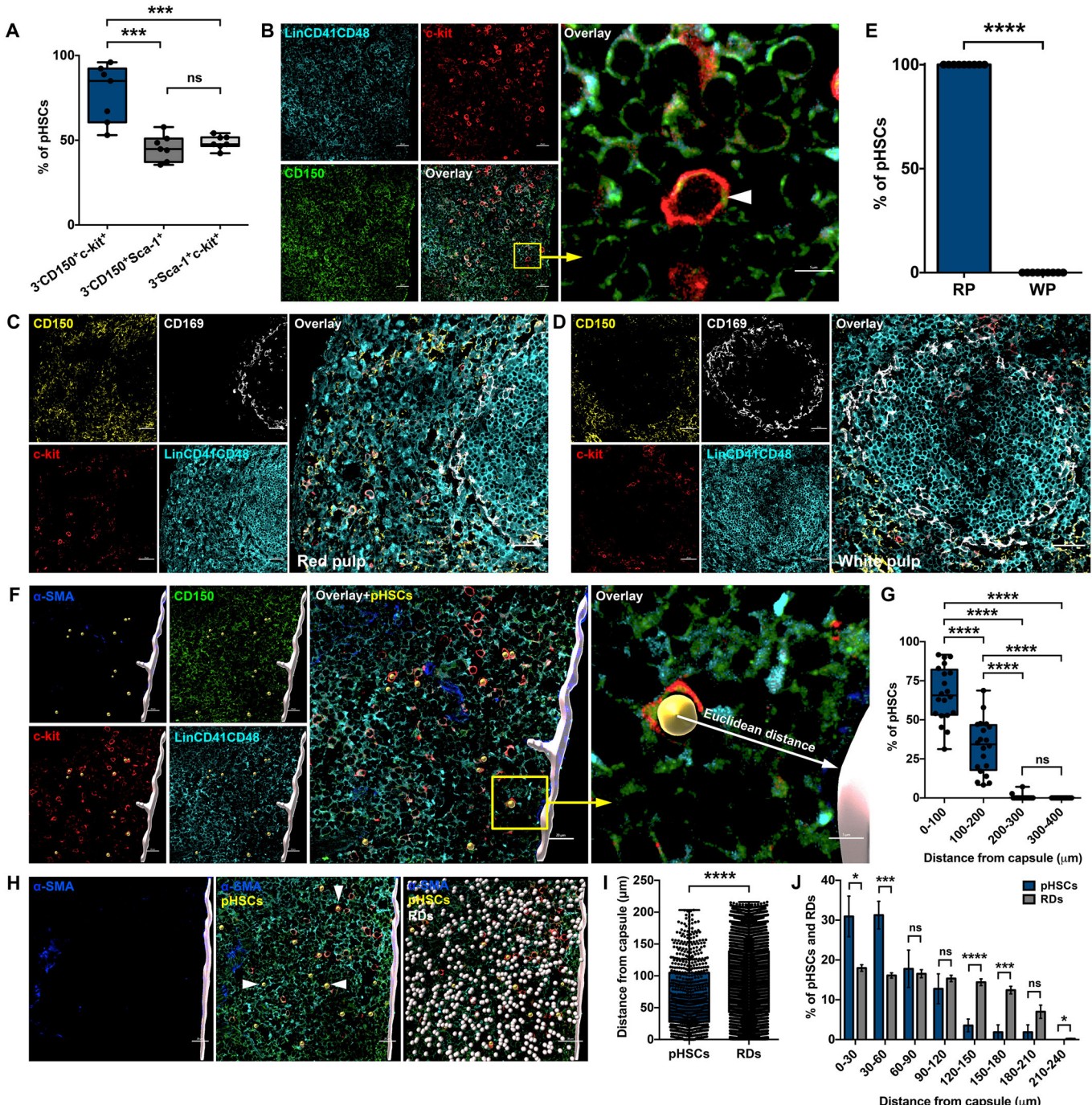

HSPCs located within the 200 μm sub-capsular hematopoietic zone (Fig. EV1C). As observed for pHSCs, the mean distance of HSPCs from myofibroblasts (75.9 ± 1.8 μm) was significantly lesser than the RDs (93.2 ± 0.5 μm) (Fig. EV1D,E). Importantly, we noted that their preferential distribution in comparison to the RDs was prominent in 30–90 μm intervals, in comparison to pHSCs that occupied the 0–60 μm intervals (Figs. 1J and EV1F). These results were further corroborated by spatial mapping, where we noted that pHSCs were positioned closer to the capsule (68.78 ± 1.9 μm) than the HSPCs (75.9 ± 1.8 μm) (Figs. 1I and EV1E). These observations

strongly suggest a functional role of the capsular myofibroblastic niche in regulating stemness.

## Proximity to myofibroblasts ascertains quiescence in splenic HSCs in male and female mice

Our experiments showed closeness of pHSCs to capsular myofibroblasts in spleen tissues. As sex-differences can be crucial in affecting the function of HSCs, we enquired if there were any differences in the spatial distribution of pHSCs in male and female

◀ **Figure 1. Splenic HSCs are spatially associated with α-SMA⁺ capsular myofibroblasts.**

(A) Flow cytometry was performed on splenic MNCs to select the marker combination to identify primitive HSCs (pHSCs or CD150⁺CD41⁻CD48⁻LSK cells) using three fluorophores. The proportion of pHSCs in Lin⁻CD41⁻CD48⁻(or 3⁻)CD150⁺c-kit⁺, 3⁻CD150⁺Sca-1⁺ and 3⁻Sca-1⁺c-kit⁺ cells was examined and plotted ($n = 7$). (B) Representative image showing confocal imaging based immunolocalization of pHSCs in a splenic cross-section. The pHSCs were identified as 3⁻CD150⁺c-kit⁺ cells; one single pHSC is shown in magnified inset. Scale bars, 20 μm (left panel) and 5 μm (right panel). (C) Confocal-based immunolocalization of pHSCs in splenic red pulp (RP). The white pulp (WP) boundary was identified by immunostaining for CD169⁺ marginal zone macrophages along with pHSC markers (scale bar = 30 μm). (D) Representative confocal image showing an entire WP zone bordered with marginal zone macrophages identified using immunostaining for CD169 as mentioned in (C) (scale bar = 30 μm). (E) Comparison of proportion of pHSCs within the RP and WP areas of the spleen. Primitive HSCs were identified using immunostaining and localized with reference to CD169 expression, marking the periphery of WP ($n = 9$). (F) Confocal imaging to localize pHSCs along with capsular and trabecular myofibroblasts in spleen sections. Immunostaining was performed for myofibroblast cell marker α-SMA, along with the markers to identify pHSCs. Pseudo-surfaces for pHSCs (illuminated yellow) and capsular myofibroblasts (illuminated white) were generated using Imaris. The Euclidean distances from surface of pHSCs with respect to the nearest observable capsular myofibroblast in the spleen was determined. Scale bars = 20 μm (left panel), 5 μm (right panel). (G) Spatial distribution frequency of pHSCs in sequential intervals of 100 μm relative to capsular myofibroblasts ($n = 4$; $N = 20$ images). (H) Confocal imaging to immunolocalize pHSCs along with α-SMA⁺ capsular myofibroblasts. Euclidean distances relative to splenic capsule were measured for each pHSC detected, and an equivalent number of RDs with 100 iterations employed (scale bar = 20 μm). (I) Comparison of Euclidean distances measured for pHSCs and RDs with reference to the nearest observable of α-SMA⁺ capsular myofibroblast in splenic sections ($n = 3$, $N = 667$ pHSCs; each dot represents a pHSC or an RD). (J) Distribution frequency of pHSCs and RDs at sequential intervals (30 μm each) from the splenic capsule identified by α-SMA immunostaining. The pHSCs were identified as 3⁻c-kit⁺CD150⁺ cells in the splenic tissue, and the surfaces and RDs were generated using Imaris ($n = 3$, $N = 667$ pHSCs). Data is presented as bar graph (mean ± SEM) in panels (E, J) or box-whiskers plot (median with min to max) in panels (A, G, I). The *p*-value in figures (A, E, G, I, J) were calculated by the Student's *t*-test, *$p < 0.05$, ***$p < 0.001$, ****$p < 0.0001$, and ns $p > 0.05$.

mice. Spatial mapping showed that pHSCs in males ($n = 3$, $N = 320$) were significantly farther (113.0 ± 2.9 μm) from the splenic capsule than pHSCs in the female mice (68.8 ± 1.9 μm; $n = 3$, $N = 667$) (Fig. 2A). Detailed analysis of pHSC distribution at sequential distance intervals confirmed that a higher proportion of pHSCs were localized closer to the capsular myofibroblasts in females (Fig. 2B). These results were consistent when we analyzed the localization of HSPC population with reference to the capsular lining (Fig. 2C). Notwithstanding these differences, the preferential localization of pHSCs in the capsular hematopoietic zone remained consistent in male and female mice.

As our experiments presented a hematopoietic niche wherein pHSCs and HSPCs were located differentially with regard to the myofibroblast lining, we next asked if this relocation was connected with the proliferation state of the cells. We tested if the proliferative state of pHSCs determined their spatial distribution within the sub-capsular zone. Lack of expression of Ki-67 was used to distinguish quiescent (Fig. 2D, left panel) from the proliferative (Fig. 2D, right panel) pHSCs and HSPCs (Fig. 2E). To perform cell proximity analysis, we used splenic capsule as the reference point for hematopoietic zone in these experiments. These experiments showed that among all the pHSCs ($n = 4$, $N = 557$) identified 56.8 ± 5.4% were Ki-67⁺ and were considered proliferative (Fig. EV1G). Spatial mapping showed that proliferative pHSCs ($n = 4$, $N = 309$) were significantly farther (77.0 ± 2.9 μm) from the splenic periphery than the quiescent pHSCs ($n = 4$, $N = 248$) (65.6 ± 3.3 μm) (Fig. 2F). Similar results were obtained when this analysis was performed on Ki67⁻ (Fig. 2E, left panel) and Ki-67⁺ (Fig. 2E, right panel) HSPC population. Interestingly, we observed that the proportion of proliferative cells in HSPC population (53.7 ± 4.9%; Fig. EV1G) was similar to what we observed in the case of pHSCs (56.8 ± 5.3%). Furthermore, for HSPC population also we noted that quiescent HSPCs were located significantly closer (68.8 ± 3.0 μm) to the splenic capsule than the proliferative HSPCs (80.1 ± 2.6 μm) (Fig. 2G). Hence, these results show that the spatial proximity of quiescent splenic pHSCs, as well as HSPCs, to the capsular layer is greater than their proliferative sub-populations. It is important to note that performing this analysis on HSPCs (3⁻c-kit⁺ cells) will allow the use of additional

fluorophore for niche marker in addition to Ki-67 immunostaining. While we observed that the mean distance for proliferative pHSCs as well as HSPCs was greater than the quiescent sub-population, their distribution within more refined proximal distance intervals from splenic capsule did not differ significantly (Fig. 2H,I).

## Vascular architecture of blood and lymphatic vessels does not correlate with the HSC distribution pattern

The HSC niche in the adult BM is known to have important contribution from the vasculature (Acar et al, 2015; Kunisaki et al, 2013) and associated structures (Méndez-Ferrer et al, 2010). Given this, we first investigated the distribution of blood and lymphatic vessels in the spleen tissue. We performed immunostaining for pan-endothelial marker CD31 along with α-SMA that marks vascular smooth muscle cells and myofibroblasts, and CD169 to distinguish between RP and WP areas. Tile scans of the splenic tissue CS were obtained (Fig. 3A; middle panel) to analyze the distribution of blood vessel in the RP (Fig. 3A; left panel) and WP areas (Fig. 3A; right panel). We noted that the blood vessel density was significantly higher in the WP compared to the RP areas (Fig. 3B). Considering pHSC as well as HSPC populations were exclusively localized in the RP, these results indicated that blood vessels might not play important role in their distribution in spleen tissues. Further, we sub-divided splenic RP area into two regions, peripheral 200 μm wide capsular hematopoietic zone and the rest of the inner splenic core. We noticed that there was no preferential distribution of blood vessels in the quiescent capsular zone where pHSCs were predominantly localized (Fig. 3C). We extended these studies to the lymphatic vessels and enquired if their distribution was correlated with the distribution of HSC populations in the spleen tissue. To this end, we performed immunostaining for Lyve-1, a lymphatic endothelial marker (Fig. EV2A). Our results demonstrate that lymphatic vessels (LV) that were rare in appearance, were detected in the RP areas with a significantly lower density than the blood vessels (BV; Fig. EV2B). In addition, we noted that the LVs were completely absent in the peripheral capsular zone, home for the quiescent pHSCs (Fig. EV2C). These

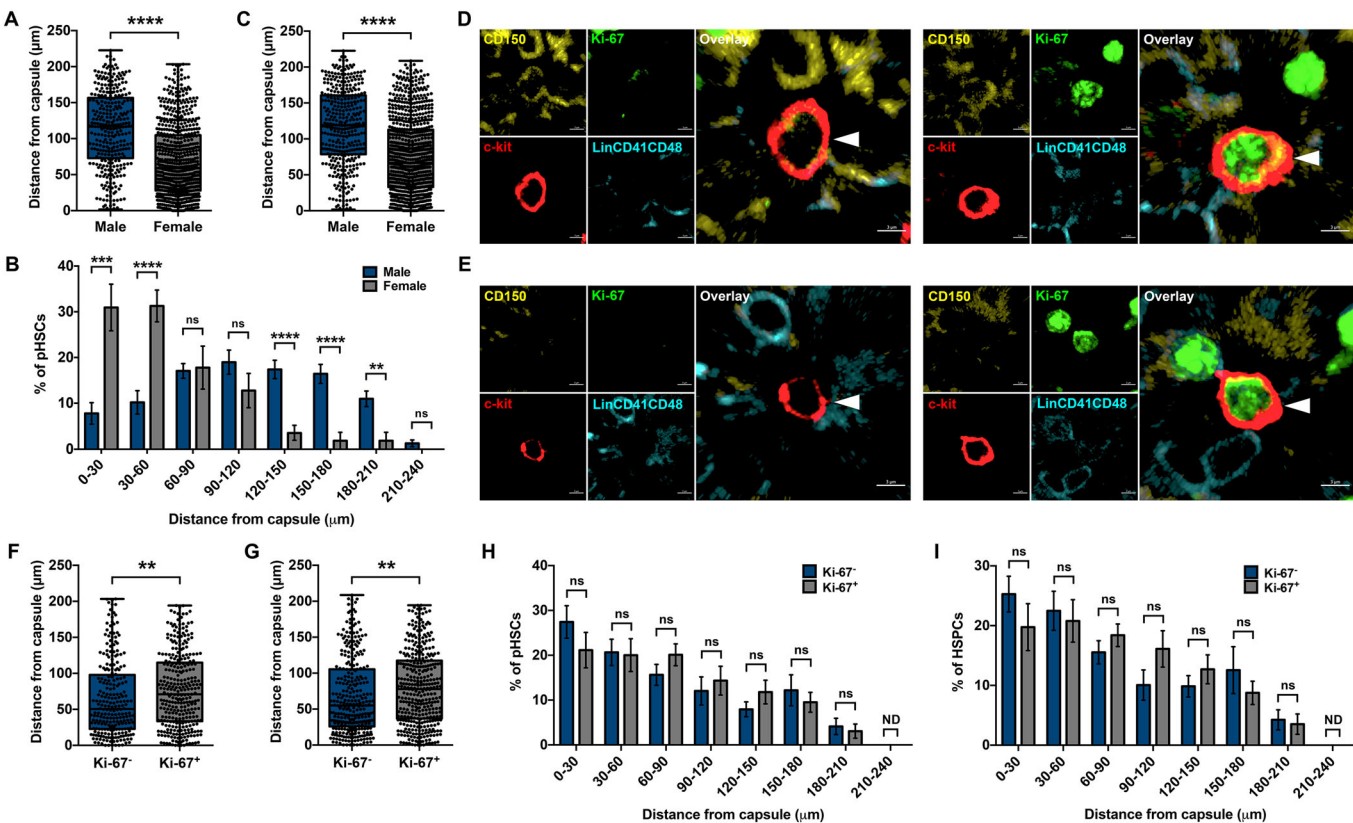

**Figure 2. Proliferation state-dependent localization of HSCs within splenic capsular niche.**

(A) Euclidean distances calculated for each pHSC with reference to pseudo-surfaces of capsular myofibroblasts in the spleen tissues harvested from male and female mice. Each dot represents a pHSC immunolocalized as a 3⁻c-kit⁺CD150⁺ cell by confocal imaging ($n = 3$, N; male = 320, female = 667 pHSCs). (B) Distribution frequency of pHSCs at sequential distance intervals (30 μm each) relative to the splenic capsule in spleen tissues isolated from male and female mice ($n = 3$, N; male = 320, female = 667 pHSCs). (C) Euclidean distances calculated for HSPCs relative to the capsular surfaces of spleen tissues from male and female mice ($n = 3$, N; male = 367, female = 818 HSPCs). (D) Confocal images showing quiescent (Ki-67⁻; left panel) and proliferative (Ki-67⁺; right panel) pHSCs identified by immuostaining (scale bar = 3 μm). (E) Confocal images showing quiescent (Ki-67⁻; left panel) and proliferative (Ki-67⁺; right panel) HSPCs identified by immunostaining (scale bar = 3 μm). (F) Comparison of Euclidean distances between Ki-67⁻ and Ki-67⁺ pHSCs from the nearest detectable capsular myofibroblast. Immunostaining was performed to detect Ki-67 expression in 3⁻c-kit⁺CD150⁺ cells, surface generation and distance analysis was performed on Imaris ($n = 4$, N; Ki-67⁻ = 248, Ki-67⁺ = 309 pHSCs). (G) Comparison of Euclidean distances between Ki-67⁻ and Ki-67⁺ HSPCs from the nearest detectable capsular myofibroblast. Immunostaining was performed to detect Ki-67 expression in 3⁻c-kit⁺ cells ($n = 4$, N; Ki-67⁻ = 307, Ki-67⁺ = 363 HSPCs). (H) Comparison between the frequency of Ki-67⁻ and Ki-67⁺ pHSCs for their distribution within the sequential distance intervals (30 μm each) relative to the capsular surface in spleen tissues ($n = 4$, N; Ki-67⁻ = 248, Ki-67⁺ = 309 pHSCs). (I) Distribution frequency of Ki-67⁻ and Ki-67⁺ HSPCs at sequential distance intervals relative to the splenic capsule in spleen tissues ($n = 4$, N; Ki-67⁻ = 307, Ki-67⁺ = 363 HSPCs). Data is presented as bar graph (mean ± SEM) in panels (B, H, I) or box-whiskers plot (median with min to max) in panels (A, C, F, G). The p-value in figures (A–C, F–I) were calculated by the Student's t-test, **p < 0.01, ***p < 0.001, ****p < 0.0001, and ns p > 0.05, and ND indicates not detected.

findings suggest that unlike BM, vascular structures do not play a significant role in maintaining pHSCs in spleen tissue.

To further check if there was any possible correlation between the vascular structures and splenic HSPC population, we performed immunostaining based spatial analysis for HSPCs with reference to the vasculature identified by immunostaining for CD31 (Fig. 3D). Our results showed no significant difference in the mean distance (Fig. 3E) or spatial distribution (Fig. 3F) of HSPCs relative to the blood vessels when compared with the random dots. In the BM niche, it has been reported that the quiescent and proliferative HSCs occupy differential location with reference to the arterioles and Ng2⁺ periarteriolar cells (Kunisaki et al, 2013). Therefore, we next analyzed if proliferative (Ki-67⁺) and quiescent (Ki-67⁻) HSCs were differentially

localized with reference to the vascular structures (Fig. 3G–I). The results from these experiments again showed no difference in the mean distance (Fig. 3H) or spatial distribution (Fig. 3I) of proliferative versus quiescent HSPCs around the vascular structures. Although we found no correlation between the distribution of lymphatic vessels with the capsular hematopoietic niche, we sought to examine the spatial localization with reference to lymphatic vessels (Fig. 3J). The results showed that HSPCs were not preferentially localized with reference to the lymphatic vessels, as observed in the mean distance (Fig. 3K) and spatial distribution analysis (Fig. 3L). Overall, through this extensive analysis our results found no involvement of splenic vasculature in influencing localization and distribution of HSC population.

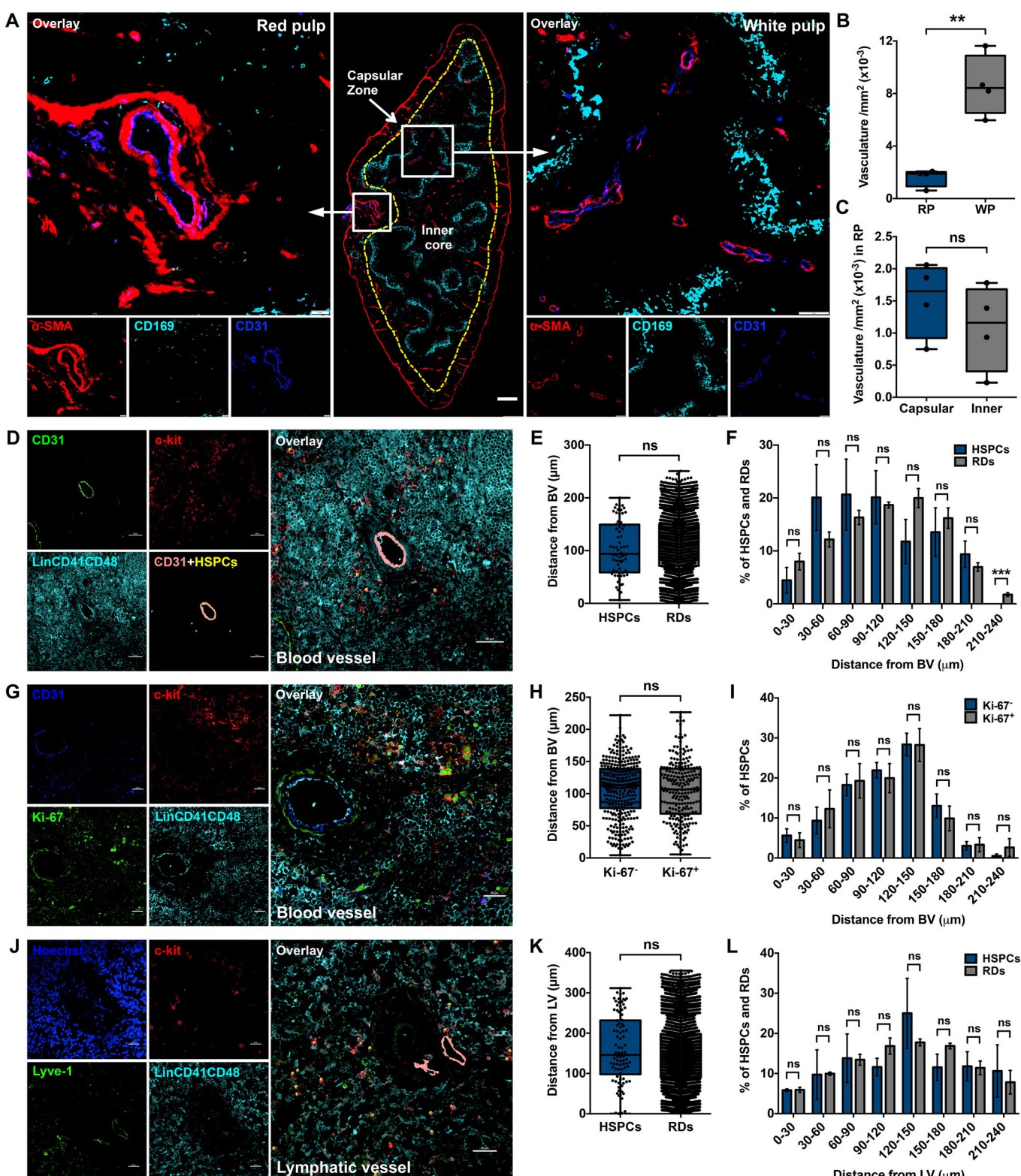

## Splenic myofibroblasts are enriched in SDF-1α expression

Chemokine CXCL12/SDF-1α has been identified as a key extrinsic regulatory factor in the maintenance and functioning of HSCs in BM. In fact, its expression has been used to identify

the cellular components of the hematopoietic niche in the BM (Ding and Morrison, 2013), fetal liver (Dong et al, 2021), and EMH-activated spleen (Inra et al, 2015). Therefore, we examined the spatial localization of SDF-1α expression in the spleen tissue by performing immunostaining. As the pHSCs

**Figure 3. Splenic vessels do not contribute to hematopoietic niche creation.**

(A) Immunostaining was performed to detect vasculature within WP and RP areas in the spleen, and tile scans of splenic CS were acquired. Splenic CS is divided into 200 μm wide peripheral capsular zone and inner core. Marginal macrophages lining the WP areas were immunostained for CD169 along with pan-endotheilal marker CD31 and vascular smooth muscle cell marker α-SMA (Middle panel; scale bar = 200 μm). Left panel; zoomed confocal images showing a large blood vessel in the RP of the splenic CS (scale bar = 20 μm). Right panel; zoomed confocal images showing blood vessels in the WP of splenic CS with (scale bar = 50 μm). (B) Comparison of vascular distribution within the RP and WP areas of the spleen. Blood vessels were identified using immunostaining and localized with reference to the CD169 demarcating WP from RP. Proportion of vascular area normalized to RP and WP areas in spleen tissues was compared ($n = 4$). (C) Comparison of vascular distribution within the 200 μm wide capsular zone and inner core of the spleen in RP area. Blood vessels were identified using immunostaining and localized with reference to the CD169. Proportion of vascular area normalized to the areas covered in capsular zone and inner splenic core was compared ($n = 4$). (D) Confocal imaging to localize HSPCs along with the vascular endothelial cells in spleen sections. Immunostaining was performed for vascular endothelial marker CD31, along with markers to identify HSPCs. Pseudo-surfaces for HSPCs (illuminated yellow) and blood vessels (illuminated salmon) were generated using Imaris. The Euclidean distance from surfaces of HSPCs and RDs with respect to the nearest observable blood vessel in the spleen was determined. Scale bars = 40 μm. (E) Comparison of Euclidean distances measured for HSPCs and RDs with reference to the nearest observable of CD31$^+$ blood vessel in splenic sections ($n = 3$, $N = 64$ HSPCs; each dot represents a HSPC or an RD). (F) Distribution frequency of HSPCs and RDs at sequential intervals (30 μm each) from the blood vessels identified by CD31 immunostaining ($n = 3$, $N = 64$ HSPCs). (G) Confocal images of spleen CS immunostained to localize Ki-67$^+$ and Ki-67$^-$ HSPCs along with CD31$^+$ blood vessels. The HSPCs were identified as 3$^-$c-kit$^+$ cells (scale bar = 20 μm). (H) Comparison of Euclidean distances between Ki-67$^-$ and Ki-67$^+$ HSPCs from the nearest detectable blood vessel ($n = 8$, N; Ki-67$^-$ = 337, Ki-67$^+$ = 239 HSPCs). (I) Comparison of the frequency of Ki-67$^-$ and Ki-67$^+$ HSPCs within sequential distance intervals (30 μm each) relative to the blood vessels in spleen tissues ($n = 8$, N; Ki-67$^-$ = 337, Ki-67$^+$ = 239 HSPCs). (J) Confocal imaging to localize HSPCs along with lymphatic vessels in spleen sections. Immunostaining was performed for lymphatic vessel marker Lyve-1, along with markers to identify HSPCs. Pseudo-surfaces for HSPCs (illuminated yellow) and lymphatic vessels (illuminated salmon) were generated using Imaris (scale bars = 30 μm). (K) Comparison of Euclidean distances measured for HSPCs and RDs with reference to the nearest observable Lyve-1$^+$ lymphatic vessel in splenic sections ($n = 4$, $N = 90$ HSPCs; each dot represents a HSPC or an RD). (L) Distribution frequency of HSPCs and RDs at sequential intervals (30 μm each) from the lymphatic vessels ($n = 4$, $N = 90$ HSPCs). Data is presented as bar graph (mean ± SEM) in panels (F, L) or box-whiskers plot (median with min to max) in panels (B, C, E, H, K). The p-value in figures (B, C, E, F, H, I, K, L) were calculated by the Student's t-test, **$p < 0.01$, ***$p < 0.001$, and ns $p > 0.05$.

were exclusively located in the splenic RP, we examined SDF-1α expression in context of RP versus WP zones using immunostaining for CD169 and pan-leukocyte antigen CD45 (Fig. 4A). We performed tile scans of the entire CS of spleen tissue, and a gross analysis showed that the expression of SDF-1α was mostly restricted to the RP in non-hematopoietic (CD45$^-$) cells (Fig. 4B, left panel) while WP showed rare SDF-1α expressing cells (Fig. 4B, right panel). A closer analysis showed that within the RP region, SDF-1α expression was associated with the capsular, trabecular, and vascular regions of the spleen tissue (Figs. 4B and EV2D). To confirm this, we performed immunostaining for α-SMA along with SDF-1α and observed an almost complete association, where its expression was restricted to capsular and trabecular myofibroblasts (Fig. EV2E) and vessel-associated smooth muscle cells (Fig. 4C). We observed SDF-1α signals from extracellular spaces also, as it has been shown to associate with ECM components (Kunz and Schroeder, 2019). Next, to confirm that SDF-1α expression was restricted to the fibrous tissue of the spleen, we performed gene expression analysis using quantitative RT-PCR. Data normalized with the expression of Sdf-1α in lineage-depleted BM cells showed that the bulk of its expression in spleen came from the fibrous tissue composed of myofibroblasts (Fig. 4D). To further confirm the involvement of myofibroblasts in the creation of hematopoietic niche in the spleen, we examined the expression of several other factors involved in hematopoiesis in comparison to the lineage-depleted splenic pulp cells (Fig. EV2F). We observed robust expression of factors such as vascular cell adhesion molecule 1 (Vcam-1), erythropoietin (Epo), stem cell factor (Scf), periostin (Postn), and Sdf-1α (Fig. EV2F,G) in the fibrous tissue. Interestingly, we did not detect the expression of thrombopoietin (Tpo) in splenic tissue, and the expression of angiopoietin-1 (Ang1), Il-6, and Icam-1 was lower in myofibroblasts than in spleen stroma (Fig. EV2F,G).

## G-CSF modulates molecular profile of hematopoietic niche in spleen

As we observed robust expression of SDF-1α in myofibroblasts in the spleen, we decided to target its expression to confirm the relevance of capsular hematopoietic niche in HSC maintenance. It has been shown that G-CSF induces HSC mobilization and activation of splenic EMH through reduction of SDF-1α expression (Petit et al, 2002; Semerad et al, 2005). We performed qRT-PCR to first test if G-CSF inhibited the expression of Sdf-1α expression in splenic pulp and fibrous myofibroblasts. We noted a robust decrease in Sdf-1α expression in lineage-depleted splenic pulp cells as well as in myofibroblasts (Fig. EV2H). In addition to Sdf-1α, the expression of other key hematopoietic regulators such as Vacm-1, Scf, IL-6, Postn, and Icam-1 was also decreased in the myofibroblasts (Fig. EV2I,J). Although these factors were found to be expressed at a lower level in the non-hematopoietic splenic pulp cells, the transcript level of these factors, notably Scf, Postn, and Sdf-1α was nevertheless reduced significantly (Fig. EV2K,L). Therefore, G-CSF-mediated reduction in Sdf-1α and other hematopoietic factors provided us a model to test the role of the myofibroblastic niche in hosting the HSC population.

Next, we wished to understand the effect of G-CSF treatment on the hematopoietic function in spleen. In confirmation of EMH induction, G-CSF treatment led to an increase in the weight (Appendix Fig. S1A), and cellularity of the spleen (Appendix Fig. S1B), also reflected in an increase in the CS area (Appendix Fig. S1C,D). In addition, we noted disorganization in the WP structure of the spleen with disruption of marginal zone reflected through CD169 immunostaining for marginal zone macrophages (Appendix Fig. S1C). Quantification of RP and WP areas showed a significant increase in the RP area with no change in the WP area (Fig. 4E), which resulted in an overall increase in the RP/WP ratio (Appendix Fig. S1E). We next examined the change in splenic HSC population following G-CSF induced EMH and performed flow cytometry analysis on the

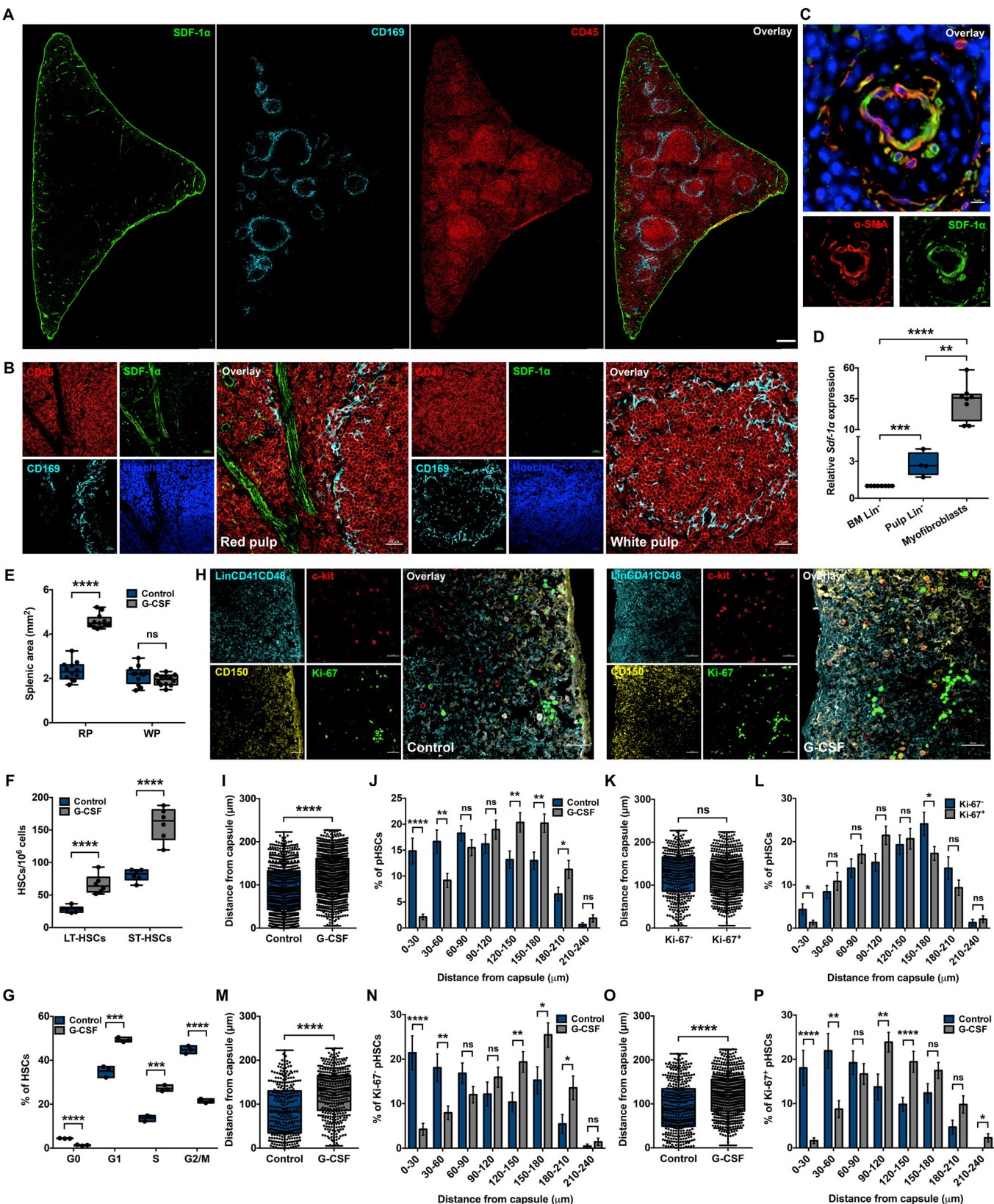

◄

**Figure 4. G-CSF mediated proliferation shifts HSCs away from capsular myofibroblasts.**

(A) Immunostaining was performed to detect SDF-1α with reference to WP and RP areas in the spleen, and tile scans of splenic CS were acquired. Marginal macrophages lining the WP areas were immunostained for CD169 along with SDF-1α and pan-leukocyte antigen CD45 (scale bar = 200 µm). (B) Confocal images showing SDF-1α expression in RP (left panel) and WP (right panel) areas in spleen tissue. Immunostaining was performed to detect the expression of CD169 marking the periphery of WP areas. The right panel shows an entire WP zone in the splenic section (scale bar = 20 µm). (C) Detection of SDF-1α expression in vascular smooth muscle cells identified by α-SMA expression. Confocal images show a vessel in the splenic section with immunostaining performed for SDF-1α along with α-SMA and nuclear staining with Hoechst 33342 (scale bar = 5 µm). (D) Comparison of *Sdf-1a* transcript levels in splenic Lin⁻ cells and myofibroblasts from the fibrous tissue relative to the BM Lin⁻ cells. Quantitative RT-PCR was performed on the RNA isolated from the murine spleen cells, and the graph shows relative gene expression (*n* = 4, 8). (E) Comparison of RP and WP areas in spleen CS following G-CSF treatment. WP areas were detected by immunostaining for CD169. RP areas were quantified exclusive of WP, vasculature, trabecular, and capsular areas (*n* = 4, *N* = 12 tile scans). (F) Flow cytometry based quantification of LT-HSC (Lin⁻CD41⁻CD48⁻Sca-1⁺c-kit⁺CD150⁺ cells) and ST-HSC (Lin⁻CD41⁻CD48⁻Sca-1⁺c-kit⁺CD150⁻ cells) frequency in spleen with and without G-CSF treatment (*n* = 6). (G) Flow cytometry analysis to examine the cell cycle status of HSCs (Lin⁻CD41⁻CD48⁻Sca-1⁺c-kit⁺ cells) in the spleen with and without G-CSF treatment. DAPI and Ki-67 based analysis for $G_0$, $G_1$, S, and $G_2$M stages of cell cycle was performed along with immunostaining for HSC markers (*n* = 3). (H) Confocal images of spleen CS immunostained to localize Ki-67⁺ and Ki-67⁻ pHSCs. The spleen tissues were taken from mice treated with (right panel) or without (left panel) G-CSF, and the pHSCs were identified as 3⁻c-kit⁺CD150⁺ cells (scale bar = 30 µm). (I) Euclidean distances calculated for each pHSC with reference to pseudo-surfaces of capsular myofibroblasts in the spleen with or without G-CSF treatment. Each dot represents a pHSC immunolocalized as a 3⁻c-kit⁺CD150⁺ cell by confocal imaging (*n* = 6, *N*; control = 707, G-CSF = 957 pHSCs). (J) Distribution frequency of pHSCs at sequential distance intervals (30 µm each) relative to the splenic capsule in spleen tissues with or without G-CSF treatment (*n* = 6, *N*; control = 707, G-CSF = 957 pHSCs). (K) Comparison of Euclidean distances for Ki-67⁻ and Ki-67⁺ pHSCs relative to the nearest capsular surfaces detected in spleen sections from G-CSF treated mice (*n* = 6, *N*; Ki-67⁻ = 404, Ki-67⁺ = 553 pHSCs). (L) Comparison between Ki-67⁻ and Ki-67⁺ pHSCs for their distribution frequency within the sequential distance intervals relative to capsular surface in spleen tissues after G-CSF treatment (*n* = 6, *N*; Ki-67⁻ = 404, Ki-67⁺ = 553 pHSCs). (M) Euclidean distances calculated for Ki-67⁻ pHSCs relative to the nearest observable capsular myofibroblast of spleen tissues from the control and G-CSF treated mice (*n* = 6, *N*; Control = 301, GCSF = 404 Ki-67⁻ pHSCs). (N) Distribution frequency of Ki-67⁻ pHSCs at sequential distance intervals relative to the capsular surface in spleen tissues with or without G-CSF treatment (*n* = 6, *N*; Control = 301, GCSF = 404 Ki-67⁻ pHSCs). (O) Euclidean distances calculated for Ki-67⁺ pHSCs relative to the capsular surfaces of spleen tissues treated with or without G-CSF (*n* = 6, *N*; Control = 406, G-CSF = 553 Ki-67⁺ pHSCs). (P) Distribution frequencies of Ki-67⁺ pHSCs at different distance intervals relative to the capsular surface in spleen tissues with or without G-CSF treatment (*n* = 6, *N*; Control = 406, G-CSF = 553 Ki-67⁺ pHSCs). Data is presented as bar graph (mean ± SEM) in panels (J, L, N, P) or box-whiskers plot (median with min to max) in panels (D–G, I, K, M, O). The *p*-value in figures (D–G, I–P) were calculated by the Student's *t*-test, *\*p* < 0.05, *\*\*p* < 0.01, *\*\*\*p* < 0.001, *\*\*\*\*p* < 0.0001, and ns *p* > 0.05.

mononuclear cells (MNCs). Analysis was performed to compare the long-term (LT-) and short-term (ST-) HSC populations identified as CD150⁺CD48⁻ and CD150⁻CD48⁻ cells gated on Lin⁻Sca-1⁺c-kit⁺ (LSK), respectively (Appendix Fig. S1F). As reported before (Oguro et al, 2017; Rajendiran et al, 2020), we noted a robust increase in the frequency (Fig. 4F) as well as the overall number (Appendix Fig. S1G) of LT-HSCs and ST-HSC population following EMH induction. We could link this increase in HSC population with the cell cycle status. We performed flow cytometry analysis to examine the effect of EMH induction on the cell cycle status of a broader HSC population identified as CD41⁻CD48⁻ LSK cells (Appendix Fig. S1H). Based on the Ki-67 and DAPI staining, we identified different cell cycle stages and observed a clear decrease in the $G_0$ population and an increase in the $G_1$ and S stages (Fig. 4G). We continued with detailed characterization of the effect of G-CSF infusion on the maintenance of HSC population by performing methylcellulose-based colony formation assays on BM (Fig. EV3A–F), peripheral blood (PB; Fig. EV3G–L) and spleen (Fig. EV3M–R). While there was minimal effect on the colony-forming unit-cells (CFU-Cs) in the BM MNC population, a major increase was observed in the MNCs isolated from PB and spleen. In the BM, without any change in CFU-M (Fig. EV3B), BFU-E (Fig. EV3D), CFU-GEMM (Fig. EV3E), and total CFU-Cs (Fig. EV3F), we noted a modest but a significant increase in the CFU-G (Fig. EV3A) and CFU-GM (Fig. EV3C) populations. Contrarily, there was a robust increase in all types of PB circulating CFU-Cs (Fig. EV3G–L), which could be linked with the expansion in hematopoietic progenitors in spleen tissue (Fig. EV3M–R).

## G-CSF induced decrease in SDF-1α expression shifts HSCs away from the capsular myofibroblasts

To examine if this change in splenic hematopoietic activity was associated with a change in spatial localization of pHSC population

following G-CSF treatment, we performed cell positioning analysis of pHSC population relative to the capsule with (Fig. 4H, right panel) and without G-CSF infusion (Fig. 4H, left panel). Upon G-CSF-mediated activation of EMH, we observed a clear increase in the mean distance at which pHSCs (*n* = 6, *N*; control = 707, G-CSF = 957 pHSCs) were located from the splenic capsule (Fig. 4I), indicating their move away from the capsule. Detailed analysis of pHSC distribution at sequential distance intervals showed a robust decrease in the proportion of pHSCs in the first two distance intervals covering a distance of 0–60 µm, shifting the peak farther from the capsule (Fig. 4J). These results showed a clear shift in the pHSC population away from the capsule. We also examined if the spatial metrics were different after G-CSF infusion for the quiescent and proliferative pHSCs, identified based on Ki-67 expression (Fig. 4K,L). In contrast to the observations made for splenic pHSC at homeostatic conditions, we found no change in the mean distance at which the Ki-67⁻ and Ki-67⁺ pHSCs were located with reference to the capsule (Fig. 4K). However, we noted a significant decrease in the proportion of Ki-67⁺ pHSCs within 30 µm distance from the splenic capsule (Fig. 4L). Strikingly, both Ki67⁻ (Fig. 4M,N) as well as Ki67⁺ (Fig. 4O,P) pHSC fractions showed increased mean distance and distribution away from the capsule after G-CSF treatment. As observed for the homeostatic conditions, following G-CSF induced EMH activation, spatial distribution of HSPCs followed the same pattern, and both Ki-67⁻ (Fig. EV3S,T) and Ki-67⁺ (Fig. EV3U,V) HSPCs showed move away from the capsule. Our results had presented differences between male and female mice for pHSC localization in spleen. Albeit the differences in the mean distance at which the pHSCs were localized, a preferential localization closer to the capsular myofibroblasts was noted independent of the sex. Our analysis showed that even after G-CSF treatment, although pHSCs moved away from the capsular lining consistently, mean

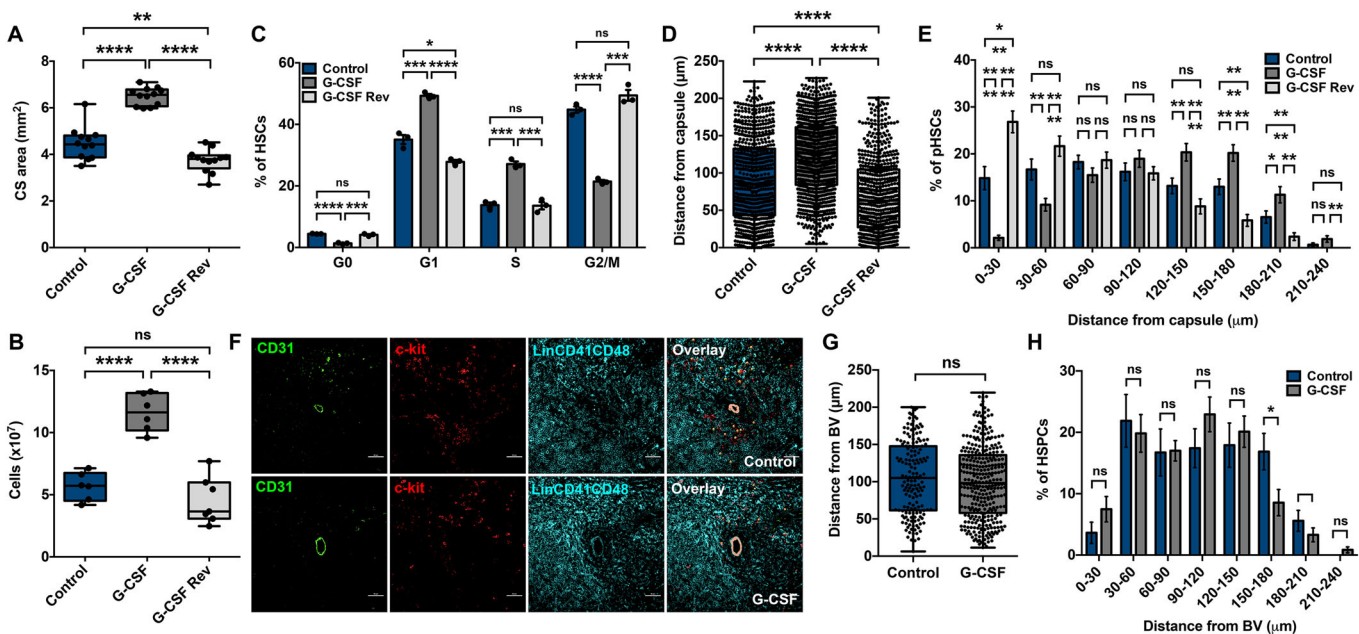

**Figure 5. Reversal of spatial distribution of pHSCs following re-attainment of steady state.**

(A) Comparison of cross-sectional area of the murine spleen tissues harvested from control and G-CSF treated animals. G-CSF treatment was given for 5 days and the mice were sacrificed one day (G-CSF) or 30 days (G-CSF Rev) after the treatment ($n = 6$, $N = 12$ tile scans). (B) Comparison of total spleen cellularity in control and the two groups of G-CSF treated mice. MNCs were harvested from spleen tissues without enzymatic treatment, and viable cell counts were taken using a Neubauer chamber after RBC lysis ($n = 6$). (C) Comparison of the proportion of splenic HSC population in different stages of cell cycle analyzed by flow cytometry. DAPI and Ki-67 based analysis for $G_0$, $G_1$, S, and $G_2$M stages of cell cycle was performed along with immunostaining for HSC markers (Gating strategy is represented in EV3X) ($n = 3$). (D) Euclidean distances calculated for each pHSC with reference to pseudo-surface of capsular myofibroblasts in the spleen from control, G-CSF and G-CSF Rev groups of mice. Each dot represents a pHSC immunolocalized as a $3^-$c-kit$^+$CD150$^+$ cell by confocal imaging ($n = 6$, $N$; control $= 707$, G-CSF $= 957$ and G-CSF Rev $= 629$ pHSCs). (E) Distribution frequency of pHSCs at sequential distance intervals (30 µm each) relative to the splenic capsule in spleen of control, G-CSF treated and mice undergone one month incubation after G-CSF treatment (G-CSF Rev) ($n = 6$, $N$; control $= 707$, G-CSF $= 957$ and G-CSF Rev $= 629$ pHSCs). (F) Confocal imaging to localize HSPCs ($3^-$c-kit$^+$ cells; illuminated yellow) along with vascular endothelial cells (identified by CD31 expression) in spleen sections from control (upper panel) and G-CSF treated mice (lower panel). Scale bars $= 40$ µm. (G) Comparison of Euclidean distances measured for HSPCs with reference to the nearest observable of CD31$^+$ blood vessel in splenic CS of control and G-CSF treated mice ($n = 4$, $N$; control $= 161$, G-CSF $= 326$ HSPCs). (H) Distribution frequency of HSPCs in splenic CS of control and G-CSF treated mice at sequential intervals (30 µm each) from the blood vessels ($n = 4$, $N$; control $= 161$, G-CSF $= 326$ HSPCs). Data is presented as bar graph (mean ± SEM) in panels (C, E, H) or box-whiskers plot (median with min to max) in panels (A, B, D, G). The p-value in figures (A–E, G, H) were calculated by the Student's t-test, *$p < 0.05$, **$p < 0.01$, ***$p < 0.001$, ****$p < 0.0001$, and ns $p > 0.05$.

distance at which HSCs were located was higher in male mice than in the females (Fig. EV3W).

## Rearrangement in pHSC distribution coincides with the recovery from the effects of G-CSF

As the effects of G-CSF has been shown to be transient wherein several effects such as neutrophil production disappear in days to weeks after the treatment (Lapidari et al, 2021). This gave us an opportunity to ask if the HSCs' move away from the capsular myofibroblasts was reversible upon exit from the cell cycle. To examine this, we analyzed the spleen tissues one month post-treatment and noticed a complete reversal of G-CSF effects. The size of the spleen reverted back to steady state level as can be seen by the comparison of the CS area (Fig. 5A) and overall cellularity (Fig. 5B), which was increased upon activation of EMH induced by G-CSF treatment. Next, we performed flow cytometry based cell cycle analysis of the HSC population (CD41⁻CD48⁻LSK cells) based on Ki-67 and DAPI labeling (Fig. EV3X). The results showed that the cells had reverted back to pre-G-CSF treatment phase completely (Fig. 5C). Cumulatively, these results showed that one

month after G-CSF treatment the splenic hematopoietic function reverted back to steady state and was ideal to enquire if this was correlated with their spatial distribution. Therefore, we performed immunostaining based distance analysis of pHSCs with reference to the capsular myofibroblasts. Striking observations were made as we noted that the G-CSF induced shift away from the capsular lining was completely reversed and showed a closer association with myofibroblasts after attaining quiescence (Fig. 5D). Detailed spatial distribution analysis of pHSC also showed a robust increase in the proportion of pHSCs in the distance intervals closer to the capsule (Fig. 5E). Surprisingly, both mean distance analysis and spatial distribution showed an even closer localization of pHSCs to myofibroblasts than before G-CSF treatment. This pointed to the possibility that the re-organization of the hematopoietic niche could be a longer-term phenomenon. Notwithstanding, these results strongly supported the involvement of myofibroblast in supporting HSC function.

Continuing with our examination of any possibility of vasculature in providing microenvironmental support to the splenic HSC populations, we also checked if HSPC population in spleen showed any spatial association with the blood vessels after

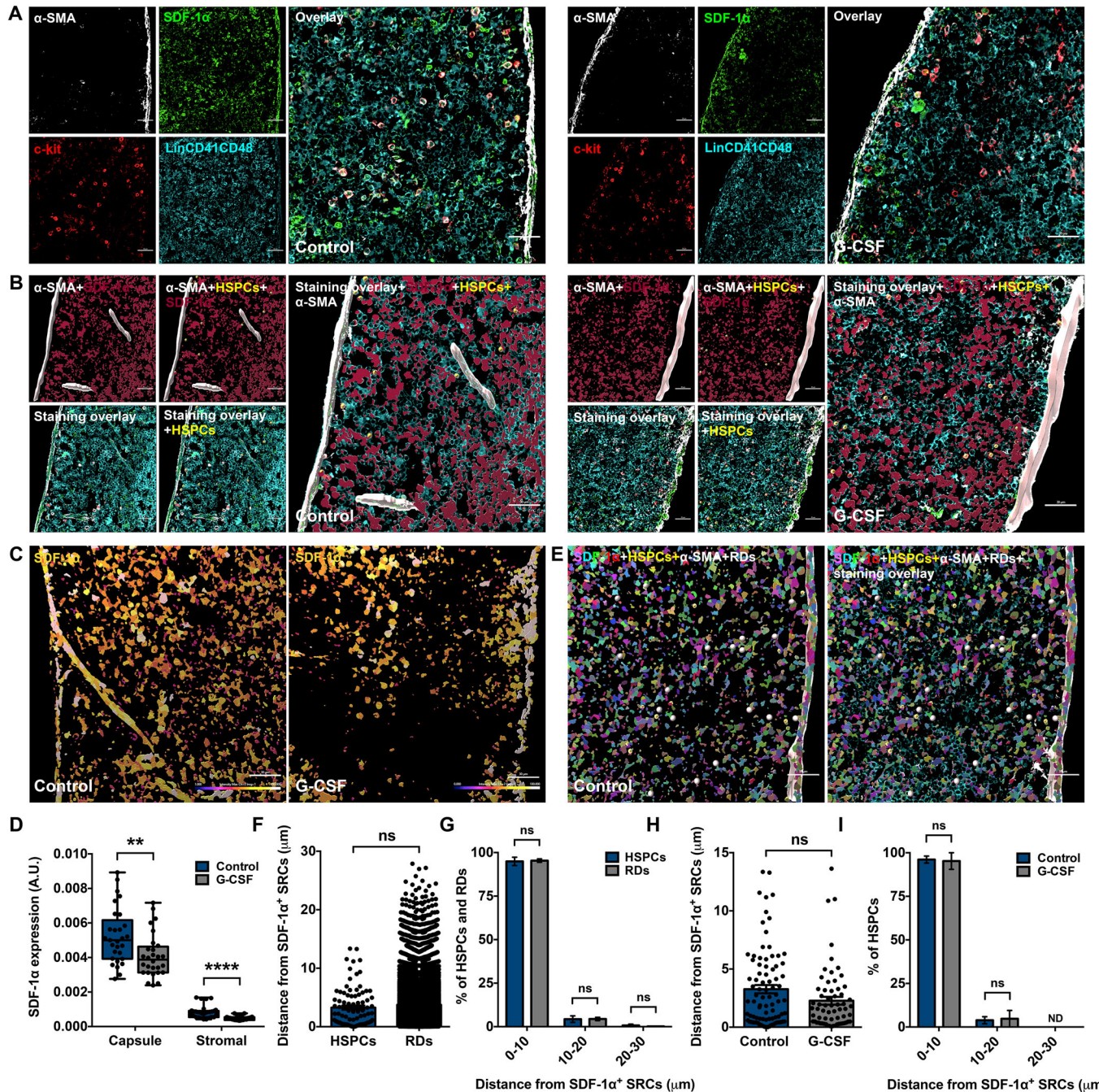

EMH activation by G-CSF treatment. We performed immunostaining based localization of HSPCs along with CD31 immunostaining to identify vascular endothelium (Fig. 5F). Euclidean distances between each HSPC detected and the nearest blood vessel surface generated on splenic CS of control mice were measured and compared with G-CSF treated mice. Our analysis showed that the proximity (Fig. 5G) as well as spatial distribution (Fig. 5H) of HSPCs when referenced to the blood vessels remained unchanged after G-CSF treatment. These results further establish the specificity of myofibroblasts in the creation of a splenic hematopoietic niche. Lack of involvement of vasculature in hematopoietic support makes splenic niche distinct from the BM

that perhaps is crucial for the differences in hematopoietic function in the two sites.

## Splenic hematopoietic cells are preferentially associated with capsular myofibroblasts

The expression of chemokine SDF-1α has been considered a hallmark feature of cells in the niche that supports HSC function (Ding and Morrison, 2013; Roy et al, 2022). Deletion of *Sdf-1α* or its receptor *Cxcr4* led to major hematopoietic defects with significant alteration in the BM and splenic hematopoietic activity (Inra et al, 2015; Nie et al, 2008). Our experiments showed a spatial association between pHSCs

**Figure 6. SDF-1α expressing myofibroblasts but not stromal cells support splenic HSPCs.**

(A) Confocal microscopy performed to locate HSPCs (3⁻c-kit⁺ cells) along with α-SMA and SDF-1α by immunostaining of splenic CS from control (left) and G-CSF treated (right) mice (scale bar = 30 μm). (B) Immunostaining images after pseudo-surface generation for each HSPC detected along with signals for α-SMA and SDF-1α expression. Surface generation was performed using Imaris for immunostained spleen CS from control (left) and G-CSF treated (right) mice (scale bar = 30 μm). (C) Heatmap generated on Imaris for comparative analysis of SDF-1α expression in confocal based immunofluorescence images from control (left) and G-CSF treated (right) mice (scale bar = 30 μm). (D) Immunofluorescence-based quantification of SDF-1α expression in capsular and stromal RP cells of spleen tissue from control and G-CSF treated mice (n = 6, N = 30 images). (E) Pseudo-surface generation on immunolocalized HSPCs with an equal number of RDs (100 iterations) generated on confocal images. Euclidean distances for HSPCs (yellow) and RDs with respect to the nearest observable SDF-1α⁺ cells in the splenic RP area were determined. (F) Comparison of Euclidean distances between HSPCs and RDs from the nearest detected SDF-1α⁺ splenic RP cells (SRCs) (n = 3, N = 87). (G) Distribution frequencies of HSPCs and RDs in splenic tissue at sequential intervals of 10 μm, relative to the SDF-1α⁺ SRCs (n = 3, N = 87). (H) Comparison of Euclidean distances between HSPCs and the nearest detected SDF-1α⁺ SRCs in control and G-CSF treated mice. Each dot represents an HSPC detected based on immunostaining (n = 3, N; Control = 87, G-CSF = 64 pHSCs). (I) Distribution frequency of HSPCs relative to the SDF-1α⁺ SRCs in control and G-CSF treated mice. The proportion of HSPCs at sequential intervals (10 μm each) with reference to the candidate niche cells is presented for the two groups of mice (n = 3, N; Control = 87, G-CSF = 64 pHSCs). Data is presented as bar graph (mean ± SEM) in panels (F–I) or box-whiskers plot (median with min to max) in panel (D). The p-value in figures (D, F–I) were calculated by the Student's t-test, **p < 0.01, ****p < 0.0001, and ns p > 0.05, and ND indicates not detected.

and the capsular myofibroblast population that was found to be the key source of SDF-1α. G-CSF mediated decrease in *Sdf-1α* expression resulted in the pHSCs moving away from the splenic capsule. In addition, we noted a decrease in the expression of *Sdf-1α* in fibrous myofibroblastic cells as well as the lineage-depleted splenic pulp cells. Therefore, in order to further confirm the exclusive nature of capsular hematopoietic niche, we performed spatial mapping for HSPCs relative to non-myofibroblastic SDF-1α⁺ cells in splenic tissue. We examined if G-CSF induced spatial relocation of pHSCs was specifically in reference to capsular myofibroblasts. We performed immunostaining to identify HSPCs along with detection of SDF-1α and α-SMA expression (Fig. 6A). We generated pseudo-surfaces over HSPCs, detected based on the cell surface markers (Lin⁻CD41⁻CD48⁻c-kit⁺ cells), as well as SDF-1α and α-SMA (Fig. 6B). Heatmap analysis of fluorescence intensity for SDF-1α expression confirmed that capsular and trabecular myofibroblasts were the primary source of the chemokine and that the treatment with G-CSF significantly reduced its expression (Fig. 6C,D). Next, we examined the spatial localization of HSPCs with reference to the SDF-1α⁺α-SMA⁻ splenic RP cells (hereafter SRCs). To this end, Euclidean distances between each HSPC detected or RD generated (100 iterations) and the nearest SRC surface were measured (Fig. 6E). In strong corroboration of our hypothesis, we did not detect any preferential HSPC localization in comparison to the RDs (Fig. 6F). It must be noted that as SRCs were spread across the tissue, the overall distance of HSPCs from these cells was shorter than what was observed in reference to α-SMA⁺ cells (Figs. 1I, 6F and EV1E). However, a comparison with RDs showed that HSPCs were randomly located in reference to the SRCs, and their positioning was of no significance (Fig. 6F,G). Similar observations were made when this analysis was repeated for G-CSF treated tissues (Appendix Fig. S2A–C). Importantly, our analysis showed that G-CSF did not alter the micro-location of HSPCs (n = 3, N = 64–87) when referenced to SRCs (Fig. 6H,I), in sharp contrast to significant move away from the splenic capsule (Fig. 4I,J). Overall, these results strongly support a specific role for capsular myofibroblasts in supporting the splenic HSC population.

## Myeloablation induced proliferation in splenic HSCs expands the hematopoietic zone and reorients HSCs

Results from our experiments presented a role of capsular myofibroblastic niche in supporting splenic HSC population. To further substantiate a direct association between the spatial localization of pHSCs with quiescent state, we used 5-Fluorouracil (5-FU) induced myeloablation that results in HSC proliferation (Harrison and Lerner, 1991; Ni et al, 2019). Seven days following the treatment with a sub-lethal dose (200 mg/kg) of 5-FU, we first characterized its effect on overall structural details of the spleen tissue. In contrast to G-CSF treatment, we noted a significant decrease in the weight (Fig. 7A) and cellularity (Fig. EV4A) of spleen following 5-FU treatment. To gain further insights into the change in splenic hematopoietic function induced by 5-FU mediated myeloablation, we performed immunostaining for CD169 to identify marginal zone macrophages that demarcate WP from the RP (Fig. 7B). Consistent with the noted effects of 5-FU, we noted a significant decrease in the CS area of the spleen (Fig. 7B,C). In addition, the decrease was consistent in both RP and WP (Fig. 7D), resulting in an unchanged RP/WP ratio. We confirmed myeloablation induced proliferation by performing flow cytometry based analysis on Ki-67 and DAPI staining in HSC population (Fig. 7E). We noted that the proportion of HSCs in the G₁ stage of cell cycle increased 7 days after 5-FU treatment at the expense of cells in G₀ stage (Fig. 7F). This resulted in significantly increased frequency (Fig. 7G,H) and pool (Fig. EV4B) of both LT-HSC and ST-HSC population. Next, we tested if 5-FU induced proliferation of HSCs also impacted their spatial localization relative to the proposed capsular myofibroblastic niche. We performed immunostaining based spatial analysis to evaluate the Euclidean distance between each pHSC (n = 4, N = 660–707) detected from the nearest capsular myofibroblast (Fig. 7I). Following 5-FU treatment, we noted a substantial increase (from 89.9 ± 2.1 μm to 190.3 ± 3.0 μm) in the mean distances between the pHSC and proposed niche (Fig. 7J). We also noticed that the closer spatial localization of pHSCs to the myofibroblasts in female mice than the male mice remained unaffected, although a robust pHSC movement in both the sexes (Fig. EV4C). As noted in the case of G-CSF treatment, a relocalization of the HSPC population (increase in the mean distance from 92.6 ± 1.9 to 193.4 ± 2.7 μm) similar to the pHSC population was observed (Fig. EV4D). Detailed analysis of the proportion of pHSCs (Fig. 7K) and HSPCs (Fig. EV4E) localized at sequential distance intervals of 100 μm from the capsule showed a substantial shift away from the capsular myofibroblasts. It also struck us that 5-FU treatment also led to a significant expansion of the overall size of the hematopoietic zone, and the pHSCs and HSPCs were detected up to a distance of ≈400 μm from the capsule (Figs. 7J,K and EV4D,E). We noted that

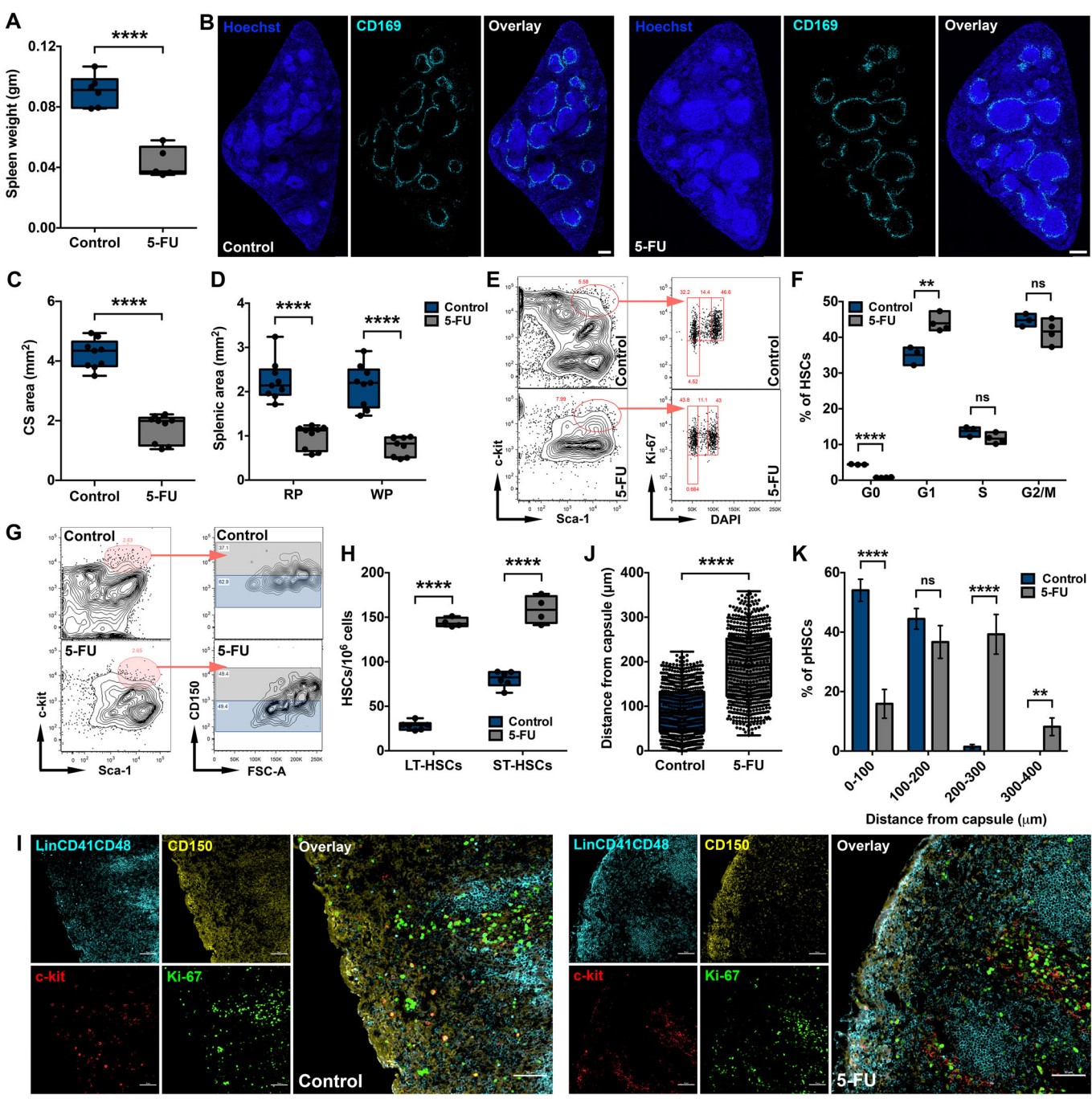

Ki-67+ pHSCs (Fig. EV4F,G), as well as HSPCs (Fig. EV4H,I), were placed moderately but significantly away from the myofibroblasts in comparison to the Ki-67- cells. Intriguingly, the relocation of the stem cell population away from the capsule in response to 5-FU treatment was consistent for Ki-67+ (Fig. EV4J,K) and Ki-67- (Fig. EV4L,M) pHSCs, as well as Ki-67+ (Fig. EV4N,O) and Ki-67- (Fig. EV4P,Q) HSPCs. These results clearly underscored the importance of hematopoietic zone in the capsular myofibroblastic niche in supporting splenic HSC population.

It has been well known that BM failure such as in the case of myeloablation leads to mobilization of BM HSCs that home into the spleen tissue and involve in EMH processes (Short et al, 2019). Our results showed that 5-FU treatment led to reorientation of pHSCs with reference to the myofibroblasts. We sought to test if spleen resident and spleen homed BM derived HSCs localized differently in the splenic hematopoietic niche following 5-FU treatment. In the absence of a specific marker to differentiate between the splenic and BM HSCs, we used transplanted BM HSCs as the proxy for mobilization. CFSE labeled lineage-depleted BM cells were transplanted intravenously one day after 5-FU treatment and spleen tissues were analyzed one day and seven days after transplantation. Analysis of splenic MNCs one day after

Figure 7. Myeloablation induced proliferation relocates pHSCs away from the capsule in an expanded hematopoietic zone.

(A) Change in the spleen weight (in grams) after 7 days of 5-FU treatment ($n = 5$–6). (B) Splenic CS immunostained for CD169 to identify marginal zone macrophages lining the WP areas. The tissues were harvested from the control (left panel) and 5-FU (right panel) treated mice. Nuclei were counterstained with Hoechst 33342 (scale bar = 200 μm). (C) Comparison of cross-sectional areas of the spleen tissues harvested from control and 5-FU treated mice ($n = 3$, $N = 9$ tile scans). (D) Comparison of RP and WP areas in spleen sections following 5-FU treatment. WP areas were detected by the presence of marginal zone macrophages identified by immunostaining for CD169. RP areas were quantified exclusive of WP, vasculature, trabecular, and capsular areas ($n = 3$, $N = 9$). (E) Flow cytometry analysis to compare cell cycle status of HSCs in the spleen with and without 5-FU treatment (same control samples were used for data presented in Fig. EV3X and Appendix Fig. S1H). DAPI and Ki-67 based analysis for $G_0$, $G_1$, S and $G_2M$ stages of cell cycle was performed along with immunostaining for HSC (Lin$^-$CD41$^-$CD48$^-$Sca-1$^+$c-kit$^+$ cells) markers. (F) Comparison of proportion of splenic HSCs from control and 5-FU treated mice in different stages of cell cycle (Gating strategy is represented in (E)) ($n = 3$–4). (G) Flow cytometry analysis to examine LT-HSC (Lin$^-$CD41$^-$CD48$^-$Sca-1$^+$c-kit$^+$CD150$^+$ cells) and ST-HSC (Lin$^-$CD41$^-$CD48$^-$Sca-1$^+$c-kit$^+$CD150$^-$ cells) populations within the MNCs from spleen of control (same data used for Appendix Fig. S1F) and 5-FU treated mice. (H) Comparison of LT-HSC and ST-HSC frequency (cells/million) in spleen tissues treated with and without 5-FU ($n = 4$–6). (I) Confocal images of spleen CS immunostained to localize Ki-67$^+$ and Ki-67$^-$ pHSCs. The spleen tissues were taken from mice treated with (right panel) or without (left panel) 5-FU, and the pHSCs were identified as 3$^-$c-kit$^+$CD150$^+$ cells (scale bar = 50 μm). (J) Euclidean distances calculated for each pHSC with reference to nearest capsular surface in the spleen with or without 5-FU treatment. Each dot represents a pHSC immunolocalized as a 3$^-$c-kit$^+$CD150$^+$ cell by confocal imaging ($n = 4$–6, $N$; Control = 707, 5-FU = 660 pHSCs). (K) Distribution frequency of pHSCs at sequential intervals (30 μm each) relative to capsular myofibroblast in spleen tissues with or without 5-FU treatment ($n = 4$–6, $N$; Control = 707, 5-FU = 660 pHSCs). Data is presented as bar graph (mean ± SEM) in panel (K) or box-whiskers plot (median with min to max) in panels (A, C, D, F, H, J). The $p$-value in figures (A, C, D, F, H–K) were calculated by the Student's $t$-test, **$p < 0.01$, ****$p < 0.0001$, and ns $p > 0.05$.

transplantation detected CFSE$^+$ cells with no contribution from the HSC population (Appendix Fig. S3A). Confocal imaging based immunolocalization of HSPCs also showed the same results as observed by flow cytometry analysis. We detected CFSE$^+$ cells randomly distributed in the spleen tissue and showed no correlation with the myofibroblasts identified by α-SMA expression (Appendix Fig. S3B). Up to 7 days after transplantation, we could not detect any transplanted CFSE$^+$ HSCs by flow cytometry (Appendix Fig. S3C) or by microscopy (Appendix Fig. S3D), although spleen resident CFSE$^-$ cells were clearly seen. These show that the results obtained on the immunolocalization of HSCs following 5-FU treatment were solely based on spleen-resident HSPCs. These findings are in line with the recently published results where BM HSCs detected based on photo-activable fluorescence remained insignificant in the spleen tissues up to a period of 8 days (Johansson et al, 2024).

## Quantitative proteomic profiling reveals molecular support from capsular myofibroblastic niche

Our experiments described a splenic niche that supports pHSCs under homeostatic condition, within which the pHSCs distribute themselves according to the proliferative state. To further confirm the relevance of this niche and to uncover the molecular basis of preferential HSC localization, we performed label-free mass spectrometry (MS) based quantitative proteomic analysis. Niche samples were prepared from FACS sorted Lin$^-$CD45$^-$ splenic cells (hereafter, stromal cells or STCs) and myofibroblasts from the fibrous splenic tissue (hereafter, capsular and trabecular myofibroblasts or CTMs). In addition, FACS sorted LSK cells were used as the hematopoietic progenitors to perform comparative regulatory studies with the candidate niche cell types. The proteins were thermally denatured, reduced, alkylated, and the tryptic digests were desalted before analyzing on nano-LC coupled with high resolution mass spectrometer (nano LC-MS/MS) (n = 6; each sample was analyzed with 3 runs, $N = 18$; Fig. 8A). Using Proteome Discoverer software (version 2.3), a total of 2431 proteins were identified that were used to analyze the global protein expression profile. Principal component analysis (PCA) showed a distinct proteomic profile for each of the three sets of samples that were distinctly clustered (Fig. 8B). Next, the heatmap analysis using the entire datasets was performed to

visualize the abundance of individual proteins in different cell types (Fig. 8C). The proteins that were differentially enriched following comparison between different pairs of cell populations (fold change > 2.0, and adjusted $p$-value < 0.05) were then used to generate Venn diagram (Fig. 8D). To highlight these proteins, we generated volcano plots by plotting the $\log_2$ fold change against the corresponding $-\log_{10}$ adjusted $p$-value (fold change > 2.0 and adjusted $p$-value < 0.05) followed by pairwise comparison for each sample (Figs. 8E and EV5A,B). To evaluate the hematopoietic support potential of the two niche cell types, we performed heatmap analysis on the secretory proteins enriched in CTMs and STCs (Fig. EV5C), which were then visualized on Volcano plots (Fig. 8F). In support of our data that CTMs play a supporting role in the splenic HSC niche, we detected that several known hematopoietic regulators such as CXCL12, POSTN, and Thrombospondin (TSP) are highly enriched in CTMs (Fig. 8F). Overall, CTMs exhibited higher proportion and enrichment of secretory proteins with hematopoietic regulatory function compared to STCs (Fig. 8F), supporting their crucial role in the creation of HSC niche in spleen. Among the 76 secretory proteins that were highly enriched in CTMs, extracellular matrix proteins, such as Collagen, Laminin, Fibronectin, and Fibulin, were highly represented (Fig. EV5C). We then used Reactome database to perform pathway enrichment analysis on the proteins highly enriched in CTMs (Fig. EV5D) and STCs (Fig. EV5E). The results validated the identity of cell types, as the pathways linked to muscle contraction were more represented in CTMs. This analysis also corroborated the data on ECM protein secretion, as pathways involved in ECM protein synthesis were highly enriched in CTMs (Fig. EV5D) than in the STCs (Fig. EV5E). Finally, to establish a preferential regulatory network of the HSC population with myofibroblasts, we performed pathway analysis on the proteins detected in LSK cells. We selected the pathways regulated by the secretory factors that were significantly enriched in CTMs and STCs to predict the regulation of HSPCs by the two candidate niche cell types (Fig. 8G). We found that out of 80 target pathways detected in LSK cells, for an overwhelming proportion (56 in number), the ligand was a secretory protein found highly enriched in the CTMs. In contrast, regulation of only 8 was influenced by the ligands that were enriched in the STCs. For 16 of the detected pathways, the secretory proteins were detected in both of the cell types (Fig. 8G). Interestingly, among the ECM proteins secreted by the niche

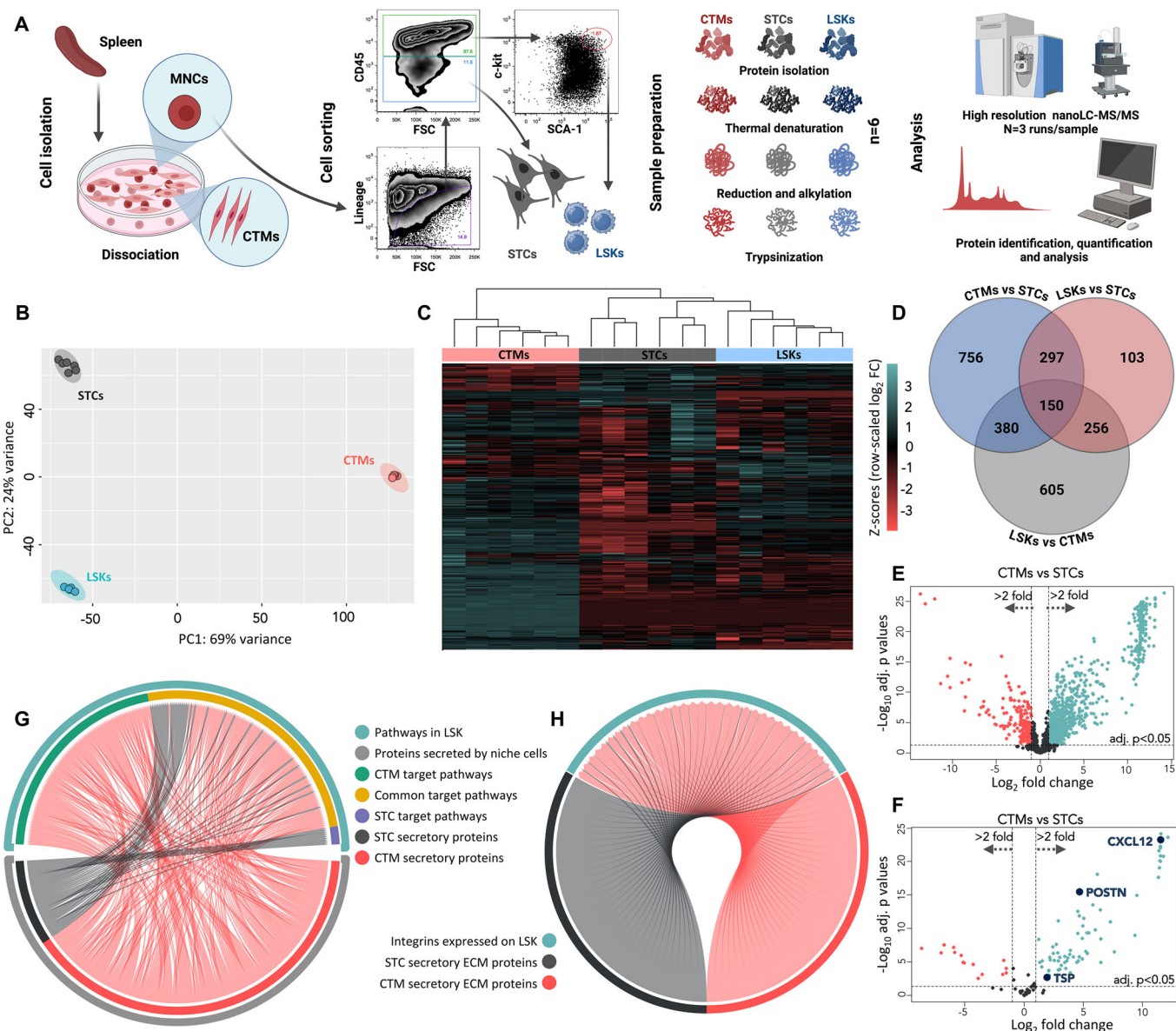

cells (Fig. EV5F) there were several ligands for integrin proteins that were detected in the LSK cells (Fig. EV5G). Therefore, we used CellPhoneDB analysis to construct an HSC-niche cell interaction network using integrins detected in LSK cells and ECM proteins expressed by the two niche cell types (Fig. 8H). While the number of integrin ligands detected in the two niche cell types was not different, the interaction scores represented by cord thickness had a strong bias towards CTM secretome (Fig. 8H). These results further establish the role of myofibroblastic cells in constructing the physical space with required molecular assembly for the maintenance of HSC population.

## Discussion

The size of the HSC pool at a given time is regulated by the pathophysiological state and developmental stage of an organism. Extrinsic signals play a crucial role in maintaining the rate of cell

division as well as the outcome of every proliferative event in HSCs. A significant volume of information has emerged on the molecular and physical factors that regulate BM HSC function in addition to identification of cellular components of their niche (Gao et al, 2018). In this regard, the importance of vascular endothelium (Kunisaki et al, 2013) and associated perivascular cells (Asada et al, 2017) have been well established. Several hematopoietic-intrinsic stresses or altered microenvironment have been shown to result in functional decline in HSCs, resulting in the activation of facultative hematopoietic niches (Yamamoto et al, 2016). Among these extramedullary niches, the spleen has also been shown to host HSCs under homeostatic conditions. Lesser in abundance than in the BM, the splenic HSCs cycle more rapidly and are maintained in $G_1$ stage of cell cycle (Morita et al, 2011). Despite a clear indication of functional equivalence with the BM HSCs, their contribution to homeostatic blood cell production and activation of EMH response, has not been clear. Better HSC marker identification and advanced

◄ **Figure 8.   Interactome analysis based on quantitative proteomics identifies myofibroblasts-biased HSPC associations.**

(A) Schematic showing the experimental design followed for quantitative proteome assessment of the candidate cell types that constitute HSC niche in the spleen. Spleen tissue is ruptured and processed to obtain spleenocytes and fibrous material containing capsular and trabecular myofibroblasts (CTMs). MNCs were used to FACS sort LSK cells and Lin⁻CD45⁻ stromal cells (STCs). The cells were lysed by sonication and the lysates were further processed for protein alkylation with IAA and reduction by DTT treatment before trypsinization and desalting. Trypsinized peptides were analyzed by label-free nano-LC-MS/MS followed by computational analysis (n = 6; each sample was analyzed with 3 runs, N = 18). (B) Principal component analysis was performed on untargeted proteomic profiles of CTM, STC, and LSK cell samples from the spleen tissues of young adult mice. (C) Hierarchical clustering of differentially expressed proteins detected and quantified by mass spectrometry. (D) Venn diagram analysis of the proteomes detected from the CTM, STC, and LSK cells. The number of proteins differentially expressed (adjusted p-value < 0.05 and fold change > 2) between each pair (commonly or exclusively) of cell populations (CTMs versus STCs, LSKs versus STCs, and LSKs versus CTMs) is shown. (E) Volcano plots showing proteins differentially expressed between CTM versus STC populations (n = 6; each sample was analyzed with 3 runs, N = 18). Fold change (Log₂ values) and adjusted p-values (−log₁₀ values) are plotted on the x and y axes, respectively. Cyan colored dots represent proteins with statistically significant (adjusted p-value < 0.05) increase in abundance in the CTMs with fold change > 2.0. Red colored dots represent proteins having statistically significant (adjusted p-value < 0.05) decrease in abundance in CTMs with fold change > 2.0. (F) Volcano plots showing differentially enriched secretory proteins compared between CTM versus ST populations (n = 6; each sample was analyzed with 3 runs, N = 18). (G) Chord diagram illustrating the pathways detected in the LSK cells (upper half; cyan semicircle) that are regulated by (adjusted p-value < 0.05 and fold change > 2) the secretory proteins (lower half; gray semicircle) enriched in the STCs (black line of inner circle) and CTMs (red line of inner circle). The inner circle of the top section represents cellular processes influenced exclusively (CTMs in green and STMs in violet) or commonly (yellow) by the two niche cell types. Each top-bottom chord connection represents a predicted regulation of HSPC by niche cells. (H) CellPhoneDB based chord diagram illustrating the ligand-receptor communications between the ECM proteins expressed by niche cell types (STC proteins in black, CTM proteins in pink) and potential integrin dimers of monomers detected in LSKs (in cyan). Each chord represents a predicted interaction between an ECM protein with the cell surface expressed integrin protein. The thickness of the chord in the upper section represents the interaction score between the ligand-receptor pair. Represented data are based on six independent biological replicates (n = 6), each with three technical replicates (N = 18).

imaging tools have supported the efforts to better understand the extrinsic regulation of splenic hematopoiesis. We have identified a capsular myofibroblastic niche for splenic pHSCs based on a stringent immunolabelling based identification. The pHSCs identified as Lin⁻CD41⁻CD48⁻c-kit⁺CD150⁺ cells, located themselves preferentially closer to the splenic capsule composed of myofibroblasts. This was consistent for splenic tissues harvested from the male as well as female mice. However, spatial proximity of pHSCs to the myofibroblasts was more pronounced in females, indicative of subtle microenvironmental changes in support of differential physiological demands. Sex-specific differences in the functioning of hematopoietic system have been reported (Chaudhary et al, 2024). Our study present the first evidence of microenvironmental differences between males and females. These results can be further explored to understand the functioning of hematopoietic system in the two sexes. It is also important to note that the alteration in the micro-location of pHSCs in response to G-CSF and 5-FU treatment was consistent in both sexes.

These myofibroblasts expressed most of the genes that code for known hematopoietic regulators, with the exception of *Tpo*. Importantly, we noted that myofibroblasts expressed significantly higher levels of SDF-1α/CXCL12 than the non-hematopoietic stromal cells in splenic RP. Our mass-spectrometry based global comparative proteomic profile supported these results as SDF-1α, along with other known regulators of hematopoietic function, including POSTN and TSP, were also highly enriched in myofibroblasts. Analysis of specific cell types for the expression of SDF-1α had shown its highest expression in PDGFR-β⁺ RP stromal cells (Inra et al, 2015). Another study showed that Tlx1⁺ mesenchymal cells that also expressed PDGRFR-β showed significant enrichment of *Sdf-1α* expression (Oda et al, 2018). Although not explicitly stated, the images presented in this study did show significant expression of Tlx1 in the splenic capsule. Unlike in the case of BM (Ding and Morrison, 2013; Sugiyama et al, 2006), our experiments did not show a significant association of SDF-1α expression with endothelium. While our results confirmed that the expression of SDF-1α was widespread in the spleen RP tissue, the preferential localization of pHSCs under homeostatic conditions was restricted to SDF-1α expressing capsular

myofibroblasts. Decreased SDF-1α expression following G-CSF treatment was associated with a shift of pHSC away from the splenic periphery, albeit within the capsular zone.

Change in SDF-1α expression in different cell types has been shown to affect hematopoietic processes distinctly. The HSC population in BM was significantly depleted when *Sdf-1α* was deleted in endothelial cells, but not upon *Vav1-Cre* or *Nestin-Cre* mediated deletion in hematopoietic or mesenchymal cells (Ding and Morrison, 2013). In fact, *Lepr-Cre* mediated deletion of *Sdf-1α* resulted in mobilization of BM HSCs. It is interesting to note that G-CSF resulted in reduced expression of SDF-1α both in myofibroblasts and splenic stroma. However, we could observe pHSCs move only with reference to the splenic capsule, further pointing to the importance of myofibroblastic niche in capsular zone. Earlier reports have emphasized the importance of splenic stromal cells in supporting hematopoietic activities. *Tcf21⁺* stromal cells in the spleen expressed hematopoietic regulators SCF and SDF-1α; their conditional deletion severely impacted splenic EMH without affecting BM hematopoiesis (Inra et al, 2015). Notably, the expression of *Tcf21* has been linked with the origin of *Postn*-expressing myofibroblasts in heart tissue (Kanisicak et al, 2016). Therefore, a change in the expression of SDF-1α in the myofibroblast population following its genetic deletion in *Tcf21⁺* cells cannot be ruled out. Our study took strength from the spatial distribution of pHSCs during steady state and presented the differences resulting from stress hematopoiesis.

We noted a strong preference of Ki-67⁻ quiescent pHSCs to the capsular myofibroblasts in comparison to the Ki-67⁺ proliferative cells. It is important to note that all detected pHSCs, irrespective of their proliferation state, were found to be in the ≈200 μm wide zone from the splenic capsule. Interestingly, 5-FU induced myelosuppression that led to activation of quiescent HSCs resulted in a robust expansion of the hematopoietic zone, corroborating the importance of capsular niche. Notably, the expansion of the splenic niche was based on the relocalization of spleen resident HSCs and not the incoming BM derived HSCs. In support of a recent study using Kaede mice (Johansson et al, 2024), our results based on

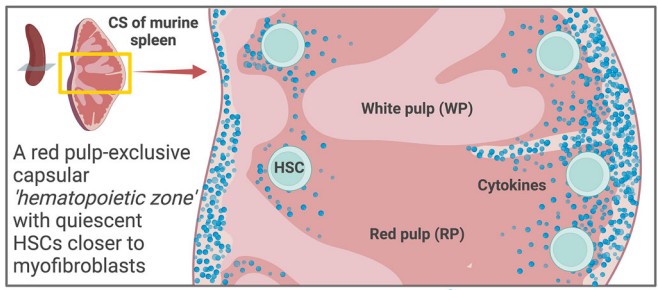

**Figure 9.    A new model of HSC niche in spleen under homeostatic conditions.**

A narrow hematopoietic zone close to capsular myofibroblasts hosts HSCs. The distribution of HSCs within this capsular niche is determined by their proliferative state. Within the hematopoietic zone, the proliferative HSCs get located away from the myofibroblasts in an inducible manner.

transplanted HSPCs as proxy for mobilized BM derived hematopoietic cells showed that the contribution of BM HSPCs in splenic hematopoietic activity was negligible up to 8 days after 5-FU treatment. Expansion of hematopoietic zone following 5-FU was in sharp contrast to the results obtained from G-CSF treatment that led to the pHSCs moving away from the capsule without the expansion of hematopoietic zone. We attributed these differences to changes in splenic architecture due to massive loss of proliferative cells following 5-FU treatment. It is evident that the processes leading up to and following EMH induction vary based on the model under consideration.

The heterogeneity of the HSC population in terms of proliferation state has been used to verify the importance of niche components. BM resident quiescent HSCs were detected based on Edu labeling and Ki-67 immunostaining occupied arteriolar niches (Kunisaki et al, 2013). As compared with the proliferative HSCs, they were found closer to the arteriolar Nestin-GFP^bright cells. Contrarily, no such differences were noted when their association was tested with Col1-GFP cells or sinusoidal vessels (Acar et al, 2015), indicating the importance of arteriolar niches within a broad vascular microenvironment. Unlike in the BM, splenic HSPCs did not show any spatial proximity towards blood or lymphatic vessels. In addition, the HSPCs were seen to be randomly located in reference to the vascular structures independent of the proliferation state. Even G-CSF induced changes in the molecular profile of splenic niche did not alter their location in terms of vasculature. These findings show that vasculature does not play any significant role in the creation of hematopoietic niche spleen and highlight the exclusivity of myofibroblasts as the niche components. Reversal of alteration in the spatial distribution of pHSCs upon alleviation of effects of G-CSF on splenic hematopoietic function also supports the involvement of myofibroblasts in the creation of a steady state splenic niche.

Our proteomics data provided support to the importance of the capsular myofibroblastic niche. Splenic fibrous tissue containing myofibroblasts expressed a vast proportion of ECM components and secretory proteins, several of which have known roles in hematopoietic function. Notably, a comparative interactome analysis with reference to the proteins enriched in splenic LSK population showed a major bias towards capsular myofibroblastic niche. Hence, these experiments have established spatial

correlation with capsular myofibroblasts and show molecular associations between the two niche cell types (Fig. 9). Overall, this study provides the first model of hematopoietic niche in spleen at steady state which can be employed to reveal novel hematopoietic regulators to activate HSCs while supporting their function. A more homogeneous splenic niche structure than in the BM can also provide an interesting model to understand physical attributes of the microenvironment crucial for maintaining HSCs. This cumulative knowledge could be used in preventing the loss of HSCs due to spleen homing in BM donors. More studies are warranted to explore spleen as a hematopoietic site that has remained underexplored and has the potential to act as a source of HSCs apart from presenting a model to study hematopoietic processes.

## Methods

**Reagents and tools table**

| Reagent/Resource | Reference or Source | Identifier or Catalog Number |
|---|---|---|
| **Experimental models** | | |
| Mouse | Jax Mice-NCBS | C57BL/6J-NCBS |
| **Antibodies** | | |
| Antibody list | – | Appendix Table S2 |
| **Oligonucleotides and other sequence-based reagents** | | |
| List of primers | Merck and Europhins | Appendix Table S1 |
| **Chemicals, Enzymes, and other reagents** | | |
| G-CSF | INTAS Pharmaceuticals | Neukine 300 |
| 5-Flurouracil | Sigma | F6637 |
| CellTrace CFSE | Invitrogen | C34554 |
| TB Green Premix | Takara | RR820 |
| PrimeSceipt RT Enzyme | Takara | RR037A-2 |
| SOLu-Trypsin | Merck | EMS0004 |
| **Software** | | |
| IMARIS | Imaris x64 10.0.0 | – |
| Leica application suite | LAS AF | – |
| ZEN | 2.3SP1 | – |
| FlowJo | V9.9.4 | – |
| GraphPad Prism | 6.0e | – |
| Rstudio | 2022.12.0.352 | – |
| Proteome Discoverer | Version 2.3 | – |
| **Other** | | |
| EasySep HSPC isolation kit | STEMCELL Technologies | 19856A |
| Pierce C-18 Spin Columns | Thermo Scientific | 89870 |
| Acclaim PepMap nanoViper Trap coloumn | Thermo Scientific | 164946 |
| Acclaim PepMap nanoViper analytical coloumn | Thermo Scientific | 164940 |

## Animals

Eight- to twelve-week-old C57BL/6J-NCBS mice were bred and maintained in the animal facility at IISER Thiruvananthapuram. During the experiments, mice were maintained in isolator cages at a humidified constant temperature (22 °C), with a 12 h light-dark cycle. The mice were fed with autoclaved water, and irradiated food, ad libitum. All experiments were conducted following the guidelines provided by the Committee for the Purpose of Control and Supervision of Experiments on Animals (CPCSEA), Ministry of Environment and Forests, Government of India. All animal experiments were approved by the Institutional Animal Ethics Committee.

## In vivo treatments and tissue harvesting

Granulocyte-colony stimulating factor (G-CSF) was diluted in PBS (pH 7.4), and a daily dose (250 µg/kg) was administered for 5 days. The mice were sacrificed one day or 30 days after the final dose was injected. 5-Fluorouracil (5-FU) was diluted in PBS (pH 7.4) and administered intraperitoneally at a dose of 200 mg/kg. Control mice were injected with an equal volume of vehicle (PBS). Injected mice were sacrificed 7 days after the treatment. BM Lin⁻ cells (10 million) were labeled with 25 µM CFSE following the manufacturer's protocol (Cell TraceTM CFSE Cell Proliferation Kit; Invitrogen, Waltham, MA). The cell were transplanted intravenously via lateral tail vein 24 h after 5-FU treatment and the mice were sacrificed 1 or 7 days post-transplantation. Spleen tissues from adult mice were harvested, washed with PBS, and fixed with 4% paraformaldehyde (PFA) for 2 h at 4 °C. Fixed tissues were subjected to cryoprotection with a 30% sucrose gradient for 48 h prior to cryo-block preparation using PolyFreeze (P0091; Sigma).

## RNA isolation and quantitative RT-PCR

Spleen tissues were collected and dissociated to isolate spleen mononuclear cells (MNCs) and fibrous tissues containing myofibroblasts. EasySep™ Mouse Hematopoietic Progenitor Cell Isolation Kit was used to isolate lineage negative cells from MNCs of splenic pulp. Total RNA was isolated from Lineage-depleted cells and fibrous tissue using TRIzol™ reagent and subjected to cDNA synthesis using PrimeScript RT reagent kit (Takara). Real-time quantitative PCR was performed using TB Green Premix (Takara) in a CFX96 Real-Time PCR system (Biorad) following the standard protocols using primers listed in Appendix Table S1. Relative gene expression was calculated by comparative $C_t$ method and normalized to *Gapdh* expression.

## Immunostaining and imaging

Spleen tissues fixed in PFA were used to cut cryosections of 10 µm thickness using a cryostat (HM525 NX; Thermo Fisher Scientific). The sections were transferred on frosted slides and immunostained using specific antibodies against α-smooth muscle actin (α-SMA; Abcam), CD31 (Abcam), Lyve-1 (Ebio), CD169 (Biolegend), CD150 (Abcam), Ki-67 (biotin conjugated; Ebio), c-kit (R&D), CD41 (R&D) and CD48 (R&D) along with anti-lineage antibody cocktail (anti-Ter119, anti-B220, anti-CD3e, anti-Gr-1, anti-CD11b and anti-F4/80, all from R&D Systems) and fluorescently labeled secondary antibodies (Jackson ImmunoResearch Laboratories Inc.).

A List of all antibodies used is provided in Appendix Table S2. Stained spleen sections were mounted using ProLong Gold antifade mounting medium (Invitrogen_P36934). Fluorescence imaging was performed using a Leica TCS SP5 II upright and Carl ZEISS-LSM880 upright confocal microscope. Images were captured using oil immersion objectives (Leica: HCX PL APO CS 63.0x/1.40 Oil, HCX PL APO 40x/1.30 Oil and Zeiss: 63x/1.4oilDIC_420782-9900) with software LAS AF and ZEN 2.3SP1 (Leica and Zeiss, respectively).

## Image processing and analysis

The 2D images generated by confocal imaging were converted into a .ims file format and analyzed using Imaris (Imaris x64 10.0.0). pHSCs and HSPCs were identified as 3⁻ (Lin⁻CD41⁻CD48⁻) c-kit⁺CD150⁺ and 3⁻c-kit⁺ cells, respectively. Co-immunostaining for Ki-67 was performed to distinguish proliferative and quiescent pHSCs and HSPCs. The pHSCs and HSPCs were marked on software and generated pseudo-cell surfaces, which were further separately identified as Ki-67⁺ and Ki-67⁻ cells. The Euclidean distances between pHSCs/HSPCs and candidate niche cells were measured using pseudo-surfaces generated for each cell type; the distance data was exported in a .xls file format and plotted. The 2D images with pseudo-surfaces created were also used to generate random dots (RDs) with dimensions proportionate to the average size of pHSCs detected (diameter ≈ 6 µm). RD coordinates were generated in numbers equivalent to the pHSCs or HSPCs detected and used in the analysis. The coordinates were then imported to Imaris through MATLAB XTension to generate the RDs. Euclidean distances between the pseudo-surfaces of pHSC/HSPC and the niche cells were then calculated and compared with RDs. A hundred such iterations were performed on each image to cover the imaged spleen tissue to maximize the randomization process.

## Flow cytometry and cell sorting

Flow cytometry was used to perform an immunostaining based analysis of spleen resident HSCs, HSPCs, and lineage-committed hematopoietic cell populations. HSC and HSPC populations were analyzed by immunostaining RBC depleted spleen MNCs using PE conjugated anti-mouse c-kit, BB700 conjugated anti-mouse Sca-1, PE-Cy7 conjugated anti-mouse CD150, FITC conjugated anti-mouse CD41, FITC conjugated anti-mouse CD48 and APC conjugated anti-mouse lineage antibody cocktail (BD Pharmingen). In the transplantation experiments, CFSE label was used to identify transplanted hematopoietic cells. LT-HSC and ST-HSC populations were identified as Lin⁻CD41⁻CD48⁻Sca-1⁺c-kit⁺CD150⁺ and Lin⁻CD41⁻CD48⁻Sca-1⁺c-kit⁺CD150⁻ cells, respectively.

For cell cycle analysis, DAPI and Ki-67 immunostaining based analysis was performed on the Lin⁻CD41⁻CD48⁻Sca-1⁺c-kit⁺ cells. Briefly, the cells were immunostained for c-kit, Sca-1, CD41, CD48, and Lineage markers, followed by permeabilization and fixation using BD Cytofix/Cytoperm buffer. The cells were then washed with Perm/Wash buffer and immunostained with APC conjugated anti-mouse Ki-67 antibody and labeled with DAPI for 15 min at room temperature. A list of all antibodies used is provided in Appendix Table S2. Samples were acquired on FACS ARIA III (BD Biosciences, San Jose, CA) and analzed using FlowJo_v9.9.4 software (TreeStar, Ashland, OR).

## Methylcellulose based hematopoietic colony-forming assay

Change in hematopoietic function after G-CSF treatment was confirmed by performing in vitro methylcellulose-based colony-forming cell assays. MNCs from BM, peripheral blood (PB), and RBC depleted MNCs from the spleen were harvested from control or G-CSF treated animals. MNCs from PB were isolated by Histopaque density gradient centrifugation method from 200 µl blood. The MNCs ($1 \times 10^5$ cells/ml for BM, spleen, and all MNCs from 200 µl PB) were then plated in duplicate wells (1 ml each) in Methocult (STEMCELL Technologies™). The cultures were maintained at 37 °C, and colonies were scored after 14 days.

## Sample preparation for proteomic analysis

Proteomic analysis was performed on Lin⁻Sca-1⁺c-kit⁺ cells (LSKs), capsular and trabecular myofibroblasts (CTMs) from the fibrous spleen tissue and Lin⁻CD45⁻ spleen cells (stromal cells or STCs). Spleen tissues from young adult mice were dissociated in PBS, followed by the separation of MNCs that were used to FACS sort LSK cells and STCs; CTMs were obtained from the fibrous tissue. The cells (≈50,000) were suspended in 100 µl of lysis buffer [0.01 M tris base (pH 7.4), 0.14 M NaCl, 1% Triton X-100, and 1 mM phenylmethylsulfonyl fluoride (PMSF)] prepared in LiChrosolv water for chromatography and sonicated for 15 min. Ice-cold acetone (900 µl) was added to the cell lysate, and after incubation for 20 min at −20 °C, the lysates were centrifuged at $17,400 \times g$ for 10 min. The protein pellets were suspended in 100 µl of 50 mM ammonium bicarbonate and thermally denatured at 65 °C for 10 min. Protein samples were then reduced by adding 2.5 µl of 200 mM dithiothreitol (DTT) and incubating at 60 °C for 45 min. This was followed by alkylation with 10 µl of 200 mM iodoacetamide (IAA), followed by a second reduction step with the addition of 2.5 µl of an aliquot of 200 mM DTT. Protein samples were then digested overnight with 4 µg of SOLu-Trypsin at 37 °C. The samples were speed-vacuum-dried and resuspended in 100 µl of ammonium bicarbonate before desalting using Pierce™ C-18 spin columns.

## Nano LC-MS/MS analysis

All protein samples were resuspended in 0.1% formic acid, and 8 µl of each sample (0.75 mg/ml) in triplicates was analyzed on a Thermo Fisher Scientific EASY-nLC 1000 system directly connected to a Thermo Fisher Scientific The Q Exactive - A Benchtop Orbitrap Mass Spectrometer. Acclaim PepMap 100 C18 trap column (3 µm, 75 µm × 20 mm; nanoViper Trap, 1200 bar) and Acclaim PepMap 100 C18 HPLC analytical column (2 µm, 75 µm × 150 mm; nanoViper FS, 1200 bar) were used for the separation of proteins with the parameters and settings employed earlier (Pv et al, 2024). The mobile phases A and B consisted of 0.1% formic acid in LiChrosolv water for chromatography and LiChrosolv Reag. Ph Eur. Acetonitrile, respectively. The applied 140-min multistep gradient ranged from 5 to 95% mobile phase B at a constant flow rate of 300 nl/min. The eluent was directly introduced into the mass spectrometer via Thermo Fisher Scientific Nanospray Flex ion source connected with the Thermo Fisher Scientific Nano Bore Emitter Stainless Steel of 40 mm with an outer diameter of 1/32 inch. The applied spray voltage was 18,000.00 (+)

and 2000.00 (−), and the capillary temperature was set to 250 °C. We measured in positive ion mode. Full-scan MS spectra were recorded from 350 to 1500 mass/charge ratio at 70,000 resolution, followed by MS/MS spectra recorded from 200 to 2000 mass/charge ratio at resolution of 17,500.

## Computational analysis

Protein identification and quantification were performed using Thermo Fisher Scientific Proteome Discoverer Software (version 2.3). The parameters for protein identification were employed in the Sequest ST node of the processing workflow within the Proteome Discoverer software. The maximum number of missed cleavage sites was set to 2, and the detected peptides with lengths ranging from a minimum of 6 to a maximum of 144 amino acids were considered for the analysis. Dynamic modification was specified for methionine oxidation (+15.995 Da) with a maximum of three modifications per peptide. While no dynamic modifications were applied for the peptide terminus, the protein N-terminus modification was specified for acetylation (+42.011 Da). Static modification was specified for cysteine carbamidomethylation (+57.021 Da). Decoy database q-values were used to calculate the false discovery rate (FDR) in the Percolator node. At the peptide spectral match level, the data was filtered using a strict FDR threshold of 0.01 and relaxed FDR threshold of 0.05. For proteomic analysis, averages of relative abundances calculated from all technical replicates were used for imputation of non-detected samples. For the peptides that remained undetected across replicates, the lowest abundance value detected was used. Differential protein expression analysis was carried out using Rstudio (2022.12.0.352) and proteins with fold change > 2.0 and adjusted $p$-value < 0.05 ($p$-values were adjusted by Benjamini and Hochberg's method for multiple testing) were considered as differentially expressed. Stastical analysis was carried out using Limma package (version 3.50.3) in R/Bioconductor. To establish a secretome profile of the niche cells, differentially expressed secretory factors (fold change > 2.0, adjusted $p$-value < 0.05) were selected from the list of detected proteins based on the publicly available database UniProt. Pathways were considered based on the involvemet of secretory protein in the listed pathways, as annotated in the UniProt2Reactome.txt file from reactome database by comparing proteomes of each cell type by Rstudio. Proteins upregulated with fold change > 2.0 and adjusted $p$-value < 0.05 were taken into account for pathway analysis. The pathways that were up- or down-regulated with adjusted $p$-value < 0.05 were considered as differentially regulated. Pathways detected in the LSK cells regulated by significantly upregulated secretory proteins from the two niche cell types were listed. Generated data was then used to create a matrix and chord diagram was generated using the circlize package in Rstudio (Gu et al, 2014).

Cell-cell interactome was performed on secretory ECM proteins enriched in niche cell types and their known interacting partners (integrins) expressed on LSKs using CellPhoneDB (Efremova et al, 2020). Secretory proteins from CTMS and STCs were identified as ligand-expressing "sender" cells, while integrins expressing LSKs were designated as the receptor-expressing "receiver" cells. Interaction pairs with $p < 0.05$ were determined by CellPhoneDB and were selected for further analysis. These interaction pairs used to

create a matrix and chord diagram was generated by the circlize package in Rstudio (Gu et al, 2014).

## Quantification and statistical analysis

All data are represented as bar graphs or box and whiskers plots. In the bar graphs, data are presented as mean values, with error bars representing ±SEM (Standard error of the mean). In the Box-and-Whisker plots, data are presented median with min to max; the boxes represent the interquartile range (IQR) with the upper and lower quartiles, while the whiskers indicate the range from the upper to the lower quartiles. Comparisons between samples from two groups with normally distributed data with equal variance were made using the unpaired two-tailed Student's $t$-test. Statistical analyses were performed with Microsoft Excel or GraphPad Prism 6.0e. For all analyses, $p$-values $\leq 0.05$ were accepted as statistically significant. Asterisks in the figures indicate statistical significance $*p < 0.05$, $**p < 0.01$, $***p < 0.001$, $****p < 0.0001$, ns $p > 0.05$.

## Data availability

All data are available in the manuscript or the supplementary materials. The proteomics dataset generated has been deposited to the MassIVE data repository and ProteomeXchange Consortium under accession numbers MSV000095385 and PXD054056, respectively. The source data used for the study is deposited on BioImage Archive via BioStudies submission tool with accession ID S-BSST1913 or https://doi.org/10.6019/S-BSST1913. The code for the bioinformatics analysis and analyzed data is available at https://github.com/stemcellbiologylab/Spleen-project.git.

The source data of this paper are collected in the following database record: biostudies:S-SCDT-10_1038-S44318-025-00477-2.

## Peer review information

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

## Acknowledgements

This work was supported by the DBT/Wellcome Trust India Alliance Fellowship (IA/I/15/2/502061), Core Research Grant (CRG) from Science and

Engineering Research Board, Department of Science and Technology (CRG/2022/000834) and intramural funds from Indian Institute of Science Education and Research Thiruvananthapuram (IISER TVM) awarded to SK. IISER TVM Institutional animal facility is supported by funds from the Department of Science and Technology, Government of India (under FIST scheme; SR/FST/LS-II/2018/217). SHM and HA received support from IISER TVM. SS and AVH received INSPIRE fellowship from Department of Science and Technology, Government of India. ASK was supported by E-Grantz scholarship from Government of Kerala. MTA is supported by the UAE University internal research grants (startup grant code G00003688 and UPAR grant code G00004152). We acknowledge the support from the mass-spectrometry unit of the Central Instrumentation Facility (CIF) at IISER Thiruvananthapuram for the proteomic analysis. The authors would like to thank Mr. Pradeep Kumar G and Mrs. Ansumol Cherian for their technical support.

## Author contributions

**Shubham Haribhau Mehatre**: Data curation; Software; Formal analysis; Investigation; Visualization; Methodology; Writing—original draft; Writing—review and editing. **Sreelakshmi Sanam**: Data curation; Software; Formal analysis; Visualization; Methodology; Writing—review and editing. **Harsh Agrawal**: Software; Formal analysis; Investigation; Visualization; Methodology; Writing—review and editing. **Amulya V Hejjaji**: Data curation; Formal analysis; Investigation; Visualization; Writing—review and editing. **Akhila S Kumar**: Data curation; Formal analysis; Visualization; Methodology; Writing—review and editing. **Mohammad Tauqeer Alam**: Formal analysis; Supervision; Validation; Visualization; Methodology; Writing—review and editing. **Satish Khurana**: Conceptualization; Resources; Supervision; Funding acquisition; Validation; Investigation; Writing—original draft; Project administration; Writing—review and editing.

Source data underlying figure panels in this paper may have individual authorship assigned. Where available, figure panel/source data authorship is listed in the following database record: biostudies:S-SCDT-10_1038-S44318-025-00477-2.

## Disclosure and competing interests statement

The authors declare no competing interests.

# Expanded View Figures

**Figure EV1.  Capsular myofibroblasts constitute to the hematopoietic niche in adult spleen.**

(**A**) Flow cytometry was performed to evaluate different marker combinations for identifying primitive HSCs (pHSCs or $CD150^+CD41^-CD48^-$LSK cells) using three fluorophores. The $Lin^-CD41^-CD48^-$ (or $3^-$) cells (**Ai**) were further gated on $CD150^+$c-kit$^+$ (**Aii**), $CD150^+$Sca-1$^+$ (**Aiii**), and Sca-1$^+$c-kit$^+$ (**Aiv**) cells; and the proportion of pHSCs in each one of them was examined ($n = 7$). (**B**) Representative confocal images showing immunofluorescence-based localization of $3^-$c-kit$^+$ HSPCs, along with α-SMA$^+$ capsular myofibroblasts. Pseudo surfaces for HSPCs (illuminated yellow) and capsular myofibroblast (illuminated white) were generated using Imaris. The Euclidean distance from the surfaces of HSPCs with respect to the nearest observable capsular myofibroblast in the spleen was determined. Scale bars = 20 μm (left panel), 3 μm (right panel). (**C**) Spatial distribution frequency of HSPCs in sequential intervals of 100 μm relative to capsular myofibroblasts ($n = 4$; $N = 20$ images). (**D**) Confocal based immunofluorescence imaging to locate $3^-$c-kit$^+$ HSPCs along with α-SMA$^+$ capsular myofibroblasts. Pseudo surfaces for HSPCs (illuminated yellow) and capsular myofibroblast (illuminated white) were generated using Imaris. An equivalent number of RDs, as that of HSPCs identified, were generated, and 100 iterations were performed for analysis. The Euclidean distances from the HSPC cell surfaces and RDs, with respect to the nearest observable capsular myofibroblast in the spleen were determined (scale bar = 20 μm). (**E**) Comparison of Euclidean distances measured for HSPCs and RDs with reference to the nearest observable of α-SMA$^+$ capsular myofibroblast in splenic sections ($n = 3$, $N = 818$ HSPCs; each dot represents an HSPC or an RD). (**F**) Distribution of HSPCs and RDs at sequential intervals (30 μm each) from the splenic capsule identified by α-SMA immunostaining. The HSPCs were identified as $3^-$c-kit$^+$ cells in the splenic tissue, and the surfaces and RDs were generated using Imaris ($n = 3$, $N = 818$ HSPCs). (**G**) The proportion of Ki-67$^-$ (quiescent) and Ki-67$^+$ (proliferative) cells with pHSC and HSPC populations identified by confocal based imaging of immunostained spleen sections ($n = 4$, $N$; pHSCs = 557, HSPCs = 670). Data is presented as bar graph (mean ± SEM) in panel (**F**) or box-whiskers plot (median with min to max) in panels (**C**, **E**) or stacked bars in (**G**). The *p*-value in figures (**C–F**) were calculated by the Student's *t*-test, *$p < 0.05$, **$p < 0.01$, ****$p < 0.0001$, and ns $p > 0.05$.

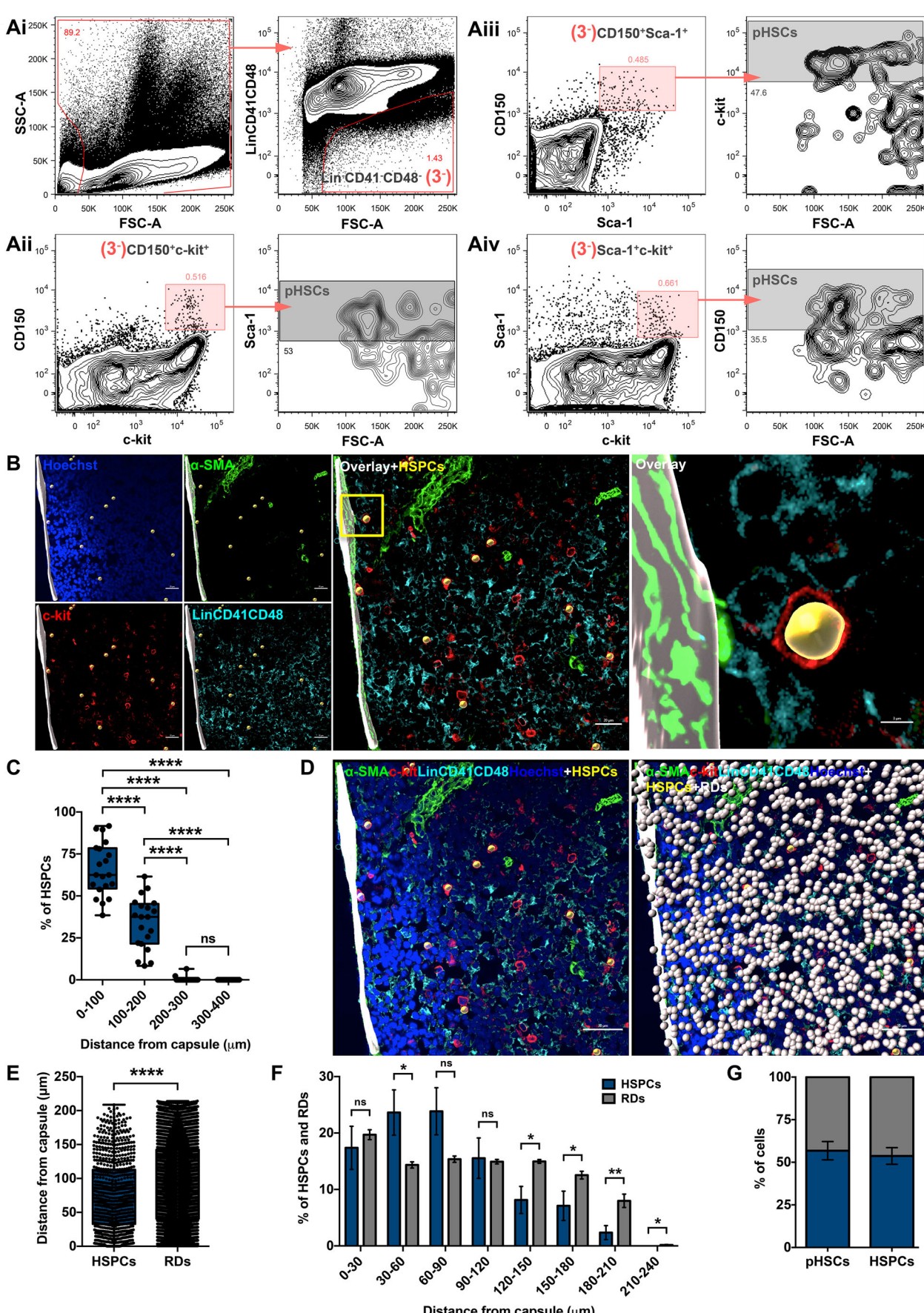

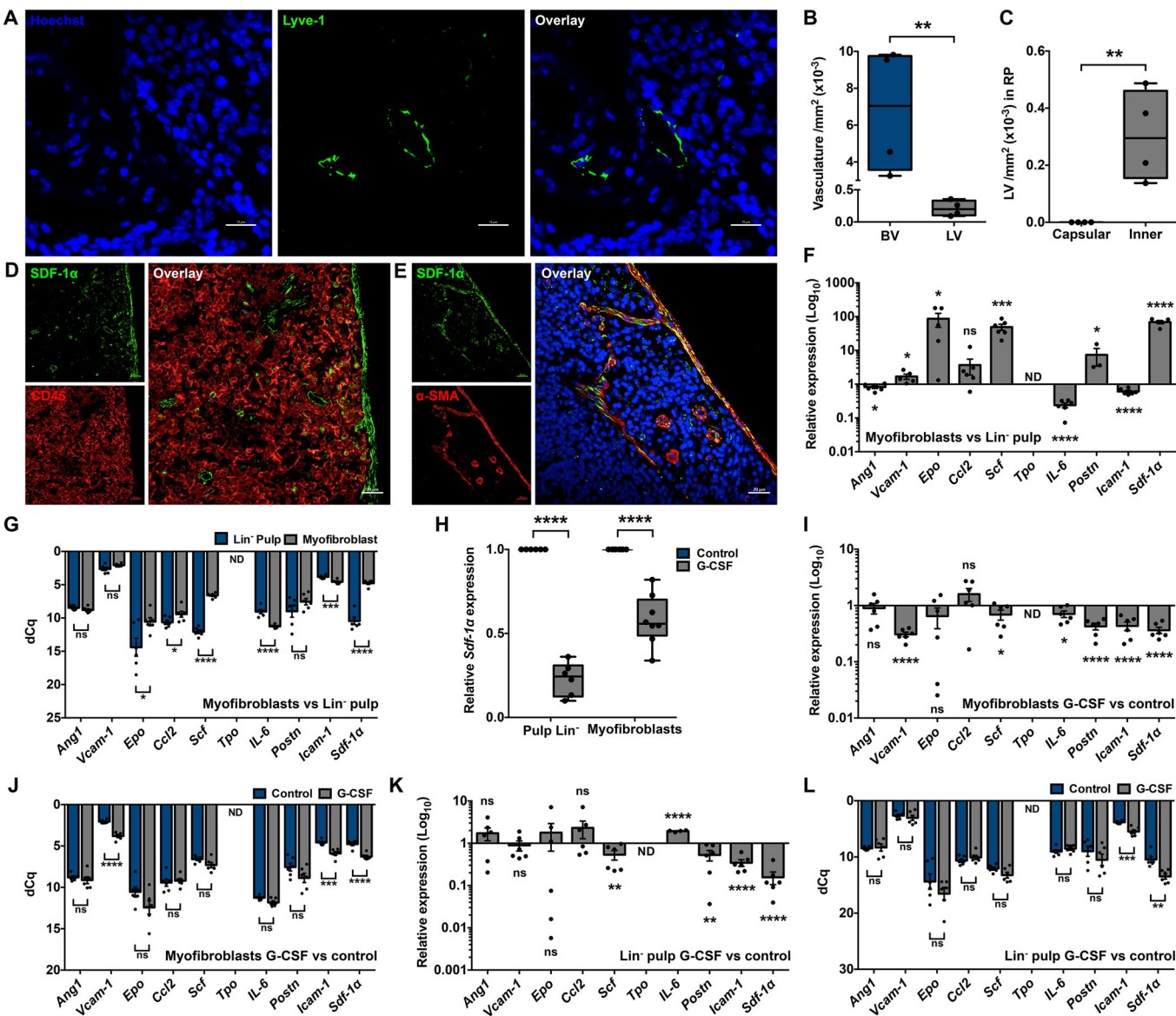

**Figure EV2. G-CSF infusion leads to alterations of hematopoietic niche factors in spleen.**

(A) Immunostaining was performed to detect lymphatic vessels in the spleen using antibodies against lymphatic vessel endothelial hyaluronan receptor 1 (Lyve-1). Counterstaining was performed with Hoechst 33342 and confocal based imaging was performed (scale bar = 15 µm). Splenic CS is divided into 200 µm wide peripheral capsular zone and inner core. (B) Comparison of blood and lymphatic vessel area in splenic tissue normalized to total splenic CS area ($n = 4$). (C) Comparison of vascular distribution of lymphatic vessels in the 200 µm wide peripheral capsular zone and inner core area in the RP of splenic CS ($n = 4$). (D) Confocal microscopy was performed to locate SDF-1α expression in the spleen tissue with reference to hematopoietic and non-hematopoietic fractions identified based on the expression pan-leukocyte antigen CD45 (scale bar = 20 µm). (E) Immunodetection of SDF-1α expression in splenic capsular and trabecular myofibroblasts identified by α-SMA expression. Confocal images show small vessels and trabecular area in the splenic section with immunostaining performed for SDF-1α along with α-SMA and nuclear staining with Hoechst 33342 (scale bar = 20 µm). (F) Detection and comparison of transcript levels of known hematopoietic regulators Ang1, Vcam-1, Epo, Ccl2, Scf, Tpo, IL-6, Postn, Icam-1, and Sdf-1α in splenic myofibroblasts in comparison to Lin⁻ splenic cells by qRT-PCR ($n = 3$–6). (G) Detection and comparison of transcript levels of known hematopoietic regulators Ang1, Vcam-1, Epo, Ccl2, Scf, Tpo, IL-6, Postn, Icam-1, and Sdf-1α in splenic myofibroblasts in comparison to Lin⁻ splenic cells by qRT-PCR; dCq values are compared for each gene ($n = 6$). (H) Effect of G-CSF treatment on the expression of Sdf-1α levels in Lin⁻ spleen cells and myofibroblasts. Quantitative RT-PCR was performed to quantify the transcript levels in the harvested cells and the graph shows relative change in the gene expression with and without G-CSF treatment ($n = 6, 8$). (I) Change in the transcript levels of Ang1, Vcam-1, Epo, Ccl2, Scf, Tpo, IL-6, Postn, Icam-1, and Sdf-1α in splenic myofibroblasts after G-CSF treatment. The gene expression level was quantified by performing qRT-PCR, and relative change in the abundance of transcripts following G-CSF treatment was plotted ($n = 6$). (J) Effect of G-CSF treatment on the transcript levels of Ang1, Vcam-1, Epo, Ccl2, Scf, Tpo, IL-6, Postn, Icam-1, and Sdf-1α in splenic myofibroblasts. The gene expression level was quantified by performing qRT-PCR, and the change in dCq values for each gene was examined following G-CSF treatment ($n = 6$). (K) Quantitative RT-PCR to examine the change in the transcript levels of Ang1, Vcam-1, Epo, Ccl2, Scf, Tpo, IL-6, Postn, Icam-1, and Sdf-1α in lineage-depleted splenic pulp cells after G-CSF treatment. Relative change in the abundance of transcripts is plotted ($n = 4$–6). (L) Change in the transcript levels of hematopoietic regulators in lineage-depleted splenic pulp cells after G-CSF treatment; dCq values before and after G-CSF treatment are compared for each gene ($n = 6$). Data is presented as bar graph (mean ± SEM) in panels (F, G, I–L) or box-whiskers plot (median with min to max) in panels (B, C, H). The p-value in figures (B, C, F–L) were calculated by the Student's t-test, *$p < 0.05$, **$p < 0.01$, ***$p < 0.001$, ****$p < 0.0001$, and ns $p > 0.05$, and ND indicates not detected.

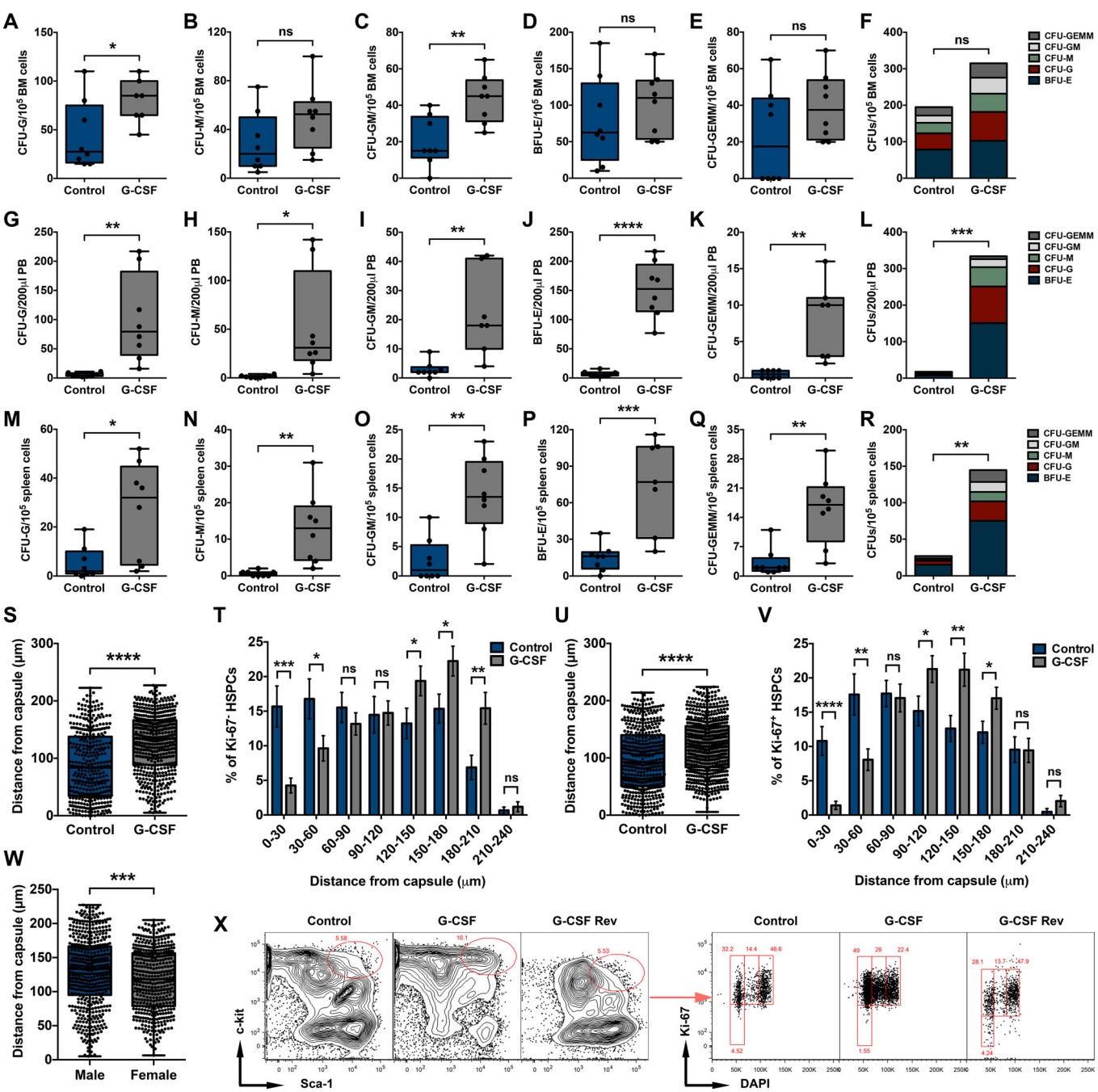

◀ **Figure EV3. G-CSF treatment induces HSC mobilization and shift in HSCs away from capsular myofibroblasts.**

(A–F) Methylcellulose based colony formation assays were performed to examine hematopoietic progenitor populations in the BM ($n = 7$–8). The frequency of (A) CFU-Gs, (B) CFU-Ms, (C) CFU-GMs, (D) BFU-Es, (E) CFU-GEMMs, and (F) total CFUs per $1 \times 10^5$ cells in the BM was compared with and without G-CSF treatment. (G–L) Comparison of the circulating hematopoietic progenitors in PB of mice treated with and without G-CSF ($n = 7$–8). The frequency of (G) CFU-Gs, (H) CFU-Ms, (I) CFU-GMs, (J) BFU-Es, (K) CFU-GEMMs, and (L) total CFUs in the MNCs harvested from 200 μl blood was plotted. (M–R) Colony formation assays were performed to examine the effect of G-CSF treatment on splenic hematopoietic progenitor populations ($n = 7$–8). The frequency of (M) CFU-Gs, (N) CFU-Ms, (O) CFU-GMs, (P) BFU-Es, (Q) CFU-GEMMs, and (R) total CFUs per $1 \times 10^5$ MNCs harvested from spleen tissues was compared. (S) Euclidean distances calculated for Ki-67$^-$ HSPCs ($3^-$c-kit$^+$ cells) relative to the nearest observable capsular myofibroblast of spleen tissues from control and G-CSF treated mice ($n = 6$, N; Control $= 370$, GCSF $= 500$ Ki-67$^-$ HSPCs). (T) Distribution of Ki-67$^-$ HSPCs at sequential distance intervals (30 μm each) relative to the capsular surface in spleen tissues with or without G-CSF treatment ($n = 6$, N; Control $= 370$, GCSF $= 500$ Ki-67$^-$ HSPCs). (U) Euclidean distances calculated for Ki-67$^+$ HSPCs relative to the capsular surfaces of spleen tissues treated with or without G-CSF ($n = 6$, N; Control $= 466$, G-CSF $= 604$ Ki-67$^+$ HSPCs). (V) Distribution frequency of Ki-67$^+$ HSPCs at sequential distance intervals relative to the capsular surface in spleen tissues with or without G-CSF treatment ($n = 6$, N; Control $= 466$, G-CSF $= 604$ Ki-67$^+$ HSPCs). (W) Euclidean distances calculated for each pHSC with reference to pseudo-surfaces of capsular myofibroblast in the spleens harvested from male and female mice after G-CSF treatment. Each dot represents a pHSC immunolocalized as a 3$^-$c-kit$^+$CD150$^+$ cell by confocal imaging ($n = 3$, N; male $= 510$, female $= 447$ pHSCs). (X) Flow cytometry analysis performed to analyze the cell cycle status of HSCs (Lin$^-$CD41$^-$CD48$^-$Sca-1$^+$c-kit$^+$ cell population) from the spleen tissues. Spleen tissues were harvested from control and G-CSF treated animals (same samples were used for data presented in Fig. 7E and Appendix Fig. S1H). G-CSF treatment was given for 5 days and the mice were sacrificed one day (G-CSF) or 30 days (G-CSF Rev) after the treatment. Data is presented as bar graph (mean ± SEM) in panels (T, V) or box-whiskers plot (median with min to max) in panels (A–E, G–K, M–Q, S, U, W) or stacked bars in panel (F, L, R). The p-value in figures (A–W) were calculated by the Student's *t*-test, $*p < 0.05$, $**p < 0.01$, $***p < 0.001$, $****p < 0.0001$, and ns $p > 0.05$.

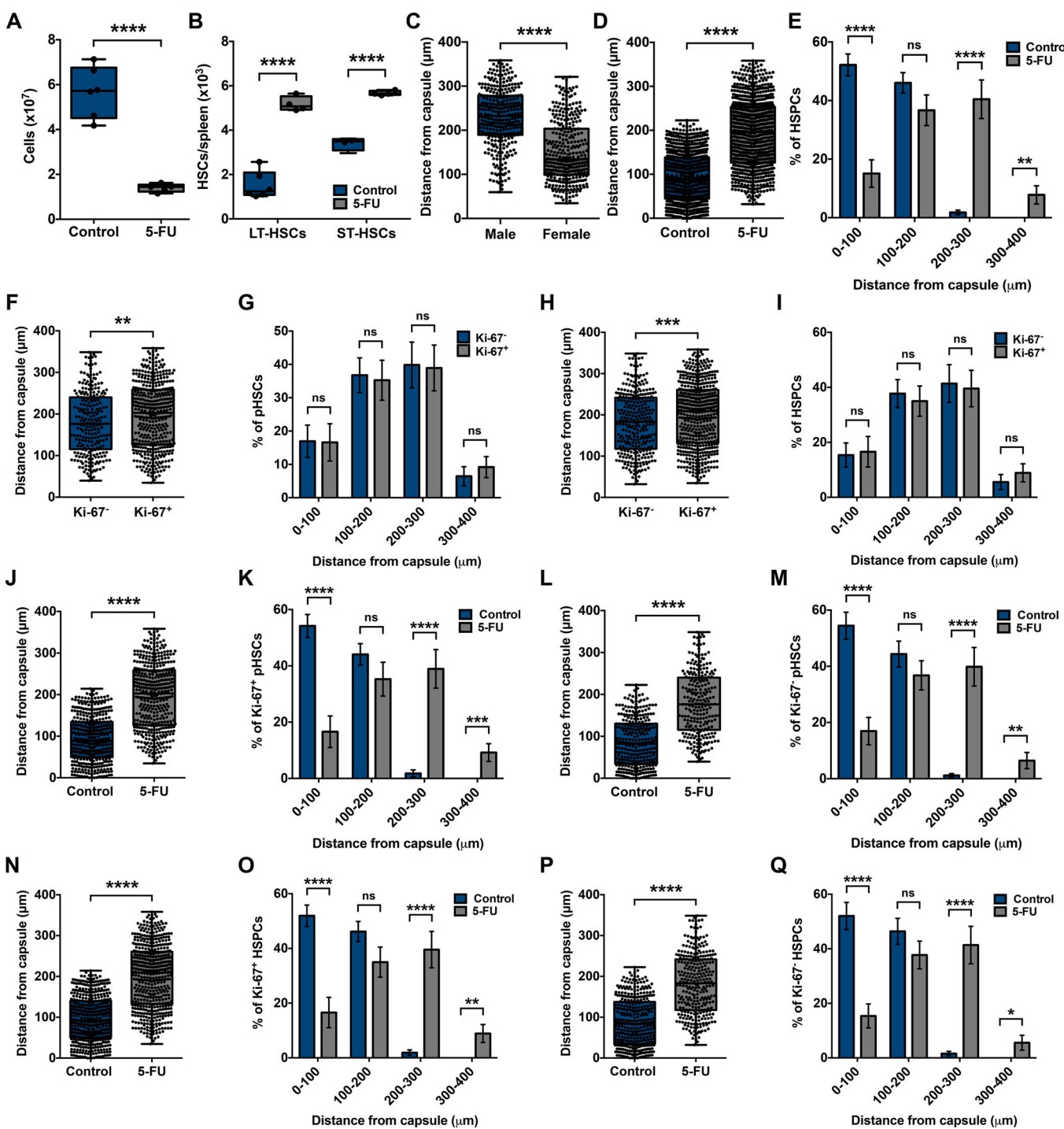

◀ **Figure EV4.  5-FU mediated myeloablation reorganizes hematopoietic niche in spleen.**

(A) Comparison of total spleen cellularity with and without 5-FU treatment. MNCs were harvested from the spleen tissues without enzymatic treatment, and viable cell counts were taken after RBC lysis using a Neubauer chamber ($n = 5–6$). (B) Comparison of total LT-HSC and ST-HSC in spleen tissues treated with and without 5-FU. Flow cytometry analysis was performed to identify and examine LT-HSC (Lin⁻CD41⁻CD48⁻Sca-1⁺c-kit⁺CD150⁺ cells) and ST-HSC (Lin⁻CD41⁻CD48⁻Sca-1⁺c-kit⁺CD150⁻ cells) populations in splenic MNCs harvested from control and 5-FU treated mice ($n = 4–6$). (C) Euclidean distances calculated for each pHSC with reference to pseudo-surfaces of capsular myofibroblasts in the spleen tissues harvested from male and female mice after 5-FU treatment. Each dot represents a pHSC immunolocalized as a 3⁻c-kit⁺CD150⁺ cell by confocal imaging ($n = 2$, $N$; male $= 327$, female $= 333$; each dot represents a pHSCs). (D) Euclidean distances were calculated for each HSPC with reference to nearest capsular surface in the spleen with or without 5-FU treatment. Each dot represents a HSPC immunolocalized as a 3⁻c-kit⁺ cell by confocal imaging ($n = 4–6$, $N$; Control $= 836$, 5-FU $= 796$ HSPCs). (E) Distribution frequency of HSPCs at sequential distance intervals relative to capsular surface in the spleen tissues with or without 5-FU treatment ($n = 4–6$, $N$; Control $= 836$, 5-FU $= 796$ HSPCs). (F) Comparison of Euclidean distances for Ki-67⁻ and Ki-67⁺ pHSCs relative to the nearest capsular surface detected in spleen sections from 5-FU treated mice ($n = 4$, $N$; Ki-67⁻ $= 243$, Ki-67⁺ $= 417$ pHSCs). (G) Comparison between Ki-67⁻ and Ki-67⁺ pHSCs for their distribution frequency within sequential distance intervals relative to capsular surface in the spleen tissues after 5-FU treatment ($n = 4$, $N$; Ki-67⁻ $= 243$, Ki-67⁺ $= 417$ pHSCs). (H) Comparison of Euclidean distances for Ki-67⁻ and Ki-67⁺ HSPCs relative to the nearest α-SMA⁺ myofibroblastic capsular surfaces in spleen sections from 5-FU treated mice ($n = 4$, $N$; Ki-67⁻ $= 300$, Ki-67⁺ $= 496$ HSPCs). (I) Comparison between Ki-67⁻ and Ki-67⁺ HSPCs for their distribution frequency within sequential distance intervals relative to capsular surface in the spleen tissues after 5-FU treatment ($n = 4$, $N$; Ki-67⁻ $= 300$, Ki-67⁺ $= 496$ HSPCs). (J) Euclidean distances were calculated for each Ki-67⁺ pHSC with reference to nearest capsular surface in the spleen with or without 5-FU treatment. Each dot represents a Ki-67⁺ pHSC immunolocalized by confocal imaging ($n = 4–6$, $N$; Control $= 406$, 5-FU $= 417$ Ki-67⁺ pHSCs). (K) Distribution frequency of Ki-67⁺ pHSCs at sequential distance intervals relative to capsular myofibroblastic surface in the spleen tissues with or without 5-FU treatment ($n = 4–6$, $N$; Control $= 406$, 5-FU $= 417$ Ki-67⁺ pHSCs). (L) Comparison of Euclidean distances for Ki-67⁻ pHSCs relative to the nearest capsular surfaces detected in spleen sections from control and 5-FU treated mice ($n = 4–6$, $N$; Control $= 301$, 5-FU $= 243$ Ki-67⁻ pHSCs). (M) Comparison of Ki-67⁻ pHSCs for their distribution frequency within sequential distance intervals relative to capsular surface in the spleen tissues with and without 5-FU treatment ($n = 4–6$, $N$; Control $= 301$, 5-FU $= 243$ Ki-67⁻ pHSCs). (N) Euclidean distances calculated for Ki-67⁺ HSPCs relative to the nearest observable capsular myofibroblastic surface of spleen tissues from control and 5-FU treated mice ($n = 4–6$, $N$; Control $= 466$, 5-FU $= 496$ Ki-67⁺ HSPCs). (O) Distribution of Ki-67⁺ HSPCs at sequential distance intervals relative to capsular surface in the spleen tissues with or without 5-FU treatment ($n = 4–6$, $N$; Control $= 466$, 5-FU $= 496$ Ki-67⁺ HSPCs). (P) Euclidean distances calculated for Ki-67⁻ HSPCs relative to the capsular surfaces of spleen tissues treated with or without 5-FU ($n = 4–6$, $N$; Control $= 370$, 5-FU $= 300$ Ki-67⁻ HSPCs). (Q) Distribution frequency of Ki-67⁻ HSPCs at different distance intervals relative to capsular surface in the spleen tissues with or without 5-FU treatment ($n = 4–6$, $N$; Control $= 370$, 5-FU $= 300$ Ki-67⁻ HSPCs). Data is presented as bar graph (mean ± SEM) in panels (E, G, I, K, M, O, Q) or box-whiskers plot (median with min to max) in panels (A–D, F, H, J, L, N, P). The p-value in figures (A–Q) were calculated by the Student's t-test, *$p < 0.05$, **$p < 0.01$, ***$p < 0.001$, ****$p < 0.0001$, and ns $p > 0.05$.

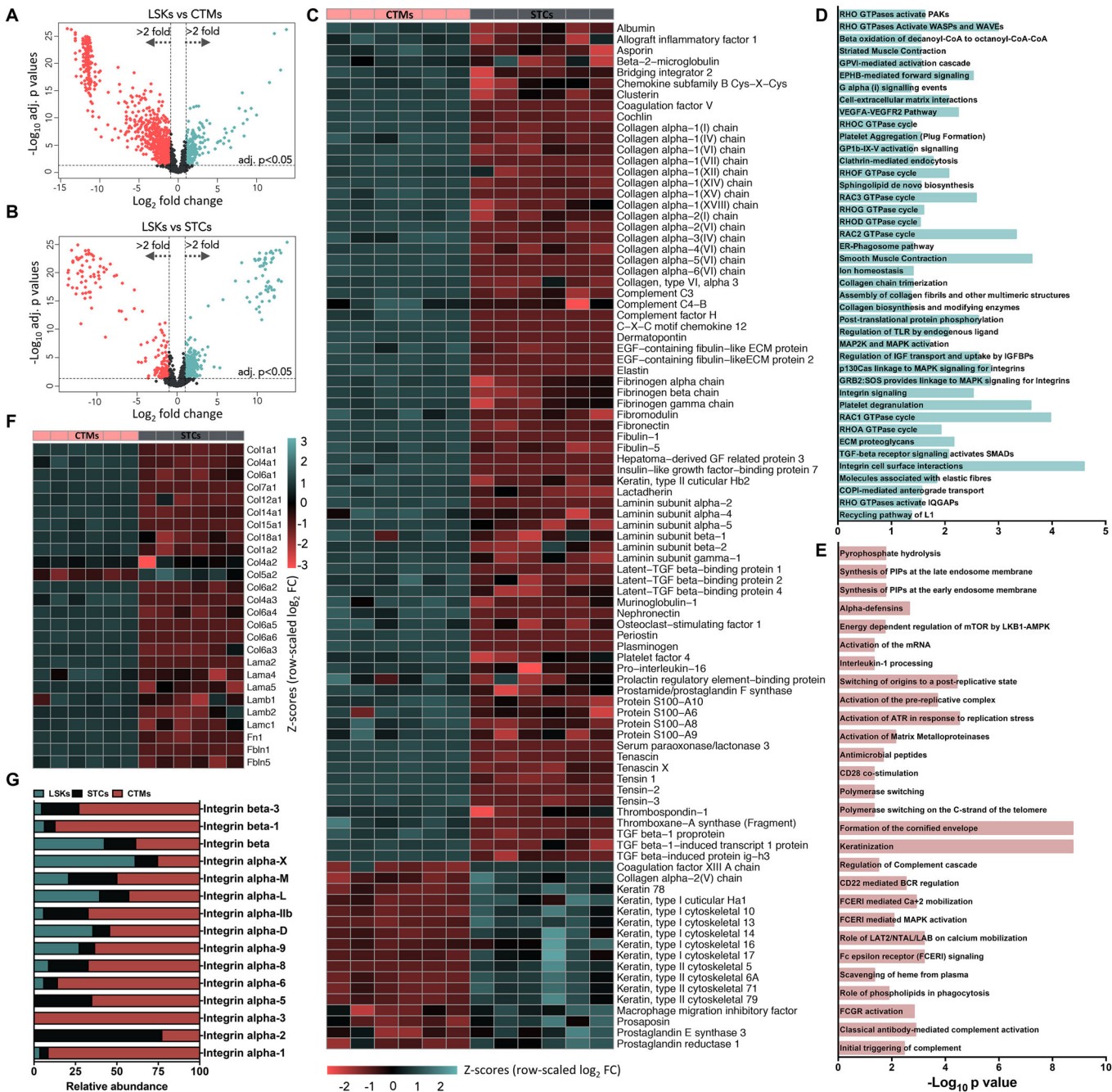

**Figure EV5. Global proteome-based interactions show HSPC interactions preferential to myofibroblastic cells.**

(A) Volcano plot showing proteins differentially enriched between LSK versus CTM populations. Log$_2$ fold change and −log$_{10}$ adjusted *p*-values are plotted on the x and y axes, respectively. Represented data are based on six independent biological replicates (*n* = 6), each with three technical replicates (*N* = 18). Cyan dots represent proteins with statistically significant (adjusted *p*-value < 0.05) higher abundance of >2 fold change. Red dots represent proteins having statistically significant (adjusted *p*-value < 0.05) lower abundance with >2 fold change. (B) Volcano plot showing differentially enriched proteins between LSK versus STC populations (*n* = 6, *N* = 18). (C) Heatmap analysis of differentially enriched secretory proteins in CTMs compared to STCs. The analysis was performed on the secretory proteins with differential enrichment with the adjusted *p*-values of <0.05 and fold change > 2.0. (D) Reactome pathway analysis of differentially enriched pathways based on proteins upregulated with fold change > 2.0 and adjusted *p*-value < 0.05. Significantly (*p*-value < 0.05) up-regulated pathways in CTMs versus STCs are illustrated. (E) Reactome pathway analysis was performed on proteins upregulated with fold change > 2.0 and adjusted *p*-value < 0.05. Differentially enriched pathways significantly (*p*-value < 0.05) up-regulated in STCs compared to CTMs are illustrated. (F) Heatmap analysis of differentially enriched secretory ECM proteins. Secretory ECM proteins with significantly (adjusted *p*-value < 0.05) altered enrichment (fold change > 2.0) between CTMs and STCs are illustrated. (G) Relative abundance of integrin sub-units enriched in LSKs, STCs, and CTMs. Represented data are based on six independent biological replicates (*n* = 6), each with three technical replicates (*N* = 18).

