## [Peer Review File · The EMBO Journal]

A capsular myofibroblastic niche maintains hematopoietic stem cells in the spleen

Shubham Haribhau Mehatre, Sreelakshmi Sanam, Harsh Agrawal, Amulya V. Hejjaji, Akhila S Kumar, Mohammad Tauqeer Alam, Satish Khurana

Corresponding author: Satish Khurana (satishkhurana@iisertvm.ac.in)

Review Timeline:

Submission Date:	1st Nov 24
Editorial Decision:	19th Dec 24
Revision Received:	3rd Apr 25
Editorial Decision:	30th Apr 25
Revision Received:	6th May 25
Accepted:	9th May 25

Editor: Daniel Klimmeck

Transaction Report:

Dear Dr Khurana,

Thank you for the submission of your manuscript (EMBOJ-2024-119500) to The EMBO Journal, as well as for your patience with our feedback at this time of the year. As mentioned earlier, your study was assessed by three reviewers with expertise in developmental hemopoiesis, HSC niche analysis and stromal proteomics, whose comments are enclosed below.

As you will see from the experts' reports, the referees acknowledge the analysis and potential interest and value of your findings. However, they also express important issues regarding the completeness of the results, which need to be addressed thoroughly to make them supportive of publication in the EMBO Journal. Further, the reviewers raise a number of issues related to the presentation of the findings, additional controls and improved methods annotation required, statistics applied and overall discussion of related literature, that would need to be conclusively addressed to achieve the level of robustness and clarity needed for The EMBO Journal.

Given the overall interest stated and broader angle of your findings, we are able to invite you to revise your manuscript experimentally to address the referees' comments. I need to stress though that we do require strong support from the referees on a revised version of the study in order to move on to publication of the work.

I would appreciate if you could contact me during the next weeks for exchange e.g. a video call to discuss your perspective on the comments and potential plan for revisions.

Please feel free to contact me if you have any questions or need further input on the referee comments.

When submitting your revised manuscript, please carefully review the instructions below.

Please feel free to approach me any time should you have additional questions related to this.

Thank you for the opportunity to consider your work for publication.

I look forward to your revision.

Kind regards,

Daniel Klimmeck

Daniel Klimmeck, PhD
Senior Editor
The EMBO Journal

Instruction for the preparation of your revised manuscript:

- 1) a .docx formatted version of the manuscript text (including legends for main figures, EV figures and tables). Please make sure that the changes are highlighted to be clearly visible.
- 2) individual production quality figure files as .eps, .tif, .jpg (one file per figure).
- 3) a .docx formatted letter INCLUDING the reviewers' reports and your detailed point-by-point response to their comments. As part of the EMBO Press transparent editorial process, the point-by-point response is part of the Review Process File (RPF), which will be published alongside your paper.
- 4) a complete author checklist, which you can download from our author guidelines ([https://wol-prod-cdn.literatumonline.com/pb-assets/embo-site/Author Checklist%20-%20EMBO%20J-1561436015657.xlsx](https://wol-prod-cdn.literatumonline.com/pb-assets/embo-site/Author%20Checklist%20-%20EMBO%20J-1561436015657.xlsx)). Please insert information in the checklist that is also reflected in the manuscript. The completed author checklist will also be part of the RPF.

6) It is mandatory to include a 'Data Availability' section after the Materials and Methods. Before submitting your revision, primary datasets produced in this study need to be deposited in an appropriate public database, and the accession numbers and database listed under 'Data Availability'. Please remember to provide a reviewer password if the datasets are not yet public (see <https://www.embopress.org/page/journal/14602075/authorguide#datadeposition>).

7) Our journal encourages inclusion of *data citations in the reference list* to directly cite datasets that were re-used and obtained from public databases. Data citations in the article text are distinct from normal bibliographical citations and should directly link to the database records from which the data can be accessed. In the main text, data citations are formatted as follows: "Data ref: Smith et al, 2001" or "Data ref: NCBI Sequence Read Archive PRJNA342805, 2017". In the Reference list, data citations must be labeled with "[DATASET]". A data reference must provide the database name, accession number/identifiers and a resolvable link to the landing page from which the data can be accessed at the end of the reference. Further instructions are available at .

8) At EMBO Press we ask authors to provide source data for the main and EV figures. Our source data coordinator will contact you to discuss which figure panels we would need source data for and will also provide you with helpful tips on how to upload and organize the files.

Numerical data can be provided as individual .xls or .csv files (including a tab describing the data). For 'blots' or microscopy, uncropped images should be submitted (using a zip archive or a single pdf per main figure if multiple images need to be supplied for one panel). Additional information on source data and instruction on how to label the files are available at .

9) We replaced Supplementary Information with Expanded View (EV) Figures and Tables that are collapsible/expandable online (see examples in <https://www.embopress.org/doi/10.15252/embj.201695874>). A maximum of 5 EV Figures can be typeset. EV Figures should be cited as 'Figure EV1, Figure EV2" etc. in the text and their respective legends should be included in the main text after the legends of regular figures.

11) For data quantification: please specify the name of the statistical test used to generate error bars and P values, the number (n) of independent experiments (specify technical or biological replicates) underlying each data point and the test used to calculate p-values in each figure legend. The figure legends should contain a basic description of n, P and the test applied. Graphs must include a description of the bars and the error bars (s.d., s.e.m.).

We realize that it is difficult to revise to a specific deadline. In the interest of protecting the conceptual advance provided by the work, we recommend a revision within 3 months (19th Mar 2025). Please discuss the revision progress ahead of this time with the editor if you require more time to complete the revisions.

Referee #1:

Mehatre and colleagues provide a novel insight into the niche of splenic hematopoietic stem cells (HSCs). They showed that an exclusive myofibroblastic niche in the red pulp supports the quiescence of HSCs in the spleen. By utilizing a quantitative proteomic approach, they were able to show that the myofibroblastic niche plays a supportive role in maintaining the population of HSCs by secreting factors including CXCL12/SDF-1a that regulate their survival and proliferation. Interestingly, they showed that G-CSF induces a decrease in SDF-1a, which leads to a change in the spatial distribution of HSCs away from the myofibroblastic niche. Finally, myeloablation with 5-FU induced proliferation of splenic HSCs, leading to expansion of the haematopoietic zone.

Overall, this is a very well-written and presented manuscript describing an interesting and highly relevant mechanism of HSC regulation. The authors present their findings clearly and describe them accurately within the text using imaging approaches. I really enjoyed reading this paper. A real strength of this study is the detailed description of the splenic myofibroblastic niche that regulates HSC quiescence and proliferation.

There are only some minor open questions in the study design and the current set of data as listed below.

Below are comments that aim to further increase the clarity and significance of this exciting study:

1. Do the authors find any differences between male and female mice in the splenic niche, especially after stresses such as G-CSF or 5-FU administration?
2. Can the authors comment on the distribution of blood and lymphatic vessels in the red and white pulp, and comment whether the myofibroblastic niche has a greater / lower concentration of these structures?
3. In the G-CSF, the authors demonstrated that HSC migrate away from the capsular myofibroblast. Could the authors provide insight into the distance to blood vessels and lymphatic vessels in this context?

Referee #2:

This manuscript presents results highlighting an HSC niche of myofibroblasts in the red pulp of the spleen. Imaging data are supported by cellular proteomic data, and I think the analyses that follow the proteomic data are logically selected. In my view the study shows valuable insights into the splenic HSC niche and the findings are of relevance not only to steady state HSC biology but also contexts of dysregulated hematopoiesis.

Major concerns:

1) I have some doubts regarding the choice of markers for identification of the pHSCs by fluorescence microscopy. The authors exclude Sca-1 from the panel based on the FACS-data shown in Appendix Figure S1 and that the proportion of Lin-CD41-CD48-cKit+Sca1+CD150+ is highest when excluding Sca1 compared to cKit or CD150. My concern is the gating for positivity of CD150 has been set lower in Aii than Aiii and Aiv, which might affect that conclusion and therefore also the validity of the imaging data. I argue the authors need to show the data of proportion pHSC when they set the gates at the same positivity level in Aii, Aiii and Aiv.

2) The proteomic analysis is interesting and valuable but is lacking information to understand the level of interpretation that can be made. The authors need to add description of the statistical part in the proteomic analysis and provide more details to the expression 'followed by the generation of secretome profile of niche cells' in the methods. The analysis behind Figure 3H also needs clarification in the methods and legend - it is especially unclear whether the figure shows interaction (which would be interpreted as direct binding) or regulation (which could be downstream effects of a binding), and this needs to be clear in the main text as well. On the one hand words like 'regulated' are used, and on the other hand 'interaction' and 'ligand' are used.

3) In relation to Figure 3G, I think it is worthwhile to highlight that the CTMs have a higher proportion of higher abundant secretory proteins than STCs (which could indicate that they are more prone to influence their microenvironment).

4) Regarding the secretory proteins, I think the results are over-interpreted when it comes to secretion, since the data is a global proteomics dataset, and the detection of proteins that are known to be secreted does not equal actual secretion (some proteins can play a role intracellularly as well). And a higher intracellular expression of a 'secretory protein' does not necessarily mean that it is secreted from those cells to a higher degree; the other cell type could still have a higher secretion activity. Without data on actual secretion (e.g. secretome analysis from in vitro cultures or proteomic data of extracellular fluid of the spleen), I think the authors can tone down the claims on secretion. Amongst other things, on page 8, the expression "the ligand was exclusively secreted by.." could e.g. be changed to "the ligand was a secretory protein found higher expressed in CTMs".

Minor concerns and suggestions:

5) Figure legend of Figure 2B: I think 'upper panel' and 'lower panel' have been mixed up with 'left panel' and 'right panel'.

6) In the methods on page 20, 'MS/MS spectra recorded from 200 to 2000 scans' should be 'MS/MS spectra recorded from 200 to 2000 mass/charge ratio'.

7) The methods description for protein identification merit details on missed cleavages allowed as well as the settings for variable or fixed modifications (particularly since there was an additional step of reduction with DTT in the sample preparation) and FDR rate.

8) On page 7, the STCs have been defined as Lin-CD45+ instead of Lin-CD45-.

9) On page 7, 'The cells were thermally denatured' is an incorrect expression since it is the proteins that are referred to.

10) Figure 3A: It looks like the CTMs have been part of the FACS sort - placing them more logically outside the FACS plot area would avoid misunderstandings.

11) Figures 3D and E: In three pairs of comparisons, I find that it makes little sense to display the overlap separate for "up-" and "down-regulated" proteins, since the direction of that expression difference will depend on how the authors have selected the numerator (and denominator). I suggest either one figure, with three circles of "differentially expressed" (i.e. up- and down-regulated together for each comparison), or six figures, of 2 circles each, showing overlap of differentially expressed proteins between pairs with the same denominator and same direction of change, e.g. up-regulated in CTMs vs STCs and upregulated in LSKs vs STCs.

12) The legend for Figure 3D and 3E does not state that fold change of >2 was used, while the main text referring to those figures on page 7 do.

13) The sentence "Next, we identified the differentially expressed proteins ..." comes a bit out of place since the differentially expressed proteins have already been mentioned and defined in the previous sentence.

14) Heatmaps in Figure 3C, and Appendix Figure S2 F and C: the colour bars need to show what ratio it refers to so the reader knows which colour is high and what is low expression, respectively. In addition, the colour "direction" has been reversed in panel S2F compared to the others - ideally they would use the same direction to avoid confusion.

Referee #3:

In this study, the authors identify a capsular myofibroblastic niche in the spleen that supports HSCs under steady-state conditions. Through imaging analysis, they show that HSCs are closer to the capsule than HPSCs, with quiescent HSCs even more spatially linked to myofibroblasts. Proteomic profiling shows that myofibroblasts express key hematopoietic regulators, like high levels of CXCL12/SDF-1 α , and pathway analysis supports that they secrete most ligands influencing HSPCs. To test the functional relevance, the authors performed a G-CSF treatment as a tool to reduce SDF-1 α levels, which led to altered HSC spatial distribution in the spleen. Similarly, through 5-FU-induced myeloablation, they linked HSC proliferation to their spatial positioning, reinforcing the connection between quiescence and the niche. Overall, their findings highlight the splenic capsular myofibroblastic niche as an important microenvironment for HSC maintenance.

The study is well executed, and the experiments support their conclusions. That said, one aspect that remains unclear to us is whether the changes in HSC spatial distribution following 5-FU or G-CSF treatment involve resident splenic HSCs or mobilized HSCs from the bone marrow. Additionally, further comparison of how these findings relate to the BM niche would provide valuable context and further emphasize the importance of understanding niche-specific regulation of HSCs.

Other/Minor comments:

• Typo in the title: "myofibroblastsic" should be corrected.

• Page 7: "Niche samples were prepared from FACS-sorted Lin-CD45+ splenic cells (hereafter referred to as stromal cells or STCs) and myofibroblasts from the fibrous splenic tissue (hereafter referred to as capsular and trabecular myofibroblasts or CTMs)". - Do the authors mean Lin-CD45- or Lin-CD45+ cells? Please clarify.

- Figure 8: The authors refer to quiescent HSCs as "dormant." It would be more accurate to use "quiescent" to avoid any potential misinterpretation.
- Methodology: While the authors describe the proteomics workflow, including the population and downstream analysis, they do not provide information on the number of cells used. This should be included for clarity and reproducibility.
- The authors use two different stress conditions to examine how the splenic niche affects HSC spatial distribution. This is an interesting finding. Could the authors discuss its translational relevance, particularly in the context of extramedullary hematopoiesis? How might this knowledge be applied in clinical or pathological settings?

Authors' response to the reviewers' comments:

First of all, we thank the reviewers for going through the manuscript extremely carefully and critically. We appreciate the overall positive commentary from all the reviewers. We feel satisfied that the message and importance of the presented study has been correctly conveyed. We have gone through each comment and suggestion made by the reviewers. We have found the comments very supportive, encouraging and insightful. We have made every effort to address the concerns raised and, wherever required, have added additional data/information. We believe that working on these suggestions has enhanced the scope and impact of the manuscript, with confirmation of our conclusions. In the following section, we respond to each comment from the reviewers and explain the changes that we have brought about in the manuscript to address the queries.

Most of the additional data that we have added in the revised manuscript is also presented here along with our responses to the comments made. This was done to assist the referees to follow the responses in light of the new data. A table linking each panel of these figures with the revised manuscript figures has been provided at the end of these responses. In the manuscript, all changes brought about during revision have been indicated in blue.

Referee #1:

Mehatre and colleagues provide a novel insight into the niche of splenic hematopoietic stem cells (HSCs). They showed that an exclusive myofibroblastic niche in the red pulp supports the quiescence of HSCs in the spleen. By utilizing a quantitative proteomic approach, they were able to show that the myofibroblastic niche plays a supportive role in maintaining the population of HSCs by secreting factors including CXCL12/SDF-1a that regulate their survival and proliferation. Interestingly, they showed that G-CSF induces a decrease in SDF-1a, which leads to a change in the spatial distribution of HSCs away from the myofibroblastic niche. Finally, myeloablation with 5-FU induced proliferation of splenic HSCs, leading to expansion of the haematopoietic zone.

Overall, this is a very well-written and presented manuscript describing an interesting and highly relevant mechanism of HSC regulation. The authors present their findings clearly and describe them accurately within the text using imaging approaches. I really enjoyed reading this paper. A real strength of this study is the detailed description of the splenic myofibroblastic niche that regulates HSC quiescence and proliferation. There are only some minor open questions in the study design and the current set of data as listed below.

Below are comments that aim to further increase the clarity and significance of this exciting study:

Comment 1. Do the authors find any differences between male and female mice in the splenic niche, especially after stresses such as G-CSF or 5-FU administration?

Figure 1. Splenic pHSCs show sex-specific spatial phenotypic differences in the hematopoietic niche. (A) Euclidean distances calculated for each pHSC with reference to pseudo-surfaces of capsular myofibroblasts in the spleens harvested from male and female mice. (B) Distribution frequency of pHSCs at sequential distance intervals (30 μm each) relative to the myofibroblastic capsule in spleen tissues isolated from male and female mice. (C) Euclidean distances calculated for pHSCs relative to the capsular surfaces of splenic CS from male and female mice treated with G-CSF. (D) Euclidean distances calculated for pHSCs relative to the capsular surfaces of splenic CS from male and female mice treated with 5-FU.

Authors: This is an interesting point raised by the reviewer. Sex differences have been shown to play important role in the functioning of stem cell population. In our experiments, consistency in splenic hematopoietic niche in the male and female mice was noted. Irrespective of the sex of the animals used for analysis, the primitive HSCs (pHSCs) were located closer to the myofibroblasts in comparison to the random dots. Therefore, data from both sexes were combined for statistical analysis and drawing conclusions. Further analysis has revealed a robust difference in the relative mean distance from the splenic capsule at which the pHSCs were located in male and female mouse. In the revised manuscript, we have added data demonstrating that the pHSCs in splenic tissue of female mice are localized in closer proximity to the capsular myofibroblasts ($68.8 \pm 1.9 \mu\text{m}$) when compared to the male mice ($113.0 \pm 3.0 \mu\text{m}$) (Fig. 1A). Spatial distribution analysis of pHSCs with reference to the myofibroblasts also confirmed these results as the proportion of pHSCs with in 0-30 as well as 30-60 μm intervals was significantly higher in females than in male mice (Fig. 1B). As queried by the reviewer, we analyzed if the impact of G-CSF and 5-FU treatment on the reorganization of the hematopoietic niche was differential in males and females. This difference in the localization of the pHSCs in male and females was consistent even after treatment with G-CSF (Fig. 1C) as well as 5-FU (Fig. 1D). It could also be easily noticed that both of the treatments increased the mean distance of pHSCs from the splenic capsule in male and female mice. However, in the case of 5-FU, which led to a significant disruption of splenic architecture, the relocation of pHSC away from the capsule was more robust in the male mice than in the females. From these results, we concluded that; 1) in both males and females pHSCs moved away from the myofibroblasts significantly, and 2) the pHSCs in female mice remained significantly closer to the capsule than in the male mice after G-CSF and 5-FU treatment. These results are added in the Figures 2, EV3 and EV4 of the revised manuscript (details in Annexe Table 1). The relevance of these observations is not immediately clear but our ongoing experiments have the potential to explain these observations. We are actively exploring proteomics data from splenic myofibroblasts to understand these differences. We have performed these experiments in the context of activation of extramedullary hematopoiesis in spleen during pregnancy. We are currently developing these studies and with these interesting observations, we will be able to explore the results in new light.

Comment 2. Can the authors comment on the distribution of blood and lymphatic vessels in the red and white pulp, and comment whether the myofibroblastic niche has a greater / lower concentration of these structures?

Authors: In deed, vasculature plays a key role in the molecular regulation of HSC function, and in the creation of HSC niche in hematopoietic sites such as BM and fetal liver. This is a pertinent question,

Figure 2. Distribution of blood vessels in RP and WP areas of spleen. (A) Immunostaining was performed to detect blood vessels distribution with reference to WP and RP areas in the spleen, and tilescans of the entire splenic cross-sections (CS) were acquired. Marginal macrophages lining the WP areas were immunostained for CD169 along with pan-endothelial marker CD31 and vascular smooth muscle cell marker α -SMA (Middle panel; scale bar=200 μm). Left panel; Confocal images showing a large blood vessel in the RP area of the splenic section (scale bar=20 μm). Right panel; Confocal images showing blood vessels in the WP area of splenic tissue (right panel; scale bar=50 μm). (B) Comparison of blood vessel distribution in the RP and WP areas of the spleen (n=4). (C) Comparison of blood vessel distribution within the capsular zone (<200 μm wide zone along splenic capsule) and inner core (>200 μm distant from the splenic capsule).

Figure 3. Lymphatic vessels rarely appear in the capsular niche. Lymphatic vessels identified by immunostaining for Lyve-1. Spleen CS were immunostained for Lyve-1, along with myofibroblastic marker α -Sma and nuclear counterstaining Hoechst 33342 (scale bar=15 μ m). Lymphatic vessels were not found in the vicinity of myofibroblasts marked by α -SMA. (B) Comparison of blood and lymphatic vessel areas normalized to the total splenic CS area (n=4). (C) Distribution of lymphatic vessels in the quiescent capsular zone versus inner splenic core CS (n=4).

following which we made significant efforts to understand if local vasculature was involved in the determination of the spatial localization of HSCs in spleen tissue. We performed immunostaining on murine splenic cryosections using antibodies against CD31 (for vascular endothelial cells) and α -SMA (for vascular smooth muscle cells and myofibroblasts), alongside CD169 to distinguish between red pulp (RP) and white pulp (WP) regions (Fig. 2A, middle panel). First, we analyzed the overall distribution of vessels within the RP (Fig. 2A, left panel) and WP (Fig. 2A, right panel) areas and noted that vessel density was significantly higher in the WP than in the RP (Fig. 2B). As our experiments showed that the pHSCs were restricted to the RP areas, these results indicated that vasculature might not be the determining factor in the creation of splenic hematopoietic niche. To further confirm this hypothesis, we examined the density of blood vessels in the capsular (200 μ m wide zone from the capsular lining) versus inner splenic core within the RP area. We observed no significant difference in the vascular density in the capsular zone and inner core of splenic RP area (Fig. 2C). These results further established that the vascular components might not be crucial for the localization of pHSCs in splenic tissue. These results have been added in the Figure 3 of the revised manuscript.

We then investigated the presence of lymphatic vessels in pHSC-enriched regions (results added in the Fig. EV2 of the revised manuscript). We performed immunostaining for Lyve-1, a lymphatic endothelial marker (Fig. 3A). We screened through a large number of spleen sections to detect the presence of Lyve-1⁺ lymphatic vessels that were rare to detect, and were restricted to RP areas. We made the following observations and the data is added in the revised manuscript;

A. As compared to the vasculature identified by CD31 immunostaining, the Lyve1⁺ lymphatic vessels were rare in appearance (Fig. 3B).

B. Within the RP areas, they did not show any preference towards the capsular zone that hosted quiescent pHSCs. In fact, they were preferentially located in the inner splenic areas devoid of pHSCs.

Overall, the studies on the distribution of blood or lymphatic vessels and spatial distribution of pHSCs could not establish any correlation. We concluded that vasculature does not play a significant role in the maintenance of hematopoietic niche in splenic tissue.

Comment 3. In the G-CSF, the authors demonstrated that HSC migrate away from the capsular myofibroblast. Could the authors provide insight into the distance to blood vessels and lymphatic vessels in this context?

Authors: We see this comment in conjunction with the earlier comment and appreciate the suggestion that HSC localization with reference to vasculature should be looked at more carefully. As we had noted a clear localization of pHSCs in the capsular zone, we had not looked into the role of vasculature in their spatial distribution. Our experiments performed in response to the previous question also showed no correlation between vasculature (blood as well as lymphatic vessels) and pHSC localization. This was in contrast to the BM HSC niche for which the importance of vasculature and associated structures has been demonstrated (Kiel MJ et al. in Cell 2005; Kunisaki Y et al. in Nature 2013; Acar M et al. in Nature 2015). In response to a similar question from reviewer 3, we have performed extensive spatial distribution analysis for HSPCs with reference to the splenic vasculature. As can be seen from our detailed response to this comment from

Figure 4. Splenic capsular niche revival following reversal of G-CSF effects: (A) Comparison of cross-sectional area of the murine spleen tissues harvested from control and G-CSF treated animals. G-CSF treatment was given for 5 days and the mice were sacrificed one day (G-CSF) or 30 days (G-CSF Rev) after the treatment. (B) Comparison of total spleen cellularity (mononuclear cell counts) in control and the two groups of G-CSF treated mice. (C) Flow cytometry analysis performed to analyze the cell cycle status of HSCs (Lin⁻CD41⁻CD48⁻Sca-1⁺c-kit⁺ cell population) from the spleen tissues. The cells were immunostained for Ki-67, Lineage markers, CD41, CD48, c-kit and Sca-1 along with labelling with nucleic acid binding dye DAPI. (D) Comparison of proportion of splenic HSC population in different stages of cell cycle analyzed by flow cytometry. (E) Euclidean distances calculated for each pHSC with reference to pseudo-surface of capsular myofibroblasts in the spleen from control, G-CSF and G-CSF Rev groups of mice. Each dot represents a pHSC immunolocalized as a 3-c-kit⁺CD150⁺ cell by confocal imaging (n=6, N; control=707, G-CSF=957 and G-CSF Rev=629 pHSCs). (F) Distribution frequency of pHSCs at sequential distance intervals (30 µm each) relative to the splenic capsule in spleen of control, G-CSF treated and mice undergone one month incubation after G-CSF treatment (G-CSF Rev).

reviewer 3, the HSPCs in splenic tissue were randomly distributed with regard to the vascular endothelium (Fig. 8A). The mean distance (Fig. 8B) as well as the spatial distribution of HSPCs relative to the vessels identified by CD31 expression did not differ when compared with the random dots (Fig. 8C). In addition, we show that the spatial distribution of HSPCs relative to the vasculature did not change following G-CSF treatment (Fig. 8D-F). The detailed analysis performed in response to the queries pertaining the involvement of vasculature in splenic hematopoietic niche is now added in Figures 3 and 5 in the revised manuscript (details in annex table 1).

We made another interesting observation with regard to the HSCs' response to G-CSF that was not queried but we believe can be an important addition to the manuscript. It is regarding the revival of the niche occupied by the pHSCs upon reversal of the effects of G-CSF treatment. When we analyzed the splenic tissues one month after G-CSF treatment, we noted a complete reversal of spleen size (Fig. 4A) and cellularity (Fig. 4B) along with HSC cell cycle status (Fig. 4C,D). This prompted us to ask if the pHSCs upon attaining quiescence regained their position with respect to the myofibroblasts. The results obtained were striking as we noted that there was a complete restructuring of the niche and spatial distribution of pHSCs with reference to myofibroblasts was regained (Fig. 4E,F). These results further confirmed our conclusions that splenic capsular niche hosts pHSCs and is crucial to maintain them in the quiescent stage. These results are added in Figure 5 and EV3 of the revised manuscript.

Referee #2:

This manuscript presents results highlighting an HSC niche of myofibroblasts in the red pulp of the spleen. Imaging data are supported by cellular proteomic data, and I think the analyzes that follow the proteomic data are logically selected. In my view the study shows valuable insights into the splenic HSC niche and the findings are of relevance not only to steady state HSC biology but also contexts of dysregulated hematopoiesis.

Major concerns:

Comment 1. I have some doubts regarding the choice of markers for identification of the pHSCs by fluorescence microscopy. The authors exclude Sca-1 from the panel based on the FACS-data shown in Appendix Figure S1 and that the proportion of Lin-CD41-CD48-cKit+Sca1+CD150+ is highest when excluding Sca1 compared to cKit or CD150. My concern is the gating for positivity of CD150 has been set lower in Aii than Aiii and Aiv, which might affect that conclusion and therefore also the validity of the imaging data. I argue the authors need to show the data of proportion pHSC when they set the gates at the same positivity level in Aii, Aiii and Aiv.

Authors: We understand the concern raised by the reviewer and appreciate the importance attached to this analysis as it was used for the selection of markers for our immunostaining method to detect pHSCs in the spleen tissue. The gating strategy shown in Appendix Fig. S1 was based on isotype controls but there were minor shifts in populations based on the fluorophore combination used for gating. In response to this comment, we have thoroughly reexamined the results and conducted complete reanalysis of the data. In the revised manuscript, we present reanalyzed data with more stringent and consistent gating strategy that

Figure 5. Flow cytometry based selection of marker combination to identify pHSCs based on three fluorophores. (A) Flow cytometry was performed on splenic MNCs to evaluate different marker combinations to identify primitive HSCs (pHSCs or CD150⁺CD41-CD48-LSK cells) using three fluorophores. The Lin-CD41-CD48 (or 3-) cells (Ai) were further gated on CD150⁺c-kit⁺ (Aii), CD150⁺Sca-1⁺ (Aiii), and Sca-1⁺c-kit⁺ (Aiv) cells; and the proportion of pHSCs in each one of them was examined (n=5). (B) The revised proportion of pHSCs in Lin-CD41-CD48 (or 3-)CD150⁺c-kit⁺, 3-CD150⁺Sca-1⁺ and 3-Sca-1⁺c-kit⁺ cells was examined and plotted (n=5). (C) The proportion of pHSCs identified by the three schemes presented in the first version of the manuscript submitted (n=5).

we believe is best suited for selecting the marker combination. This analysis has further confirmed the validity of the markers used to identify splenic pHSCs. Fig. 5 shows the updated gating strategy (Fig. 5Ai-Aiv) along with the proportion of pHSCs identified using each marker combination (Fig. 5B). While there is a slight change in the proportion of pHSCs represented by each strategy, overall conclusion that lin-CD41-CD48-c-kit+CD150⁺ cells (Fig. 5Aii) represent the highest proportion (77.5±6.4%) of pHSCs remain consistent as shown in the previously submitted manuscript (Fig. 5C). In fact, in comparison to the other marker combinations (lin-CD41-CD48-CD150⁺Sca-1⁺ at 45.6±2.9% and lin-CD41-CD48-Sca-1⁺c-kit⁺ at 48.8±1.5%; Fig. 5Aiii and Fig. 5Aiv, respectively) the chosen strategy was significantly better at identifying pHSCs. We thank the reviewer for this suggestion. This gating strategy and the resulting data has been updated in Figure 1 and EV1 of the revised manuscript.

Comment 2. The proteomic analysis is interesting and valuable but is lacking information to understand the level of interpretation that can be made. The authors need to add description of the statistical part in the

proteomic analysis and provide more details to the expression 'followed by the generation of secretome profile of niche cells' in the methods. The analysis behind Figure 3H also needs clarification in the methods and legend - it is especially unclear whether the figure shows interaction (which would be interpreted as direct binding) or regulation (which could be downstream effects of a binding), and this needs to be clear in the main text as well. On the one hand words like 'regulated' are used, and on the other hand 'interaction' and 'ligand' are used.

Authors: We used proteomic analysis to further confirm the importance of the interaction between HSCs and myofibroblasts as their niche. This was not just aimed at confirming the crucial involvement of myofibroblastic niche over the splenic stroma, but also to ultimately work towards identifying novel hematopoietic regulators. Our future goal also includes developing understanding of the differences between splenic and BM niche that can ultimately define the differential role that respective HSC population plays in overall hematopoietic processes. We are glad that the reviewer has found the proteomics data valuable and thank the reviewer for making important suggestions. We accept the point raised by the reviewer and see that there are methodology related details that were needed to be furnished. They were missed, at times unintentionally, and some were mistakenly considered avoidable details.

We have incorporated changes in response to the suggestions made by the reviewer and revised the manuscript accordingly. Details of the methods including analytical tools used have now been provided in the method section. To establish a secretome profile of the niche cells, differentially expressed secretory factors (fold change >2.0, adjusted p-value <0.05) were selected from the list of detected proteins based on the publicly available database UniProt. The term "ligand" is used to refer to these secretory proteins enriched in the niche cells. Further, the analysis of the regulatory networks, in splenic HSPC population, influenced by this set of ligands was performed. For this analysis, even the non-receptor proteins enriched in HSPC population were used. Therefore, agreeing with the reviewer's suggestion that regulation rather than interaction is a better terminology, we have revised the text.

Comment 3. In relation to Figure 3G, I think it is worthwhile to highlight that the CTMs have a higher proportion of higher abundant secretory proteins than STCs (which could indicate that they are more prone to influence their microenvironment).

Authors: We appreciate the reviewer's thorough assessment and are glad that the intended message has been accurately conveyed. We completely agree with the reviewer that along with a higher proportion of secretory proteins in CTMs, their higher abundance further strengthens the importance of capsular myofibroblasts in creating the HSC niche in the spleen tissue. As suggested, we have highlighted this point and made changes in the results section and conclusions in the revised manuscript.

Comment 4. Regarding the secretory proteins, I think the results are over-interpreted when it comes to secretion, since the data is a global proteomics dataset, and the detection of proteins that are known to be secreted does not equal actual secretion (some proteins can play a role intracellularly as well). And a higher intracellular expression of a 'secretory protein' does not necessarily mean that it is secreted from those cells to a higher degree; the other cell type could still have a higher secretion activity. Without data on actual secretion (e.g. secretome analysis from in vitro cultures or proteomic data of extracellular fluid of the spleen), I think the authors can tone down the claims on secretion. Amongst other things, on page 8, the expression "the ligand was exclusively secreted by.." could e.g. be changed to "the ligand was a secretory protein found higher expressed in CTMs".

Authors: We agree with the comment made by the reviewer that detection of a secretory protein cannot conclude its secretion into the hematopoietic niche. As per the suggestion made by the reviewer, we have amended the text related to these results and added the following statement for more accurate representation.

"We found that out of the 80 target pathways detected in splenic LSK cells, for an overwhelming proportion (56 in number), the ligand was a secretory protein found higher expressed in CTMs. In contrast, regulation of only 8 was influenced by the ligands expressed higher in the STCs. For 16 of the detected pathways, the secretory proteins were detected in both of the cell types".

Minor concerns and suggestions:

Comment 5. Figure legend of Figure 2B: I think 'upper panel' and 'lower panel' have been mixed up with 'left panel' and 'right panel'.

Authors: We thank the reviewer for pointing out this mistake that happened during figure format adjustment. Correction has been made in the revised manuscript.

Comment 6. In the methods on page 20, 'MS/MS spectra recorded from 200 to 2000 scans' should be 'MS/MS spectra recorded from 200 to 2000 mass/charge ratio'.

Authors: This is a good suggestion and we appreciate the input from the reviewer. The recommended change has been incorporated in the revised manuscript.

Comment 7. The methods description for protein identification merit details on missed cleavages allowed as well as the settings for variable or fixed modifications (particularly since there was an additional step of reduction with DTT in the sample preparation) and FDR rate.

Authors: We thank the reviewer for going through the manuscript very carefully. We appreciate the suggestion made and have elaborated the description of methodology used. The parameters for protein identification were employed in the Sequest ST node of the processing workflow within the Proteome Discoverer software. The maximum number of missed cleavage sites was set to 2, and the detected peptides with lengths ranging from a minimum of 6 to a maximum of 144 amino acids were considered for the analysis. Dynamic modification was specified for methionine oxidation (+15.995 Da) with a maximum of three modifications per peptide. While no dynamic modifications were applied for the peptide terminus, the protein N-terminus modification was specified for acetylation (+42.011 Da). Static modification was specified for cysteine carbamidomethylation (+57.021 Da). Decoy database q-values were used to calculate the false discovery rate (FDR) in the Percolator node. At the peptide spectral match level, the data was filtered using a strict FDR threshold of 0.01 and relaxed FDR threshold of 0.05. These details are now incorporated in the methods section of the revised manuscript.

Comment 8. On page 7, the STCs have been defined as Lin-CD45+ instead of Lin-CD45-.

Authors: The stromal cells used for proteomic analysis were in deed Lin-CD45- and not Lin-CD45+. This error has been corrected in the revised manuscript.

Comment 9. On page 7, 'The cells were thermally denatured' is an incorrect expression since it is the proteins that are referred to.

Authors: This was an inadvertent mistake and has been corrected in the revised manuscript.

Comment 10. Figure 3A: It looks like the CTMs have been part of the FACS sort - placing them more logically outside the FACS plot area would avoid misunderstandings.

Authors: We agree with the reviewer and thank for this suggestion. We have made changes in the schematic to represent the workflow more correctly.

Comment 11. Figures 3D and E: In three pairs of comparisons, I find that it makes little sense to display the overlap separate for "up-" and "down-regulated" proteins, since the direction of that expression difference will depend on how the authors have selected the numerator (and denominator). I suggest either one figure, with three circles of "differentially expressed" (i.e. up- and down-regulated together for each comparison), or six figures, of 2 circles each, showing overlap of differentially expressed proteins between pairs

Fig. 6. Venn diagram analysis of differentially enriched proteins: Venn diagram analysis of the differentially enriched proteins detected in the capsular trabecular myofibroblasts (CTMs), splenic stromal cells (STCs), and splenic LSK cells. The numbers indicated common and exclusive differentially enriched proteins (fold change >2 and adjusted p-value <0.05) between each pair of cell populations.

with the same denominator and same direction of change, e.g. up-regulated in CTMs vs STCs and upregulated in LSKs vs STCs.

Authors: Following this suggestion, we have revised the figure to present the overall differentially expressed proteins, rather than separately displaying upregulated and downregulated proteins. Furthermore, to ensure consistency throughout the manuscript, we have incorporated the "adjusted p-values" in this analysis instead of the previously used "p-values". These modifications have been implemented in the revised version of the manuscript.

Comment 12. The legend for Figure 3D and 3E does not state that fold change of >2 was used, while the main text referring to those figures on page 7 do.

Authors: In deed, fold change of >2 was used in the data presented in Fig. 3D and F (now Figure 8D in the revised manuscript) as the results section of the manuscript indicate. We agree with the reviewer that this information might be relevant in the figure legends and will be useful to the reader for better understanding. We have made changes in the text and added this information in the figure legend.

Comment 13. The sentence "Next, we identified the differentially expressed proteins ..." comes a bit out of place since the differentially expressed proteins have already been mentioned and defined in the previous sentence.

Authors: We agree with the reviewer's observation and have removed the unnecessary repetition in the mentioned sentence. The text is accordingly modified in the revised manuscript and is marked along with the other changes.

Comment 14. Heatmaps in Figure 3C, and Appendix Figure S2 F and C: the colour bars need to show what ratio it refers to so the reader knows which colour is high and what is low expression, respectively. In addition, the colour "direction" has been reversed in panel S2F compared to the others - ideally they would use the same direction to avoid confusion.

Authors: We recognize this oversight and appreciate the observation made by the reviewer. This was an oversight that we have rectified following the reviewer's comment. The color scale bars represent the expression levels as Z-scores (row-scaled \log_2 FC). Following the reviewer's comment, we have ensured that all color bars follow the same orientation. The updated plots have been included in the revised manuscript.

Referee #3:

In this study, the authors identify a capsular myofibroblastic niche in the spleen that supports HSCs under steady-state conditions. Through imaging analysis, they show that HSCs are closer to the capsule than HPSCs, with quiescent HSCs even more spatially linked to myofibroblasts. Proteomic profiling shows that myofibroblasts express key hematopoietic regulators, like high levels of CXCL12/SDF-1 α , and pathway analysis supports that they secrete most ligands influencing HSPCs. To test the functional relevance, the authors performed a G-CSF treatment as a tool to reduce SDF-1 α levels, which led to altered HSC spatial distribution in the spleen. Similarly, through 5-FU-induced myeloablation, they linked HSC proliferation to their spatial positioning, reinforcing the connection between quiescence and the niche. Overall, their findings highlight the splenic capsular myofibroblastic niche as an important microenvironment for HSC maintenance.

Comment 1. The study is well executed, and the experiments support their conclusions. That said, one aspect that remains unclear to us is whether the changes in HSC spatial distribution following 5-FU or G-CSF treatment involve resident splenic HSCs or mobilized HSCs from the bone marrow.

Authors: This was in deed a challenging question for us. This was primarily because there is no reliable method to distinguish between the BM and splenic HSCs. In fact, they were shown to be functionally equivalent (Morita Y et al. in *Exp. Hematol.* 2011). Based on our immunostaining method also (lin-CD41-CD48-c-kit+CD150+ cells as pHSCs), it was not possible to distinguish the two populations. Recently, Terumasa Umemoto's group at IRCMS, Japan used Kaede, a photoconvertible fluorescent protein system to mark BM HSCs and showed that the BM HSCs in deed get mobilized to the spleen following 5-FU treatment (Johansson A et al. in *Exp. Hematol.* 2024). However, HSCs detected in the spleen tissues remained negligible upto a period of 8-days after 5-FU treatment. While these results indicated that our data might largely be based on spleen resident pHSCs, we wished to confirm these results by using

Figure 7. Transplanted HSPCs do not home in the splenic hematopoietic niche upto 8 days of 5-FU treatment: (A) Flow cytometry analysis to detect transplanted CFSE+ HSPC (Lin-CD41-CD48-c-kit+) populations within the MNCs from a spleen of 5-FU treated mice one day after transplantation with CFSE labelled Lin-depleted BM cells. (B) Confocal images of spleen CS immunostained to localize transplanted CFSE+ HSPCs one day after transplantation (scale bar=40 μ m). (C) Flow cytometry analysis to examine transplanted CFSE+ HSPC population within the MNCs from the spleen tissue of 5-FU treated mice seven days after transplantation. (D) Confocal images of spleen CS immunostained to localize CFSE+ transplanted HSPCs seven days after transplantation (scale bar=50 μ m).

transplanted HSCs as a proxy for mobilized BM HSCs (Fig. 7). For these experiments, we used CFSE labelled lineage-depleted BM mononuclear cells. To detect the transplanted HSPCs in the spleen tissue, we transplanted upto 10 million lin-depleted cells in mice a day after 5-FU treatment. We started observing donor-derived cells within 24 h post-transplantation (Fig. 7A,B). However, we failed to detect HSPCs within the spleen homed CFSE⁺ population by flow cytometry as well as microscopy upto a period of 7-days (Fig. 7C,D). As the pHSC spatial distribution following 5-FU treatment was analyzed after 7-days, we concluded that the data presented does not have representation of mobilized BM HSCs and is based on the spleen resident HSCs. This is consistent with the results obtained using Kaede mouse model.

In our ongoing studies, we are trying to understand the mechanisms underlying homing of BM derived HSCs during extramedullary hematopoiesis. This is being done in order to inhibit splenic homing for increasing HSC recovery following G-CSF based mobilization regimen in BM donors. We have gained significant insight in this aspect and have unearthed mechanisms underlying this process. We are currently looking into the homing of BM HSCs in splenic niche in greater detail in that context. To not compromise the ongoing studies, we would like to not include this data in the revised manuscript and share further details in the future studies.

Comment 2. Additionally, further comparison of how these findings relate to the BM niche would provide valuable context and further emphasize the importance of understanding niche-specific regulation of HSCs.

Authors: Again, this is an extremely important question that is not easy to answer directly. This is majorly because these two hematopoietic sites differ hugely in their cellular composition and physical makeup. However, in response to this question, we wished to take first steps to understand the differences between the HSC niches in the BM and spleen. To this end, we targeted the key component of the HSC niche in the BM, namely the vasculature, with which we can draw parallel in spleen tissue as well. While initial imaging based studies had identified HSCs located close to sinusoidal endothelium (Kiel MJ et al. in Cell 2005), detailed distance analysis identified the importance of arteriolar (Kunisaki Y et al. in Nature 2013) as well as sinusoidal (Acar M et al. in Nature 2013) niches for BM HSCs. Additionally, using cell-specific deletion of key hematopoietic regulators, the involvement of endothelial and perivascular cells in the establishment of BM HSC niche was demonstrated (Ding L et al. in Nature 2012; Ding L and Morrison SJ in Nature 2013). The vasculature and its involvement in determining spatial distribution of HSCs in spleen was also the subject of major queries from reviewer 1. Therefore, we made significant effort in this direction during the revision of the manuscript. We believe that it has brought context to our study in the backdrop of understanding that we have about the BM HSC niche.

The experiments performed in response to the queries from reviewer 1 did not present any correlation between the vasculature and HSC localization in spleen tissue. We extended these studies to perform detailed spatial distribution analysis of HSPCs (identified as lin⁻CD41⁻CD48⁻c-kit⁺ cells) with reference to the vasculature identified by endothelial marker CD31 (Fig. 8A). With reference to the vasculature, we did not observe any significant difference in the mean distance (Fig. 8B) or spatial distribution (Fig. 8C) of HSPCs when compared with the random dots. In addition, we showed that the spatial distribution of HSPCs relative to the vasculature did not change following G-CSF treatment (Fig. 8D-F).

In the BM niche, it has been reported that the quiescent and proliferative HSCs occupy differential location with reference to the arterioles and Ng2⁺ periaarteriolar cells (Kunisaki Y et al. in Nature 2013). Our experiments to assess the distribution of proliferative (Ki-67⁺) and non-proliferative (Ki-67⁻) HSPCs along the vasculature showed no such differences (Fig. 8G-I). We also examined if the presence of lymphatic vessels impacted the localization of HSPCs in spleen tissue. Again, we did not observe any significant difference in their spatial distribution as compared to the random dots when referenced to the lymphatic vessels identified by Lyve-1 expression (Fig. 8J-L). Hence, extensive analysis of spatial distribution of HSPCs showed that vasculature was not crucial for the creation of hematopoietic niche in spleen. Our studies on the distribution of vasculature and micro-location of HSPCs place splenic capsular hematopoietic niche in a unique position in comparison to the BM and presents an attractive model to understand their functional regulation. The entire set of data obtained through the studies carried out to study the role of vasculature in splenic hematopoietic niche has been added in Figure 3 and 5 of the revised manuscript.

Figure 8. Distribution of splenic HSCs with reference to vasculature. (A) Immunostaining followed by confocal imaging was performed and pseudo-surfaces for HSPCs (illuminated yellow) and blood vessels (CD31+) (illuminated Salmon) were generated using Imaris. Scale bars=40 μ m. (B) Comparison of Euclidean distances measured for HSPCs and RDs from blood vessels in splenic sections. (C) Spatial distribution frequency of HSPCs and RDs at sequential intervals (30 μ m each) from blood vessels. (D) Immunostaining followed by confocal imaging was performed and pseudo-surfaces for HSPCs (illuminated yellow) and blood vessels (CD31+) (illuminated Salmon) in splenic CS of G-CSF treated mice. Scale bars=40 μ m. (E) Comparison of Euclidean distances measured for HSPCs from blood vessels in splenic sections from control and G-CSF treated mice. (F) Spatial distribution frequency of HSPCs at sequential intervals (30 μ m each) from blood vessels in splenic CS of control and G-CSF treated mice. (G) Confocal images of spleen CS immunostained to localize Ki-67+ and Ki-67- HSPCs along with CD31+ vessels. Scale bars=20 μ m. (H) Comparison of Euclidean distances between Ki-67- and Ki-67+ HSPCs from the blood vessels. (I) Distribution frequency of Ki-67- and Ki-67+ HSPCs at sequential intervals (30 μ m each) from the blood vessels. (J) Confocal images of spleen CS immunostained to localize HSPCs along with Lyve-1+ lymphatic vessels. Scale bars=30 μ m. (K) Comparison of Euclidean distances between HSPCs and RDs from the lymphatic vessels. (L) Distribution frequency of HSPCs and RDs at sequential intervals (30 μ m each) from the lymphatic vessels.

Other/Minor comments:

Comment 1: Typo in the title: "myofibroblastic" should be corrected.

Authors: We apologize for this glaring error in the manuscript title. Thanks to the reviewer for pointing it out. It has been corrected in the revised manuscript.

Comment 2: Page 7: "Niche samples were prepared from FACS-sorted Lin-CD45+ splenic cells (hereafter referred to as stromal cells or STCs) and myofibroblasts from the fibrous splenic tissue (hereafter referred to

as capsular and trabecular myofibroblasts or CTMs)". - Do the authors mean Lin-CD45- or Lin-CD45+ cells? Please clarify.

Authors: The reviewer is correct in identifying this error; the stromal cells used for proteomic analysis were in deed Lin-CD45-, and not Lin-CD45+. This error has been corrected in the revised manuscript.

Comment 3: Figure 8: The authors refer to quiescent HSCs as "dormant." It would be more accurate to use "quiescent" to avoid any potential misinterpretation.

Authors: We agree with the suggestion made by the reviewer. We have implemented this change in the revised text and have used the term "quiescent pHSCs" instead of "dormant pHSCs".

Comment 4: Methodology: While the authors describe the proteomics workflow, including the population and downstream analysis, they do not provide information on the number of cells used. This should be included for clarity and reproducibility.

Authors: In conjunction with similar comments from the second reviewer, we realized the need to expand the description of methods used for proteomic analysis. This point raised by the reviewer in specific was an inadvertent omission and in deed a key information required; it has now been included in the revised manuscript. Apart from this, significant addition has been made on the data analysis methods used to arrive at the results presented in the manuscript.

Comment 5. The authors use two different stress conditions to examine how the splenic niche affects HSC spatial distribution. This is an interesting finding. Could the authors discuss its translational relevance, particularly in the context of extramedullary hematopoiesis? How might this knowledge be applied in clinical or pathological settings?

Authors: It has been known that spleen hosts a significant number of HSCs even at steady state (Wolber FM et al. in *Exp. Hematol.* 2002). However, it was only the extramedullary hematopoietic activity in the wake of defective BM function that has been the centre of exploration. We found it interesting and useful at the same time to identify the niche for these cells as important clues for the regulation of HSC function can be revealed. We started studying spleen hematopoiesis with the following long-term objectives;

1. To understand the differences between the spleen and BM HSCs that can unearth specific markers to distinguish between the cells from these two niches, in addition to novel HSC-intrinsic regulators.
2. Compared to the splenic niche described here, the hugely heterogeneous BM niche has been hard to understand and mimic for ex vivo expansion. This study will pave way to identify novel pathways that keep splenic HSCs in poised state of proliferation rather than quiescent.
3. In the case of BM transplantation and HSC mobilization in donors, a significant entrapment of HSCs in spleen tissue happens. We are targeting mechanisms of spleen homing to prevent this loss through studies on splenic hematopoiesis, which could have direct clinical value.

These objectives in the form of potential clinical applications have now been added in the discussion section and hope they will motivate the audience to further explore the functioning of splenic hematopoiesis.

Annexe Table 1. Linking results presented in the response letter with manuscript figures.

Sr. No.	Rebuttal Fig.	Description	Manuscript Fig.
1	1A	pHSC localization in splenic CS of male and female mice	2A
2	1B	pHSC distribution in splenic CS of male and female mice	2B
3	1C	pHSC localization in splenic CS of male and female mice after G-CSF treatment	EV3W
4	1D	pHSC localization in splenic CS of male and female mice after 5-FU treatment	EV4C
5	2A	Immunolocalization of blood vessels in spleen tissue section	3A
6	2B	Blood vessel distribution in RP and WP areas of spleen	3B
7	2C	Blood vessel distribution in capsular zone and inner core of spleen	3C
8	3A	Immunostaining to identify lymphatic vessel in spleen	EV2A
9	3B	Blood and lymphatic vessel distribution in splenic CS	EV2B
10	3C	Lymphatic vessel distribution in capsular zone and inner core of spleen	EV2C
11	4A	Comparison of CS area of the spleen from control and G-CSF treated mice	5A
12	4B	Comparison of cellularity of the spleen from control and G-CSF treated mice	5B
13	4C	Flow cytometry analysis to analyze the cell cycle status of HSCs	EV3X
14	4D	Comparison of proportion of splenic HSC population in different stages of cell cycle	5C
15	4E	pHSC localization in in the spleen from control, G-CSF and G-CSF Rev groups of mice	5D
16	4F	pHSC distribution in in the spleen from control, G-CSF and G-CSF Rev groups of mice	5E
17	5A	Selection marker combination to identify primitive HSCs based on three fluorophores	EV1A
18	5B	The revised proportion of pHSCs in three marker combinations	1A
19	5C	The first version of proportion of pHSCs in three marker combinations	Replaced
20	6	Venn diagram analysis of differentially enriched proteins	8D
21	7A	Flow cytometry analysis to detect CFSE ⁺ HSPC one day after transplantation	S3A
22	7B	Immunostaining to localize CFSE ⁺ HSPCs one day after transplantation in spleen	S3B
23	7C	Flow cytometry analysis to detect CFSE ⁺ HSPC seven days after transplantation	S3C
24	7D	Immunostaining to localize CFSE ⁺ HSPCs seven days after transplantation in spleen	S3D
25	8A	Immunolocalization of HSPCs along with BV in splenic CS	3D
26	8B	Comparison of HSPCs and RDs localization from BV in splenic CS	3E
27	8C	Distribution of HSPCs and RDs localization from BV in splenic CS	3F
28	8D	Immunostaining of HSPCs along with BV in spleen CS after G-CSF treatment	5F
29	8E	HSPCs localization from BV in splenic CS from control and G-CSF treated mice	5G
30	8F	HSPCs distribution from BV in splenic CS from control and G-CSF treated mice	5H
31	8G	Immunostaining of Ki-67 ⁺ and Ki-67 ⁻ HSPCs along with BV in spleen CS	3G
32	8H	Comparison of Ki-67 ⁺ and Ki-67 ⁻ HSPCs localization from BV in splenic CS	3H
33	8I	Distribution of Ki-67 ⁺ and Ki-67 ⁻ HSPCs localization from BV in splenic CS	3I
34	8J	Immunolocalization of HSPCs along with LV in splenic CS	3J
35	8K	Comparison of HSPCs and RDs localization from LV in splenic CS	3K
36	8L	Distribution of HSPCs and RDs localization from LV in splenic CS	3L

Dear Dr Khurana,

Thank you for submitting your revised manuscript (EMBOJ-2024-119500R) to The EMBO Journal, as well for your patience with our response. Your amended study was sent back to the referees for their scientific reassessment, and we have received re-reports from all of them, which I enclose below. As you will see, the experts state that the work has been substantially enhanced by the revisions and they are now broadly in favour of publication, pending minor amendments.

Thus, we are pleased to inform you that your manuscript has been accepted in principle for publication in The EMBO Journal.

Please carefully consider the remaining minor points raised by referee #2 by enhancing the methods annotation and complementing the manuscript text where appropriate.

Also, we now need you to take care of a number of issues related to formatting and data presentation as detailed below, which should be addressed at re-submission.

Please contact me at any time if you have additional questions related to below points.

As you might have noted from our webpage, every paper at the EMBO Journal now includes a 'Synopsis', displayed on the html and freely accessible to all readers. Besides a 'model' figure, the synopsis also comprises 2-5 one-short-sentence bullet points that summarize the article. I would appreciate if you could provide these bullet points. Consequently, please remove/simplify the textual annotation in your synopsis figure.

Thank you for giving us the chance to consider your manuscript for The EMBO Journal. I look forward to your final revision.

Again, please contact me at any time if you need any help or have further questions.

Kind regards,

Daniel Klimmeck

>> Please add up to five keywords to your study.

>> Author Contributions: Remove the author contributions information from the manuscript text. Note that CRediT has replaced the traditional author contributions section as of now because it offers a systematic machine-readable author contributions format that allows for more effective research assessment. and use the free text boxes beneath each contributing author's name to add specific details on the author's contribution.

More information is available in our guide to authors.
<https://www.embopress.org/page/journal/14602075/authorguide>

>> Adjust the title of the 'Conflict of Interest' section to 'Disclosure and Competing Interests Statement' and move after Acknowledgements.

>> Section order should be corrected as follows: Title page - Abstract & Keywords - Introduction - Results - Discussion - Methods - Data Availability - Acknowledgements - Disclosure and Competing Interests Statement - References - Figure Legends - Table(s) - Expanded View Figure Legends.

>>Appendix: please remove the blue font in the final version and upload as PDF.

>> Reagents and Tools table: provide as a separate file using the existing template in the Guide For Authors, listing key reagents, experimental models, software and relevant equipment.

>> Figure callouts: Ensure that Figure 3K is called out.

>> Data availability section: the "Code Availability" paragraph should be merged with the Data Availability section'. Please remove referee tokens and ensure MassIVE and ProteomeXchange datasets are made publicly accessible.

>> Remove the "graphical abstract" paragraph from the manuscript and integrate it into the synopsis text.

>> Remove the 'data not shown' statement on p.12, or add respective information.

>> Funding: please merge with the Acknowledgements section. Please enter the following funding information into our online system: 'Indian Institute of Science Education and Research Thiruvananthapuram, INSPIRE fellowship from Department of Science and Technology, Government of India and E-Grantz scholarship from Government of Kerala'.

>> Please cite your previous Pv et al (2024) (PMID 28094257) study in the Methods section.

>> Figure checks:

>>>> please indicate redisplay of content Figure 1C in the Figure legend for Figure 1D.

>>>> please indicate redisplay of content Figure 7E in the Figure legend for Figure EV3X.

>> Consider additional changes and comments from our production team as indicated below:

- Figure legends:

1. Please note that the exact p values are not provided in the legends of figures 1A, E, G, I, J; 2A, B, C, F, G; 3B, 4D, E, F, G, I, J, L, M, N, O, P; 5A, B, C, D, E, H; 6D; 7A, C, D, F, H, J, K; EV1 C, E, F; EV2 B, C, F, G, H, I, J, K, L; EV3 A, C, G, H, I, J, K, L, M, N, O, P, Q, R, S, T, U, V, W; EV4 A-F, H, J, K, L, M, N, O, P, Q; S1A-C.

2. Please indicate the statistical test used for data analysis in the legends of figures 1A, E, G, I, J; 2A, B, C, F, G, H, I; 3B, C, E, F, H, I, K, L; 4D, E, F, G, I, J, K, L, M, N, O, P; 5A, B, C, D, E, G, H; 6D, F, G, H, I; 7A, C, D, F, H, J, K; 8E, F; EV1 C, E, F; EV2 B, C, F, G, H, I, J, K, L; EV3 A-W; EV4 A-Q; EV5 A, B, E; S1A-C; S2 B, C.

3. Please note that the box plots need to be defined in terms of centre, percentile in the legends of figures 1A, G; 2A, C, F, G; 3B, C, E, H, K; 4D, E, F, G, I, M, K, O; 5A, B, D, G; 6D; 7A, C, D, F, H, J; EV1 C, E; EV2 B, C, H; EV3 A-E, G-K, M-Q, S, U, W; EV4 A-D, F, H, J, L, N, P.

4. Please note that the box plots need to be defined in terms of minima, maxima, centre, bounds of box and whiskers, and percentile in the legends of figures S1 A-C.

5. Please note that information related to n is missing in the legends of figures 1J, 2B, H, I; 3F, I, L; 4J, N, L, P; 5E, H; 6H, I; 7K, 8E, F; EV1 F, G; EV3 A-E, G-K, M-Q, T, V; EV4 E, G, I, K, M, O, Q; S1 C, S2 C

Referee #1:

This is a revised version of the previously submitted manuscript from Mehatre and colleagues. The authors have performed a thorough revision of the manuscript and included extended series of novel observations and data.

The authors have fully addressed all of my comments, and therefore I recommend this work to be accepted for publication.

Referee #2:

The authors have improved the manuscript and made many aspects easier understood, which enhances the value and impact of the paper. I am certain the authors can address the remaining smaller points below and look forward to the publication of this interesting study.

All my comments have been fully addressed, except Comment 2, where the following remains:

- What R package was used for the statistical analysis and what approach was used for multiple testing correction?
- The details of the Reactome pathway enrichment analysis are still unclear. The legends of Figs EV5D and EV5E state 'based on global protein profile' while the methods state 'Proteins upregulated with fold change >2.0 and adjusted p-value <0.05 were taken into account for pathway analysis'. If the latter is true, it would be appropriate to clarify this filter also in the legends.
- The analysis around Fig 8G is presumably still very valuable but needs further clarification, because the explanation of the extraction of the 'regulators' of the pathways is missing. 'We selected the pathways regulated by the secretory factors...' does not tell the reader what approach was used for the match between pathway and its regulator (secretory factor). Are the regulators of each pathway defined in Reactome? If there was a specific R package used for extracting the regulators that needs to be stated.
- The authors still refer to 'comparative interaction studies' (on page 16) rather than 'regulatory studies' or similar.

Referee #3:

The authors have now improved the manuscript, and it is suitable for publication.

Authors' response to the reviewers' comments:

We are indeed pleased to note that the reviewers have found the concerns raised by them addressed satisfactorily. We sincerely thank the reviewers for their valuable time, careful evaluation, and constructive feedback throughout the review process. We are particularly grateful for the thoughtful suggestions that guided us in addressing key conceptual and methodological aspects, which have ultimately enhanced the presentation and impact of our findings. We are also thankful to the editorial team for facilitating a smooth and supportive review process.

Referee #1:

This is a revised version of the previously submitted manuscript from Mehatre and colleagues. The authors have performed a thorough revision of the manuscript and included extended series of novel observations and data.

The authors have fully addressed all of my comments, and therefore I recommend this work to be accepted for publication.

Authors: We are pleased that the reviewer has found the revised manuscript has addressed the concerns satisfactorily. We thank the reviewer for constructive and supportive feedback that helped us improve the manuscript.

Referee #2:

The authors have improved the manuscript and made many aspects easier understood, which enhances the value and impact of the paper. I am certain the authors can address the remaining smaller points below and look forward to the publication of this interesting study. All my comments have been fully addressed, except Comment 2, where the following remains:

Authors: We thank the reviewer for evaluating the revised manuscript and finding the changes valuable. The constructive feedback, particularly the insightful methodological suggestions led to an enhanced clarity and overall impact of our findings.

Comment 1. What R package was used for the statistical analysis and what approach was used for multiple testing correction?

Authors: Statistical analysis was performed using the Limma package (version 3.50.3) in R/Bioconductor, and p-values were adjusted for multiple testing using the Benjamini and Hochberg (BH) method. These details have now been added in the revised manuscript.

Comment 2. The details of the Reactome pathway enrichment analysis are still unclear. The legends of Figs EV5D and EV5E state 'based on global protein profile' while the methods state 'Proteins upregulated with fold change >2.0 and adjusted p-value <0.05 were taken into account for pathway analysis'. If the latter is true, it would be appropriate to clarify this filter also in the legends.

Authors: We thank the reviewer for pointing out this mistake. The text is accordingly modified in the revised manuscript and is marked along with the other changes.

Comment 3. The analysis around Fig 8G is presumably still very valuable but needs further clarification, because the explanation of the extraction of the 'regulators' of the pathways is missing. 'We selected the pathways regulated by the secretory factors...' does not tell the reader what approach was used for the match between pathway and its regulator (secretory factor). Are the regulators of each pathway defined in Reactome? If there was a specific R package used for extracting the regulators that needs to be stated.

Authors: We have implemented the revisions based on the the reviewer's suggestions and updated the manuscript accordingly. The analytical tools and approaches employed have been detailed in the 'Methods' section. Pathway selection was based on the involvement of secretory proteins, as annotated in the *UniProt2Reactome.txt* file from the Reactome database.

Comment 4. The authors still refer to 'comparative interaction studies' (on page 16) rather than 'regulatory studies' or similar.

Authors: We agree with the reviewer's observation, and the text has been revised accordingly in the manuscript, with all modifications clearly marked alongside the other changes.

Referee #3:

The authors have now improved the manuscript, and it is suitable for publication.

Authors: We thank the reviewer for going through the manuscript and finding the revised manuscript suitable for publication. The insightful suggestions provided have been valuable in refining the key conceptual aspects of our study, thereby enhancing both the clarity and the overall impact of our findings.

Dear Dr Khurana,

Thank you for submitting the revised version of your manuscript. I have now evaluated your amended manuscript and concluded that the remaining minor concerns have been sufficiently addressed.

I am thus pleased to inform you that your manuscript has been accepted for publication in the EMBO Journal.

On a different note, I would like to alert you that EMBO Press offers a format for a video-synopsis of work published with us, which essentially is a short, author-generated film explaining the core findings in hand drawings, and, as we believe, can be very useful to increase visibility of the work. Please see the following link for representative examples and their integration into the article web page:

<https://www.embopress.org/doi/full/10.15252/emj.2019103932>

Best regards,

Daniel Klimmeck

Daniel Klimmeck, PhD
Senior Editor
The EMBO Journal
EMBO
Postfach 1022-40
Meyerhofstrasse 1
D-69117 Heidelberg
contact@embojournal.org